# The role of land use in terrestrial support of boreal lake food webs

Ossi Keva [1,2] ✉, Matthew R. D. Cobain [1], Antti P. Eloranta [1], Heikki Hämäläinen [1], Mikko Kiljunen [1], Jos Schilder[1,3] & Roger I. Jones [1]

There is growing awareness of the importance of cross-boundary energy and nutrient transfers between adjacent ecosystems. Lake ecosystems receive inputs of terrestrial organic matter that microbes can make available to higher level consumers. However, how environmental drivers influence this terrestrial support of benthic and pelagic consumers at multiple trophic levels remains underexplored. Using hydrogen stable isotopes as a tracer of terrestrial organic matter, we find large variation in terrestrial support of aquatic consumers (i.e., consumer allochthony) among 35 boreal lakes. Of 19 different aquatic organisms, benthic consumers show the highest allochthony. Consumer allochthony decreases along an environmental gradient from forested to agricultural catchments, likely due to shifts in the origin and nature of lake organic matter. Our results demonstrate how cross-ecosystem transfer of organic matter can influence community dynamics in recipient ecosystems, with anthropogenic management of donor terrestrial ecosystems affecting the structure and function of food webs in recipient aquatic ecosystems.

Land use, such as forestry, agriculture, mining, and human settlements, is one of the major environmental issues threatening biodiversity and vital ecosystem processes and services locally and globally[1–3]. Freshwater ecosystems integrate these human impacts on terrestrial ecosystems by retaining substances like pollutants, nutrients and organic matter that flow from the surrounding land[4]. Recognising such cross-ecosystem linkages is vital for holistic understanding, sustainable management and conservation of ecosystems and their vulnerable biota[3,5,6]. The transport of organic and mineral nutrients from surrounding catchment areas is the key driver of the trophic state and energy dynamics of lake ecosystems[7,8] that process both internally generated (autochthonous) aquatic organic matter and externally derived (allochthonous) terrestrial organic matter (t-OM)[9]. The allochthonous input of t-OM has been shown, in experimental settings with a limited number of replicate systems, to influence the structure and functioning of lake food webs[10]. Although the particulate form of t-OM (t-POM) with its associated microbial community can be grazed directly, the dissolved form (t-DOM) can be exploited by lake bacterial communities and also made available to higher-level consumers via this microbial link[11,12]. Typically, the bacteria can be consumed by protozoans, while larger metazoan crustacean zooplankton consume the protozoans, and may also consume bacteria directly. The crustacean zooplankton then form a critical food resource for juvenile and some adult fishes[13]. Concurrently, t-OM mobilised by bacteria and fungi can support detritivorous zoobenthos and benthivorous fish[10,14–18]. However, the degree to which aquatic consumers assimilate organic matter derived from t-OM (hereafter referred to as consumer allochthony) in different habitats, across trophic levels and between lakes remains underexplored[19–21]. Studies of aquatic consumer allochthony have mainly focused on a few functional groups, especially crustacean zooplankton and small pelagic fish, using stable isotope analyses of key elements to differentiate between autochthonous and allochthonous sources of organic matter (OM)[9,10,13,15,18,19,22–24]. Hydrogen stable isotope ratios ($\delta^2$H) have been shown to differentiate autochthonous and allochthonous OM sources particularly well, because water loss by transpiration leads to strong enrichment in the heavy $^2$H isotope in terrestrial plants, a process that is absent for submerged aquatic primary producers[25].

---

[1]Department of Biological and Environmental Science, University of Jyväskylä, Jyväskylä, Finland. [2]Faculty of Biological and Environmental Science, University of Helsinki, Helsinki, Finland. [3]Rijkswaterstaat, Ministry of Infrastructure and Water Management, Lelystad, The Netherlands. ✉e-mail: ossi.keva@helsinki.fi

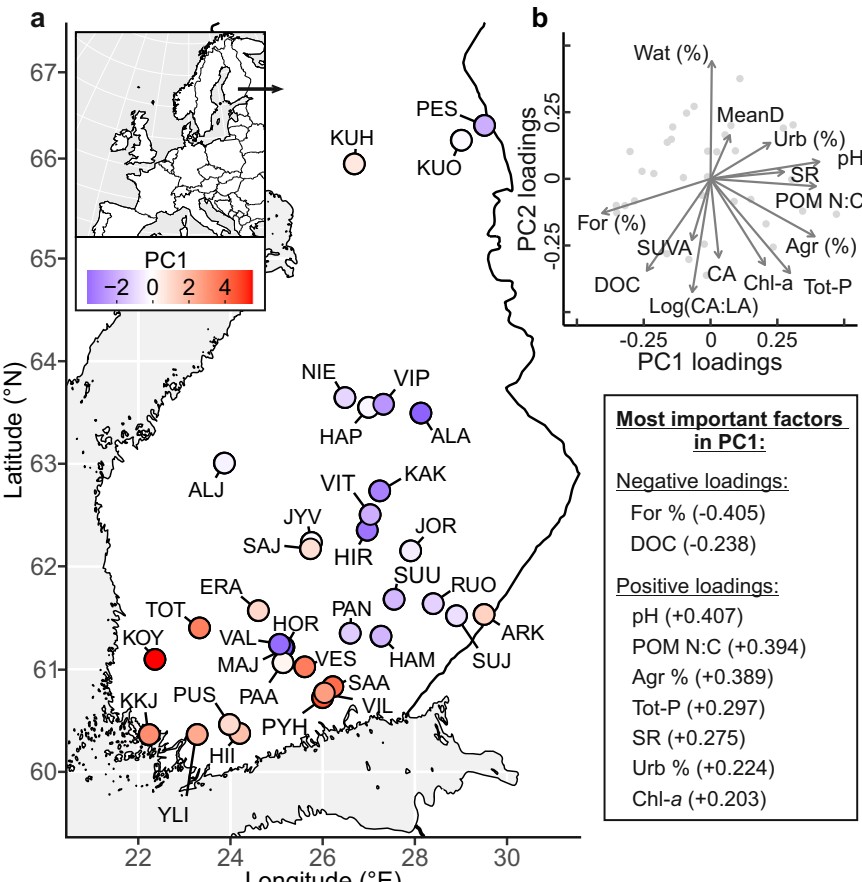

**Fig. 1 | Map of the study lakes and ordinations of their environmental variables.** Locations of the 35 study lakes in Finland, Northern Europe (**a**) and ordinations of different environmental variables included in the principal component analysis (**b**) are presented in different panels. Principal Component 1 (PC1) expresses environmental gradient from lakes with highly forested catchments (For (%)) and high dissolved organic carbon concentrations (DOC) to eutrophic lakes (Tot-P and Chl-*a*) with high agricultural coverage (Agr (%)) in the catchment. See Supplementary Table 1 for abbreviations of the lake names, environmental variables and PC scores. Environmental variables with lower than 0.2 loading (Pearson correlation coefficient) on PC1 or PC2 are not shown in **b**) to make it more readable (see Supplementary Table 2 and Supplementary Fig. 2 for the full set of variable loadings). Source data are provided as a Source Data file.

There is a growing consensus that t-OM can be an important but variable resource for lake food webs[9,10,24]. For example, a recent synthesis incorporating 147 sampled lakes indicated that zooplankton rely on t-OM sources by an average of 11% to 83%[24]. This large variation may be related to several factors, including seasonality[24,26], differences between consumer functional groups, and gradients of catchment characteristics[24]. In addition, appreciable reliance of zoobenthos and fish consumers on t-OM has been recorded in a few previous studies[13,18]. Forest coverage in catchments has been suggested to be one major environmental factor influencing consumer allochthony in lakes[13,23,24]. Other studies have indicated that high trophic state or primary productivity of lakes decrease allochthony of zooplankton and predatory fish[9,23,24]. However, these studies have focused mainly on the pelagic food web compartment or have been conducted in only a few field sites, leaving much unexplained variation in consumer allochthony. Resolving this uncertainty is critical for understanding how changes in human management of terrestrial landscapes resonates in the recipient lake ecosystems[6].

We collected data from all major metazoan food web compartments in 35 boreal lakes and their catchments (Fig. 1) to elucidate the effect of landscape properties on the importance of t-OM sources for 19 consumer groups in the benthic and pelagic food web compartments. We used $\delta^2H$ to differentiate terrestrial and aquatic OM sources and used Bayesian mixing models[27] to generate estimates of consumer allochthony, using measured $\delta^2H$ values of terrestrial inlet

dissolved OM (t-DOM) and photosynthetic fixation-adjusted water $\delta^2H$ values to represent the basal allochthonous and autochthonous resources, respectively (Fig. 2, Fig. 3, see Methods). Nitrogen stable isotope ratios ($\delta^{15}N$) were used to assign the trophic level of consumers, and fish feeding guilds were designated based on stomach contents analysis and existing knowledge of the species' ecology. We used principal component analysis (PCA) to summarise variation in catchment and lake characteristics (Fig. 1, Supplementary Table 2) and this combined environmental gradient was used as a covariate to estimate consumer allochthony in lakes with forested or more agricultural catchments.

We expected that forested catchments generate higher t-OM inputs and thus higher consumer allochthony in lakes than agricultural catchments, where higher input of limiting nutrients facilitates relatively greater within-lake production and thus lower consumer allochthony[13,23,24]. Due to high input of riparian t-POM and direct ingestion of this terrestrial detritus and its associated microbial biofilm, we expected shallow-water littoral benthic primary consumers to show higher allochthony than deep-water pelagic and profundal primary consumers. Moreover, generalist and littoral-oriented fishes were expected to be more responsive to catchment-related drivers as they can better adapt to changes in resource availability, and their food sources are more directly linked to t-OM than those of highly specialised offshore pelagic planktivorous fishes whose food is mainly reliant on phytoplankton-derived energy and nutrient sources[28].

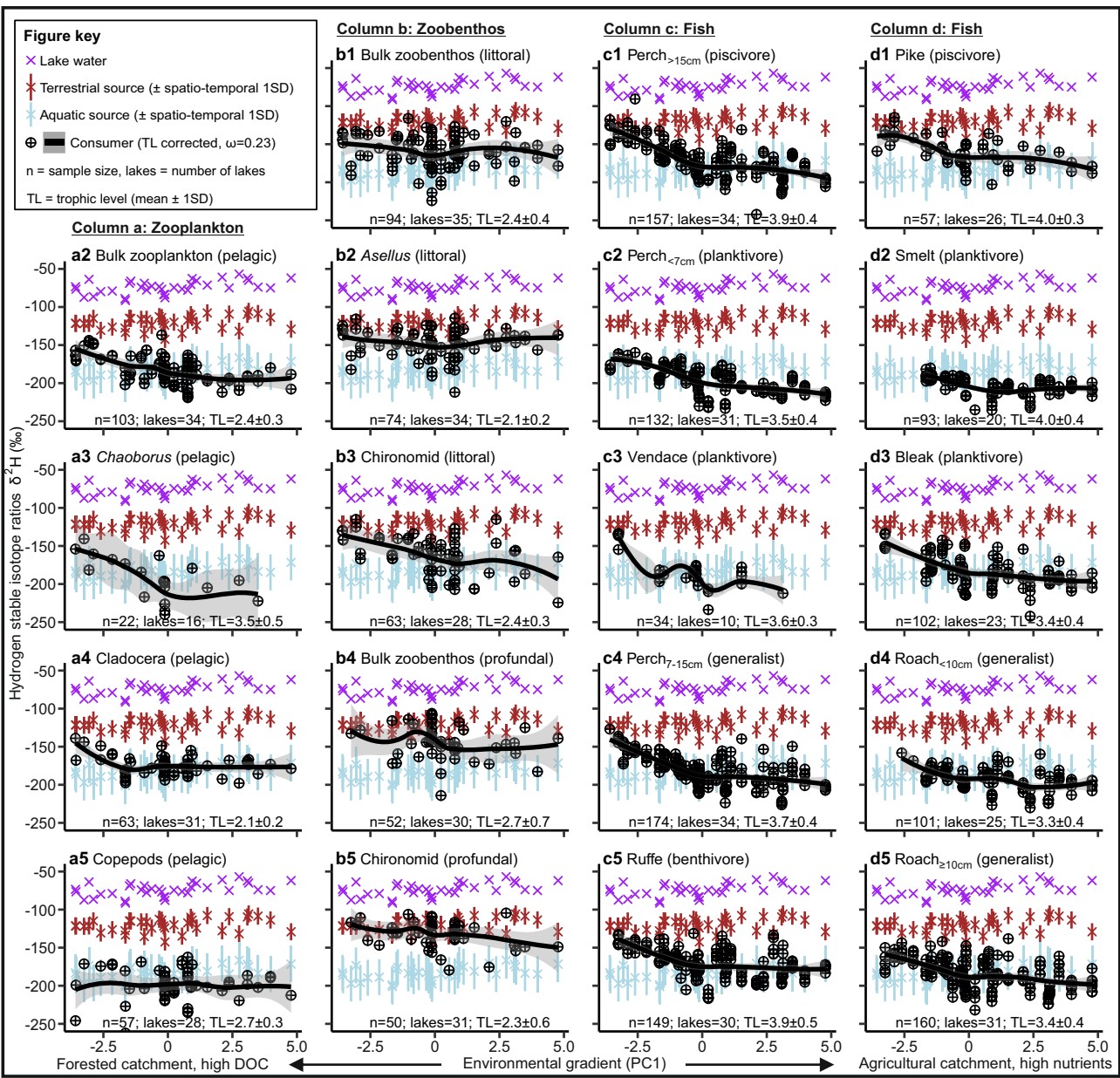

**Fig. 2 | Variability in consumer and source δ²H values along the gradient from high DOC forest lakes to eutrophic lakes in agricultural landscapes.** The consumer groups zooplankton (**a**), zoobenthos (**b**), and fish (**c**, **d**) are presented in separate columns, with text inside the brackets indicating habitats (**a**, **b**) or feeding guilds (**c**, **d**) of consumer groups. Purple, brown and light blue x-marks stand for lake-specific raw δ²H values for lake water, terrestrial and aquatic sources, respectively. Black crossed circles indicate omega corrected δ²H values of consumers (see Methods), with black lines indicating trends (i.e., loess smoother curves, with grey shading depicting 95% confidence intervals) along the studied environmental gradient. The numbers of consumer samples, study lakes and mean ± SD trophic level (TL) of consumers are shown at the bottom of each subplot. Source data are provided as a Source Data file.

In this study, we demonstrate that aquatic consumers from diverse habitats and feeding guilds exhibit distinct levels of allochthony, with benthic primary consumers displaying particularly high allochthony. Moreover, we show that consumer allochthony decreases along an environmental gradient from forested to more agricultural catchments.

## Results

### Environmental data

Environmental data for the 35 lakes and their catchments (in total 18 variables) were combined with principal component analysis (PCA) to reduce the number of environmental dimensions, thereby simplifying the data set and highlighting the variables that account for the majority of variability in lake and catchment characteristics. Although specific to these 35 lakes, this reduction of dimensionality facilitates interpretation, statistical analyses, identification of key environmental gradients, and improved visualisation of patterns and relationships among the lakes (Fig. 1, Supplementary Table 2, Supplementary Fig. 2). The first principal component (PC1) accounted for 27.3% of the total variance in the 18 environmental variables and was subsequently used in statistical analyses. Catchment forest area (Pearson correlation coefficient with PC1: −0.41), agricultural area (+0.39), urban area (+0.22), N:C ratio of POM (+0.39), pH (+0.41), DOC concentration (−0.24), total phosphorus [Tot-P] concentration (+0.30), chlorophyll-a [chl-a] concentration (+0.20), and DOM light absorbance slope range ratio [SR ratio between 275−295 nm and 350−400 nm, see Methods]

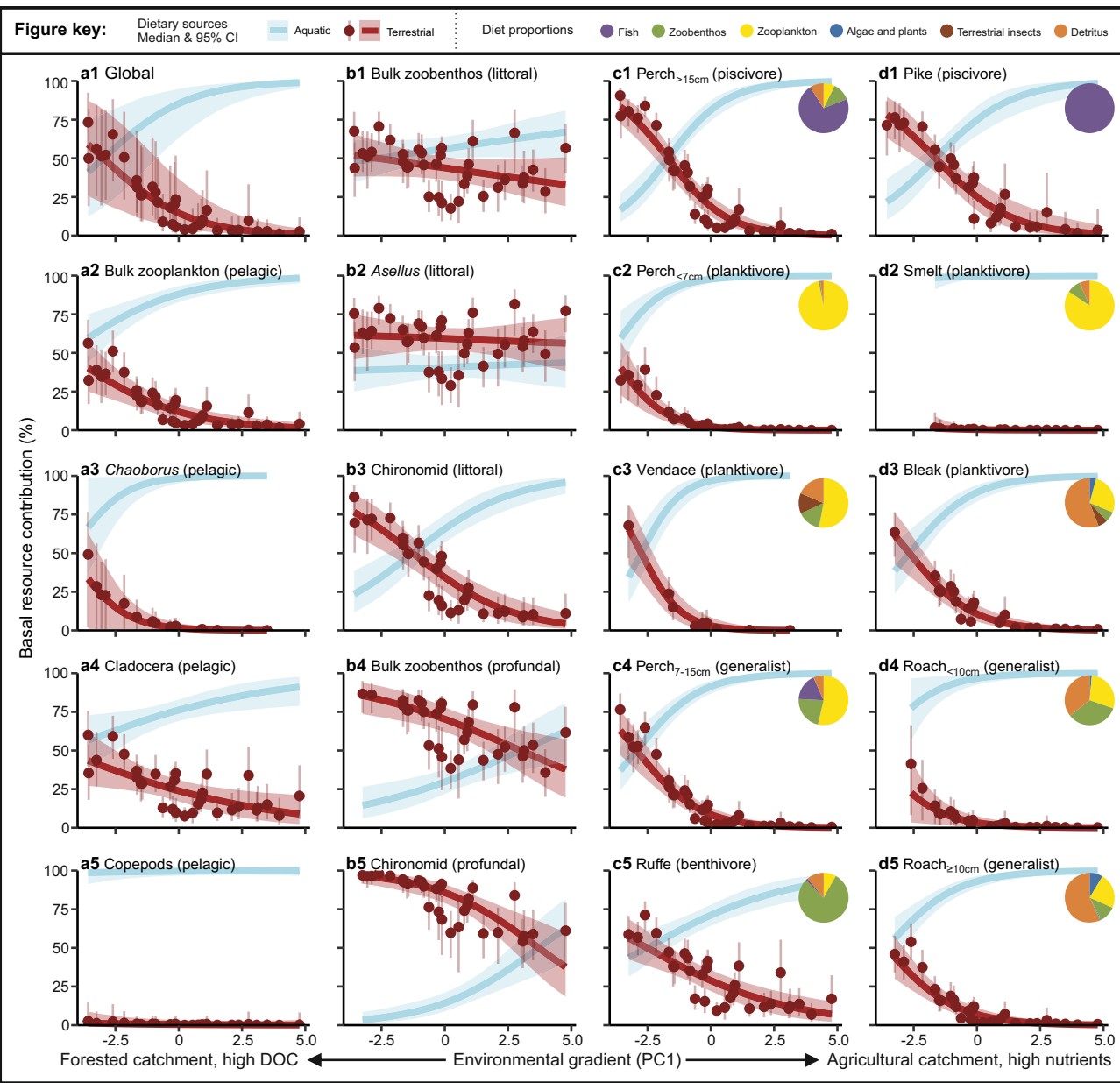

**Fig. 3 | Terrestrial support of different aquatic consumers in 35 boreal lakes.** The lakes cover a marked environmental gradient (PC1) from highly forested, high DOC lakes to nutrient-rich lakes in agricultural landscape. The consumers' reliance on terrestrial sources are estimated with the MixSIAR Bayesian isotopic mixing model comparing δ²H values of consumers to those of terrestrial and aquatic (within-lake) basal resources. Medians and 95% credible intervals of the consumer allochthony are indicated as curves and shaded areas and the lake- and consumer-specific estimates as points and whiskers, respectively. Non-overlapping 95% credible intervals indicate significant shifts in consumer allochthony along the environmental gradient. The trend in global average consumer allochthony is presented in a1. The consumer groups zooplankton (a2–5), zoobenthos (b1–5), and fish (c1–d5) are presented in different columns, with subscript numbers indicating size categories of the fish and text inside the brackets indicating habitats (**a**, **b**) or feeding guilds (**c**, **d**) of consumer groups. The pie charts with a legend in top show proportions of different prey categories found in the fish stomach contents (averaged across the study lakes). Source data are provided as a Source Data file.

(+0.28) had the highest loadings on PC1 (Fig. 1, Supplementary Table 2). Thus, the PC1 scores broadly correspond with an environmental gradient from more acidic, high DOC lakes with forested catchments, towards more eutrophic lakes with high agricultural activities in the catchment. As low N:C ratio of POM and the SR ratio are related to predominance of t-OM in the total lake OM pool[29,30] and they both had high positive loadings on PC1, there is a parallel gradient in the composition of the lake OM from a predominance of terrestrial material towards an increasing contribution from aquatic material. This was supported by the observed negative correlations between catchment forest area and SR ratio (−0.43) and N:C ratio of POM (−0.77) (Supplementary Fig. 1).

## Allochthony of consumers

The Bayesian mixing model revealed that in an environmentally average lake (i.e. PC1 = 0), the median allochthony of bulk zooplankton was 11% (95% credible intervals: 6–16%; species level intercepts shown in Supplementary Table 6). Cladocerans (Diplostraca) exhibited greater allochthony (23%, 15–30%) than copepods (Copepoda: 0%, 0–3%) or phantom midge larvae (*Chaoborus flavicans*, hereafter *Chaoborus*: 1%, 0–3%; Supplementary Table 6). We observed strong decreases in cladoceran and bulk zooplankton allochthony, and to a lesser extent in that of *Chaoborus*, along the environmental gradient (PC1) towards more eutrophic lakes with high agricultural coverage in their catchment (Fig. 3, Supplementary Fig. 6, Supplementary Table 6).

All zoobenthos groups showed higher allochthony than zoo-plankton (Supplementary Table 6). Profundal bulk zoobenthos showed higher allochthony (68%, 60–76%) than littoral bulk zoobenthos (43%, 35–50%), and similarly profundal chironomid larvae (Chironomidae) showed higher allochthony (84%, 77–89%) than littoral chironomid larvae (31%, 21–40%, Supplementary Table 6). The allochthony of littoral isopods *Asellus aquaticus* (hereafter *Asellus*) was also high, with a median of 59% (52–66%, Supplementary Table 6). Along the environmental gradient from forested to agricultural catchments, chironomids (littoral and profundal) and profundal bulk zoobenthos showed decreasing allochthony (Fig. 3, Supplementary Fig. 6, Supplementary Table 6). In contrast, the allochthony of littoral *Asellus* and littoral bulk zoobenthos did not change across the environmental gradient (Fig. 3), with model slopes for these consumers overlapping with zero (Supplementary Fig. 6, Supplementary Table 6).

Generally, fish allochthony estimates fell between those of zooplankton and zoobenthos. The highest allochthony in fish was observed for piscivorous and benthivorous fish, followed by generalists, while planktivores showed the lowest allochthony (Fig. 3). This trend is clearly demonstrated between the size classes of European perch (*Perca fluviatilis*), with the allochthony of the large-bodied piscivorous perch (15%, 9–22%) being higher than that of generalist medium-sized perch (7%, 4–11%), which themselves showed higher allochthony than small-bodied planktivorous perch (2%, 0–4%, Supplementary Table 6). Other planktivorous species, vendace (*Coregonus albula*) (2%, 0–7%) and smelt (*Osmerus eperlanus*: 0%, 0–2%), similarly had allochthony close to zero (Supplementary Table 6). Large generalist roach (*Rutilus rutilus*) showed higher allochthony (6%, 3–10%) than small roach (3%, 0–6%). Piscivorous pike (*Esox lucius*: 21%, 13–32%) and benthivorous ruffe (*Gymnocephalus cernua*: 27%, 20–34%, Supplementary Table 6) showed the highest allochthony of the studied fishes. Regardless of feeding guild, nearly all fishes showed decreasing allochthony along the environmental gradient from forest lakes with low pH and high DOC towards eutrophic lakes in more agricultural landscapes (Fig. 3, Supplementary Fig. 6, Supplementary Table 6). Smelt was the only fish species showing a slope overlapping with zero, indicating that allochthony of smelt did not change along the environmental gradient, although this species was not found in any of the lakes with highly forested catchments.

In general, we found that the modelled global average for a consumer (Fig. 3a1) showed a clearly declining allochthony trend along the studied environmental gradient (Supplementary Fig. 6. Supplementary Table 6), with a global consumer allochthony being 12% (4–32%). Moreover, for most of the consumer groups (13 out of 19) the 95% credible intervals of allochthony estimates did not overlap the full range of PC1, indicating substantial declines in allochthony along the studied environmental gradient. Six consumer groups (pelagic copepods, *Chaoborus*, *Asellus*, littoral bulk zoobenthos, smelt and small roach) did not show this clear delineation in their allochthony estimates across PC1; however, four of these groups had allochthony estimates spanning near-zero covering the majority of the PC1 gradient (Fig. 3). The Bayesian mixing models using either N:C ratio of POM or Forest % as a covariate in place of our PC1 provided corresponding results (Supplementary Figs. 3–4). The lower the N:C ratio of POM (more terrestrial OM relative to aquatic OM), or the higher the catchment forest coverage, the higher was the estimated consumer allochthony. Comparatively, based on Deviance Information Criterion (DIC) and approximate $R^2$ values, the model that included the PC1 covariate was the best model (Supplementary Table 8). However, with the Leave-One-Out cross-validation criterion the models with PC1 or N:C ratio of POM as continuous covariate predicted consumer allochthony equally well. More broadly, all models that included a continuous environmental covariate performed significantly better than models without (Supplementary Table 8). We found that the modelled allochthony results were sensitive to the omega values used

(the assumed proportion of H incorporated into consumer tissues that originates from water as opposed to from dietary sources; see Methods); higher omega values resulted in lower absolute allochthony estimates, especially in higher trophic level consumers (please see further details in the Caveats section and Supplementary Fig. 5). However, the relative patterns for consumer allochthony remained consistent irrespective of the omega value used (Supplementary Fig. 5) or adjustments to the $\delta^2$H values of terrestrial or aquatic sources (Supplementary Figs. 8–9).

## Discussion

Across the 35 studied boreal lakes, most consumer groups spanning major food web compartments showed substantially higher levels of allochthony (reliance on t-OM) in lakes with highly forested catchments, low pH and high DOC concentrations (i.e. humic or dystrophic lakes). In contrast, in more eutrophic lakes with agricultural catchments and higher within-lake production of OM, the estimated degree of allochthony was more limited. In other words, relative allochthony decreased in lake food webs across the forest–agriculture continuum. The higher consumer allochthony in lakes in forested landscapes clearly reflects the higher input of t-OM relative to autochthonous OM, as indicated by our DOM quality measurements (low SR ratio and N:C ratio of POM; see Methods). Our findings strengthen the tenet that the availability of t-OM when autochthonous OM is restricted, is a prime determinant of consumer allochthony in boreal lakes. We observed unequal responses of zoobenthos and pelagic zooplankton within lake food webs, with the former always exhibiting higher allochthony values. This highlights differences in their resource use, with zoobenthos being clearly more closely linked to t-OM, likely through feeding on detritus with its associated microbial communities, as well as the surrounding biofilm which can incorporate both particulate and dissolved forms of t-OM. The allochthony of higher trophic level consumers (fish) mirrored those of their prey, with pelagic specialists having lower allochthony than their benthivorous counterparts.

Among zooplankton, copepod allochthony was consistently near-zero, whereas cladocerans showed higher allochthony, especially in lakes with forested catchments and associated higher DOC concentrations. Cladocerans are more generalist filter feeders[31–33] than more selectively feeding copepods[34]. Therefore, in more dystrophic lakes, cladocerans are likely feeding more on bacteria and micro-zooplankton while copepods are better able to target potentially more nutritious algal food. These differences between allochthony of cladocerans and copepods were less stark in more eutrophic lakes, where the OM pool is dominated by algal material. Some cladocerans (e.g. *Daphnia magna*) are not able to grow and reproduce effectively if fed solely with bacterial biomass, likely due to a shortage of essential fatty acids, but even a small fraction of high-quality algae in their diet facilitates growth[35,36]. The likely generality of this restriction amongst lake zooplankton is not known, but since phytoplankton production in lakes is anyway rarely <10% of the heterotrophic bacterial production even in the most dystrophic lakes[37,38], zooplankton growth and reproduction are certainly possible in lakes with high t-OM inputs. Contrary to our expectations, our results show that shallow-water littoral zoobenthos have lower allochthony than deep-water profundal zoobenthos, which has also been observed elsewhere[15,39]. In our boreal study lakes, the estimated allochthony of littoral zoobenthos remained relatively stable at 50%, whereas allochthony of profundal zoobenthos decreased steadily from ca. 75% to 50% along the environmental gradient. Thus, energy and matter derived from t-OM is evidently still of importance for littoral and profundal habitats, even in the most eutrophic lakes that typically have high primary production compared to bacterial production[15,16,38]. However, even in dystrophic lakes, where the brown water colour can restrict pelagic phytoplankton production, production by periphyton in shallow littoral areas can remain substantial[40], reducing strict reliance of littoral benthic consumers on

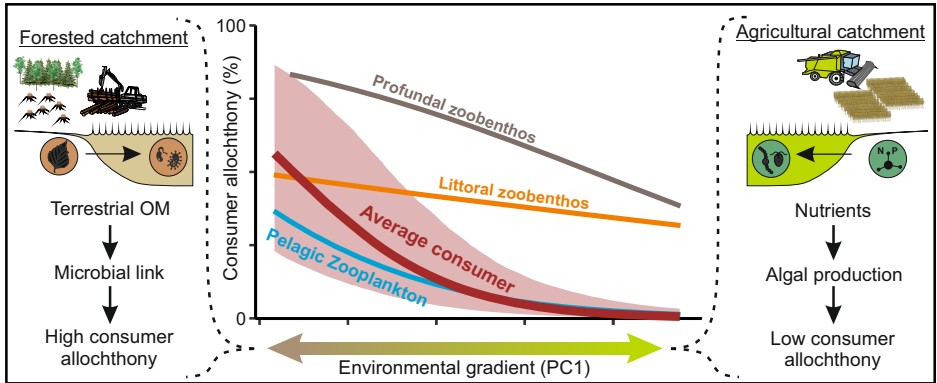

**Fig. 4 | A schematic illustration of how landscape properties drive consumer allochthony in boreal lakes.** The coloured lines in the central plot indicate median estimates of consumer allochthony along the gradient from forested to agricultural catchments, with shaded area indicating 95% credible intervals. Source data are provided as a Source Data file.

t-OM. The sedimentation of t-OM to the lake bottom before assimilation by microbiota makes allochthonous material more bioavailable in benthic habitats compared to pelagic habitats, for which allochthonous material needs to stay either in suspension or dissolved to be accessible. This likely partly explains why we observed higher allochthony in zoobenthos than in pelagic zooplankton, with zoobenthos also potentially utilising different forms of t-OM, such as large particulates that are inaccessible to small-sized filter feeding zooplankton.

The relative allochthony of the primary consumers was generally reflected in that of the higher trophic level fish consumers divided into feeding groups according to prey remains found in their stomach contents. Planktivorous small perch and smelt showed lower allochthony than bleak or vendace which have more generalist planktivorous feeding behaviours. Interestingly, smelt was also estimated to occupy a higher trophic level as compared to other planktivorous fishes, as has been reported previously[41,42]. This high trophic position of smelt is likely related to opportunistic feeding behaviour, including feeding on its own juveniles, as well as the ability to feed on large predatory *Chaoborus* larvae[43,44], which were almost one trophic level higher than cladoceran and copepod zooplankton. The top predators generally showed higher allochthony than lower trophic level fishes, and the trend in top predator allochthony more closely resembled that of zoobenthos and benthivorous fish than that of pelagic zooplankton and planktivorous fish. Thus, the littoral-oriented top predators (pike and large perch) appear to be more closely associated with the benthic than with the pelagic (planktonic) food web compartment, especially in lakes with forested catchments. These results suggests that any change to the relative importance of t-OM and aquatic OM sources (for example due to changes in catchment land use) and hence to the balance between the lake littoral versus pelagic productivity, can resonate through the food webs up to fish communities in the lakes[28].

The observed trends in consumer allochthony suggest that in dystrophic lakes with highly forested catchments, t-OM supports consumers more than in eutrophic lakes. This result is consistent with previous findings where greater catchment area forestry has been suggested to raise consumer allochthony, and that higher in-lake primary productivity is associated with lower consumer allochthony[9,13,23,24]. Based on the estimates of OM quality and origin, the OM composition in dystrophic lakes and eutrophic lakes is evidently different, with the former being more dominated by terrestrial material and the latter more by material of algal origin. This is consistent with previous findings[24,38] and the probable reason why zooplankton, zoobenthos and fish consumers are supported by terrestrial ecosystems more in dystrophic lakes than in eutrophic lakes (Fig. 4). In

summary, the availability of t-OM, either directly or more probably via a microbial link, is likely the key driver of consumer allochthony for both benthic and pelagic food web compartments in lakes. However, estimating the relative importance of t-DOM and t-POM pathways fuelling the aquatic food webs is beyond our study as these sources had too similar $\delta^2H$ signatures (Supplementary Table 10). Moreover, although adding a bacterial step to the models could make the models more realistic (Supplementary Fig. S8), we lack any realistic knowledge of how bacteria can modify the $\delta^2H$ values of t-DOM or t-POM during their metabolism, making any omega value used for this scenario too speculative. We did test applying a relatively high (17%) environmental water contribution to model bacterial $\delta^2H$ values and used those values as the terrestrial source for the models. Even with this likely extreme scenario, estimated consumer allochthony and the allochthony trends remained consistent with the model using inlet DOM as the terrestrial source (please see the Caveats section for further details). Therefore, we are confident that our main findings are robust. To resolve the proportional shares of t-DOM and t-POM pathways and to obtain reliable bacterial omega values for use in models, laboratory experiments coupled with mesocosm or ecosystem-level tracer studies will be needed.

Our findings that catchment characteristics drive allochthonous resource use of lake consumers, also has implications for the production and biomass of benthic and pelagic taxa[45]. A first order approximation of standing biomass supported by allochthony in lake food webs can be obtained by multiplying the average consumer's allochthony by the standing biomass. In the studied environmental gradient, the highest standing biomasses would most likely be found from the most eutrophic lakes[46] (high PC1 scores) and the lowest from the most dystrophic lakes with high forest coverage in the catchment (low PC1 scores) due to differences in nutrient availability and light limitation[47]. Therefore, the lake standing biomass likely increases, while the average consumer allochthony simultaneously decreases along the studied environmental gradient. We suggest that these opposing trends in consumer allochthony and in the assumed lake standing biomass would act to flatten the trend in biomass supported by allochthony across the gradient from forested to agricultural landscapes. Thus, consumer allochthony and lake standing allochthonous biomass could potentially be uncorrelated when considered at the whole lake level, even if species-specific trends occur. Such hypothesis remains to be tested in future studies.

Effective conservation and management of lakes, their vulnerable biota, and associated ecosystem services require a deeper understanding and recognition of the key abiotic and biotic factors that influence ecological communities and processes across connected ecosystems[2,3]. To combat climate change, and increase biodiversity

and resilience of ecosystems, the European Commission has guidelines for implementation of the New Forest Strategy (COM/2021/572) to support the general outlines of the EU Biodiversity Strategy 2030 that aim to enlarge overall forest cover. Our results suggest that such afforestation can change boreal lakes towards systems that are relatively more supported by t-OM. This change can cascade into structural changes of lake ecosystems[28,46,48]. However, quantifying the impacts of forest management on t-OM transport to lakes and consumer allochthony remains uncertain, in part due to the lack of pristine lakes. Forestry actions, including ditching and clear cuts, increase t-OM load to lakes[49] and likely contribute to the observed high consumer allochthony in our Finnish boreal study lakes with highly forested catchments[49,50]. Prior to the Anthropocene, boreal landscapes would have been mainly forested and therefore the lake ecosystems are likely to have received relatively higher support from t-OM compared to the current situation with more widespread agricultural activities in catchments. Our study findings highlight that the development of ongoing policy and management actions that support either the afforestation or deforestation of landscapes need to consider the downstream consequences for freshwater ecosystems and their communities.

Our study highlights how cross-ecosystem transport of organic matter from donor terrestrial landscapes to recipient lake ecosystems shapes the resource use of consumers across all trophic levels, especially through the littoral food web compartment linking t-OM to zoobenthos and finally to fish. The importance of t-OM support is greatest in dystrophic lakes with forested catchments and declines along a gradient towards eutrophic lakes in more agricultural landscapes. This gradient appears to be related to accompanying changes in the source and nature of the organic matter pool (aquatic vs. terrestrial) available to the lake microbial communities, which are the primary terrestrial energy mobilisers in lake food webs[11]. Our results indicate that forested catchments and forestry promote the availability of terrestrial organic matter in lakes, resulting in higher consumer allochthony (likely through a microbial link) and consequently altering the lake food web dynamics. Therefore, the potential impacts of land management policies on freshwater ecosystems deserve consideration when implementing local and regional management measures in the landscape that aim to support biodiversity and vital ecosystem services in highly threatened freshwaters[3].

## Methods

### Permissions for sample collection and ethical matters

Finnish Everyone's Rights, fishing permits from local fishery boards and Finnish Centre for Economic Development, Transport and the Environment (e.g. POSELY: 1271/5712/2012, please see full list of permits from the reporting summary) were applied in sample collection, whereas national guidelines (FI-564/2013 & FI-487/2013) and the European Union directive (2010/63/EU) on the protection of animals used for scientific purposes were applied in animal handling.

### Study lakes and areas

Allochthony of up to 19 consumer groups was estimated from 35 boreal lakes located in southern and central parts of Finland (Fig. 1, Supplementary Table 1), using samples collected once in summer (July–August) during 2016–2018. To take account of temporal and spatial variance in $\delta^2H$ values of lake water, benthic algae, and inlet DOM, we collected additional data from four lakes sampled at multiple seasons and within-lake locations (Supplementary Table 3). The 35 lakes covered marked gradients in morphometry (lake area: 0.01–107.5 km², mean depth: 1–23 m, shoreline development: 1.19–4.89), catchment area (0.2–1936 km²) and characteristics (consisting of: 0–30% urban areas, 0–33% agricultural areas, 49–98% forested areas, 0–8% wetland areas and 2–29% waterbodies), and lake water chemical characteristics (total phosphorus [Tot-P]:

3.9–120.4 µg L⁻¹, pH: 5.5–8.4, DOC: 4.6–24.3 mg L⁻¹) (Supplementary Table 1). We analysed lake water DOC concentration from filtered (GF/F) epilimnion samples (0–1 m depth from the centre of the lake) with a total organic carbon analyser (TOC-L, Shimadzu Co.). The other water chemistry (Tot-P, chlorophyll-a [chl-a]; pH) and morphometric (lake area, mean and maximum depth) data for each lake were derived from the HERTTA open database maintained by the Finnish Environment Institute (www.syke.fi/avointieto). We used geographical information system software (QGIS 3.16.13) and open land use raster map (CORINE 2018 land cover, level 4: pixel size: 20 × 20 m, www.syke.fi/avointieto) to derive different land use categories for each study lake (Supplementary Table 1).

Specific ultraviolet absorbance (SUVA, at 254 nm) and slope range ratio (SR ratio) of absorbance ranges (275–295 nm and 350–400 nm) were determined spectrophotometrically (Shimadzu UV-1800 spectrophotometer, Shimadzu Co.) from filtered water samples (GF/F 25 mm glass fibre filter, pore size: 0.7 µm) and used as proxies for the quality of lake dissolved organic matter (DOM). SUVA index was determined by dividing the absorption coefficient of chromophoric dissolved organic matter (CDOM) measured at 254 nm by the DOC concentration[51]. The SR ratio was determined from log-transformed absorption spectra, by taking the ratio of linear trend slopes from two distinct spectral absorption regions (275–295 nm and 350–400 nm)[52]. SUVA correlates positively with DOM aromaticity[51], whereas SR is negatively related to the molecular weight of DOM[52]. Thus, these parameters can be used as proxies of DOM reactivity/origin: fresh terrestrial DOM should be heavier and more aromatised compared to autochthonous DOM[30]. In addition, the nitrogen to carbon ratio (by mass) of lake particulate organic matter (POM; sampling described below) was determined with an elemental analyser (FlashEA 1112, Thermo Fisher Scientific Inc.). The N:C ratio of POM can be considered as a tracer of the origin of POM, where low values indicate more terrestrial material and higher values more autochthonous material in POM[29,53].

Lake abiotic variables related to lake morphometrics (mean and maximum depth, shoreline development, lake area [LA], catchment area [CA] and CA:LA ratio), chemistry (Tot-P, DOC, chl-a, pH), land use (proportion of urban, agricultural, forest, wetland and waterbody areas) and OM quality and origin (SUVA, SR ratio and N:C ratio of POM) were combined with a Principal Component Analysis (PCA). We used correlation matrix in the PCA as the scales of the variables were different. The first principal component (PC) explained 27.3%, the second 18.0% and the third 12.7% of the variation among the environmental data (Supplementary Fig. 2). Based on the variable loadings on the PCs (Supplementary Fig. 2, Supplementary Table 2), the first PC formed a meaningful environmental variable broadly representing a gradient from acidic high DOC lakes with forested catchments towards more eutrophic lakes with high agricultural activities in the catchment. PC1 was used in subsequent statistical analyses (MixSIAR) as it has a clear ecological meaning and multiple parameters change simultaneously along the studied environmental gradient. Although PC2 and PC3 explained a relatively high amount of the total variance in the environmental data, they did not relate to any meaningful ecological or environmental gradient and thus they were not used as covariates in the MixSIAR isotopic mixing model estimating consumer allochthony.

### Field sampling

Water samples from the lake epilimnion (0–1 m depth) were collected for hydrogen stable isotope analysis (hereafter SIA) from a central location in each lake. 12 mL water samples were subsequently stored at +4 °C prior to SIA. Benthic algae were sampled by scraping visually green patches of algae from hard substrate surfaces from many locations (e.g. sublittoral rocks from <1 m depth) and pooled on site to gain enough sample material for SIA. A compound sample of 5000 mL of lake water from three central locations and samples of water from lake

inlet streams were filtered to separate POM and DOM from the water samples using Millipore tangential flow filtration (Pellicon P2GVPP05 cassette, pore size 0.22 μm) to sample terrestrial DOM flowing into lakes for SIA. 500 ml of the filtrate was collected, freeze-dried and remaining DOM powder was prepared for SIA.

Zooplankton samples were collected with a zooplankton net (mesh size 100 μm, diameter 250 mm) from the three central and littoral areas around each lake. Half of the sample was used as a bulk zooplankton sample and from the other half, cladocerans (Cladocera, predominantly herbivorous *Daphnia* sp. and *Bosmina* sp.) and copepods (Copepoda) were separated with a glass plate technique and the purity of the samples was confirmed under a microscope[23]. Zooplankton samples were kept for several hours in a glass jar to allow for gut evacuation before picking and freezing the sample. Zooplankton samples were frozen likely before they had all fully emptied their stomachs. However, the contribution of gut mass to e.g. cladoceran total mass is low (ca 5%)[54] and thus any potential confounding impacts of gut contents can be considered negligible. Phantom midge larvae (*Chaoborus flavicans*) were sampled with a zooplankton net and an Ekman grab (area: 272 cm$^2$) from the central areas of each lake. Bulk littoral zoobenthos were sampled with an Ekman grab and a kick net (500 μm mesh). Isopods (*Asellus aquaticus*) and chironomid larvae (Chironomidae) were separated from the sample and the remaining fraction was used as a bulk littoral zoobenthos sample. Profundal zoobenthos were sampled from 5 to 10 m depth and from the deepest point of each lake. From these samples, chironomids were separated, and the remaining fractions were used as bulk profundal zoobenthos sample.

Fish were sampled with Nordic multi-mesh gillnets (length 30 m, height 1.5 m, mesh sizes 5, 6.25, 8, 10, 12.5, 15.5, 19.5, 24, 29, 35, 43 and 55 mm) with overnight (ca. 12 h) soaking time[55]. After removing fish from gill nets, they were euthanised with sharp blow to head, frozen and transported to the laboratory, where they were thawed, and total length (mm) and wet mass (g) were measured. Fish stomachs were dissected from oesophagus to pylorus, or in the case of cyprinids from the first third (anterior) of the intestine[56], and the food items were identified to the lowest feasible taxonomic level using a stereo microscope. The relative proportions of identified prey items in the stomachs or intestines were estimated using a modified points method or a volumetric estimation method, respectively[57,58]. The prey taxa were further grouped into five categories (fish, zooplankton, zoobenthos, macrophyte parts/algae, detritus) and the fish were categorised into trophic guilds (benthivore, generalist, planktivore, piscivore) based on the stomach content data (Supplementary Table 7) and existing knowledge of the species' ecology: ruffe (*Gymnocephalus cernua*) were categorised as benthivore; small perch (total length <70 mm), vendace (*Coregonus albula*), smelt (*Osmerus eperlanus*), and bleak (*Alburnus alburnus*) as planktivore; small roach (<100 mm; *Rutilus rutilus*), large roach (≥100 mm) and medium-sized perch (70–150 mm) as generalists; and large perch (>150 mm) and pike (*Esox lucius*) were categorised as piscivore. A small piece of dorsal muscle was dissected from each fish, placed in a plastic vial, and stored frozen at −20 °C for SIA.

## Laboratory methods

For each lake, 1–7 (mean ± SD lake$^{-1}$: 4.4 ± 1.4, total $n = 1159$, Supplementary Tables 4–5) samples of each fish species and 1–18 (2.2 ± 2.6, total $n = 578$, Supplementary Tables 4–5) invertebrate samples were prepared for final SIA. Inlet DOM samples and benthic algae were analysed for SIA by pooling samples taken from three different locations within each site to obtain representative averages. Lake water samples from the central part of the lakes were kept in airtight exetainers at +4 °C prior to SIA. All other SIA samples were stored at −20 °C prior to freeze-drying to constant weight, homogenisation into fine powder and weighing for final SIA.

For nitrogen stable isotope ($\delta^{15}$N) analyses, 0.600 ± 0.050 mg of sample powder was weighed into tin cups for final SIA conducted at the University of Jyväskylä, Finland, using a continuous flow isotope ratio mass spectrometer (Thermo Finnigan DELTA$^{plus}$Advantage, Thermo Fisher Scientific Inc) coupled to an elemental analyser (FlashEA 1112, Thermo Fisher Scientific Inc). The reference materials were lab specific standards (pike white muscle tissue and birch *Betula* sp. leaves) of known relation to the international standards of atmospheric N$_2$ (for nitrogen). The precision of $\delta^{15}$N analyses were always better than 0.2‰, based on the standard deviation of replicates of the internal standards.

For hydrogen stable isotope ($\delta^2$H) analyses, 0.350 ± 0.005 mg of sample powder was weighed into silver cups that were stored open with laboratory standards for at least five days prior to folding the cups in order to allow interchange of exchangeable hydrogen between samples and laboratory air[59]. Hydrogen stable isotope analyses were done at the University of Jyväskylä, Finland, using an Isoprime 100 isotope ratio spectrometer (Isoprime Ltd) coupled to a vario PYRO cube elemental analyzer (Elementar Analysensysteme GmbH). Multiple samples of two reference materials (caribou hoof [CBS: −157.0 ± 0.9‰] and kudu horn [KHS: −35.3 ± 1.1‰]) expressed relative to V-SMOW (Standard Mean Ocean Water)[60] were analysed in each run. Here, the standard deviations of replicate reference materials within each run were always below ~3‰. The lake water samples were stored in a refrigerator before $\delta^2$H analysis performed at the University of Oulu, Finland, with a CRDS laser instrument, Picarro L2120-i, (Picarro Inc, US). Here, the standard deviations in reference material (V-SMOW) were always below ~1‰. All stable isotope ratios are expressed as ratio to the corresponding international standard using the delta (δ) notation.

## Statistical methods

The trophic level (TL) of consumers was estimated with a single end-member nitrogen isotope mixing models (TL = 2.1 + ($\delta^{15}$N$_{consumer}$ − $\delta^{15}$N$_{baseline}$)/3.4), where herbivorous cladocerans at trophic level of 2.1 (±0.1 SD)[24] were used as the isotopic baseline for zooplankton, profundal zoobenthos and fish, whereas the baseline for littoral zoobenthos was littoral *Asellus* (TL = 2.1 ± 0.1 SD). Trophic fractionation of nitrogen was set to 3.4‰ (±1‰ SD)[61]. In four lakes (Hirvijärvi, Niemisjärvi, Pankajärvi and Valkea-Kotinen), the cladoceran $\delta^{15}$N values were lacking and thus they were derived from a linear model based on $\delta^{15}$N values of cladoceran and bulk zooplankton samples from the other study lakes (adjusted $R^2 = 0.928$, $p < 0.0001$, $\delta^{15}$N$_{cladocera} = 0.872 \times \delta^{15}$N$_{bulk\ zooplankton}$ − 0.095). In one lake (Pesosjärvi), the *Asellus* $\delta^{15}$N values were lacking and thus they were derived from a linear model based on $\delta^{15}$N values of littoral *Asellus* and littoral bulk zoobenthos samples from the other study lakes (adjusted $R^2 = 0.886$, $p < 0.0001$, $\delta^{15}$N$_{asellus\_littoral} = 0.990 \times \delta^{15}$N$_{bulk\ zoobenthos\_littoral}$ − 0.981). In the case of 14 zoobenthos samples (mainly profundal chironomids), the measured $\delta^{15}$N values were substantially lower than source $\delta^{15}$N values. These zoobenthos samples had clear methane carbon isotope signal (mean $\delta^{13}$C = 40.1‰) and as methanogenesis likely influences $\delta^{15}$N values, the observed very low $\delta^{15}$N values do not reflect the relative trophic position of these samples. Thus, the trophic levels of these samples were fixed to 2.1 according to our best ecological knowledge.

Based on the best available literature assessments and the published correction equations associated to the newly assigned $\delta^2$H values of reference material[60,62], approximately 23 ± 9% (Supplementary Table 9) of consumer hydrogen atoms are exchanged with environmental water in each trophic level. Prior to consumer allochthony estimations, the percentage of environmental hydrogen in each consumer sample was estimated with the following equation[63]: $\omega_{compound} = 1 - (1 - \omega)^{TLc - 1}$, where $\omega_{compound}$ is the proportion of water $\delta^2$H in consumer $\delta^2$H values and $\omega$ is the estimated proportion of water $\delta^2$H entering the consumer tissues (0.23). TLc − 1 represents the

difference in trophic levels between the consumer and the basal hydrogen sources. With this information, we corrected the measured consumer $\delta^2$H values to the same trophic level with sources[39,63]:
$\delta^2$H$_{consumer, \omega-corrected}$ = ($\delta^2$H$_c$ − $\omega_{compound}$ × $\delta^2$H$_{water}$) / (1 − $\omega_{compound}$), where $\delta^2$H$_{consumer, \omega-corrected}$ is the omega-corrected consumer $\delta^2$H values, $\delta^2$H$_c$ is the measured consumer $\delta^2$H value, and $\delta^2$H$_{water}$ is the lake-specific water $\delta^2$H value.

It was not possible to obtain sufficient samples of pure phytoplankton for SIA. Therefore, we used photosynthetic fixation-adjusted water $\delta^2$H values as aquatic source values for consumers. These values were obtained from lake water $\delta^2$H values with a photosynthetic discrimination estimate of −110‰ derived from the mean difference between measured benthic algae and pelagic water $\delta^2$H values across the sampling sites, assuming that photosynthetic fractionation of hydrogen is equivalent for phytoplankton and benthic algae[18]. Our photosynthesis hydrogen discrimination value corresponds very well with the recalibrated values (−111 ± 18‰) reported in available reviews[60,62]. The observed spatiotemporal standard deviation in benthic algae (aquatic source) and inlet DOM (terrestrial source) $\delta^2$H values were 23.6‰ and 11.2‰, respectively (Supplementary Table 3), which we applied to all 35 lakes. For two lakes (Arkusjärvi and Horkajärvi) we were unable to determine clear inlet and thus the inlet DOM values for these lakes were obtained from the nearest lake inlets (Suuri-Jukajärvi and Majajärvi, respectively). With lake-specific source values and the spatiotemporal variance, just 34 (5 zooplankton, 2 zoobenthos and 27 fish samples) out of the 1737 consumer samples fell outside the ±2 SD range of the basal resources. Most (>98%) of the sampled consumer $\delta^2$H values fell inside the lake-specific range of $\delta^2$H values of basal resources (Fig. 2). Only 3 samples fell outside ±3 SD range of the basal resources. However, as we had multiple samples from these consumers per lake, we did not exclude the values from the dataset prior to running the isotopic mixing models.

We based our Bayesian mixing models on the MixSIAR package (version 3.1.12) in R (version 4.2.1) to estimate terrestrial basal resource use of consumers (consumer allochthony) in each lake[27,64]. In the MixSIAR model, we set lake and consumer taxon as random factors[63]. The combined environmental gradient (PC1) was selected to be used as a covariate[65] in MixSIAR model instead of single environmental factors such as catchment Forest % or lake N:C ratio of POM, since multiple environmental parameters change simultaneously along the studied landscape gradient. We customised the MixSIAR produced JAGS script to further include a random slope with PC1 for each consumer taxon, giving the total model structure as: Allochthony ~1 + PC1 + (1 + PC1 | species) + (1 | lake). In this structure, 1 refers to an intercept value when the covariate (PC1) is equal to zero, and allochthony is taken to be the proportion of terrestrial contributions to consumer populations.

For each lake, we used two lake-specific sources, inlet t-DOM (terrestrial) and the photosynthetic fixation-adjusted water $\delta^2$H values (aquatic), with the sampled spatiotemporal standard deviations. We set the trophic fractionation factor (TDF) for $\delta^2$H to zero[66]. To account for variability in trophic levels of isotopic baselines (*Asellus* TL = 2.1 ± 0.1 & Cladocera TL = 2.1 ± 0.1)[24] and TDF of $\delta^{15}$N (3.4 ± 1.0‰)[61], we added additional uncertainty to the $\delta^2$H$_{consumer, \omega-corrected}$ estimate (± 13‰, based on mean difference of above-mentioned min–max-scenarios), following a recent study[67]. We used true, uninformative flat priors (as we only have two sources) and multiplicative error structure (process × residual errors) in the source mixing models[68]. With these setups, we ran the model with $10^6$ iterations of 3 chains, with a burn-in of $5 × 10^5$ and a thinning factor of 500. Model convergence was confirmed with Gelman and Gweke diagnostics and visual inspection of the trace plots. In addition, mean of the multiplicative error term was below 1 (0.47 ± 0.02) and thus confirmed the underlying assumptions in our mixing model[27].

The lake- and species-specific source contributions (posterior median and 95% Bayesian credible intervals [CI]) were combined with

the continuous effect on consumer allochthony to visualise the model results. Additionally, intercepts of allochthony estimates (i.e. when PC1 = 0) were used to describe relative differences between consumer groups (Supplementary Table 6). The linear coefficient with PC1 in isometric log-ratio (ILR) space (the non-euclidean space within which mixing model parameters are determined) was clearly non-zero for the average consumer and for most of the consumer taxa (e.g. global slope median [CI] = 0.84 [0.52–1.20], Supplementary Fig. 6, Supplementary Table 6). As our model included only two sources, ILR space is one dimensional, and thus non-zero covariate slopes directly indicate that PC1 had significant effect on consumer allochthony. We also ran mixing models with catchment Forest % and N:C ratio of POM as continuous variables (in place of PC1, Supplementary Figs. 3–4) to better explore the potential mechanisms behind the observed trends in consumer allochthony. The results from these additional runs and their interpretation were almost identical with those from the model with PC1 as the continuous covariate. The relative support for the models were assessed with comparing Deviance information criterion (DIC), leave-one-out cross-validation information (LOOic), Akaike weight, and Approximate $R^2$ values to the corresponding values from models without continuous covariate (Supplementary Table 8).

## Caveats

We recognise that our isotope modelling and results are based on several assumptions. For the $\delta^2$H analysis, the Comparative Equilibrium method[59] assumes that organisms have similar exchangeable hydrogen properties as those of the standards used. This is likely not exactly the case, and the method used will probably have produced some precision and accuracy issues in our results. Dual Vapour Equilibration is a more precise method to determine exchangeable hydrogen, but this procedure was still under development at the time of our analyses. The Comparative Equilibrium method is still widely applied and, due to the large isotopic separation of the initial sources (aquatic versus terrestrial), we are confident that any uncertainties will not have caused any major issues with our model results.

Most of the samples in this study were collected in late summer, restricting the temporal generality of our model outputs. For example, $\delta^{15}$N values of pelagic POM and zooplankton, and of littoral *Asellus* vary seasonally[69,70] which could influence the allocated consumer trophic levels and thus the corrected $\delta^2$H values and final model outputs. Moreover, the turnover time of fish muscle tissue varies from weeks to months, thus reflecting assimilated diet over the preceding several weeks, which is likely also the case for relatively large and slow-growing benthic organisms. In contrast, small fast-growing zooplankton will yield isotope values reflecting more recent diet. We applied relatively high variances to source $\delta^2$H values that were derived from spatiotemporal data from lakes sampled in multiple seasons and/or locations (Supplementary Table 3). Therefore, seasonal fluctuations in source isotope values should be accounted for to some extent in our models. However, the most profound effects of seasonality on consumer isotope values would be found from the fast-growing zooplankton taxa. As our consumer data were collected during the summer growing season when autochthonous algal production and exploitation is highest, our zooplankton allochthony values are likely underestimating the overall annual allochthony of these pelagic taxa[24,26].

Mixing model outputs of consumer allochthony are known to be sensitive to the selected omega values[62]. We used an omega value of 0.23 based on the best literature assessments after applying a correction equation based on a recent re-evaluation of hydrogen isotope standard material[60,62] (Supplementary Table 9). We ran additional sensitivity analyses for the models by using a reasonable omega range based on the recalibrated previously reported omega values (0.14 and 0.32, Supplementary Fig. 5). With omega values of 0.14, 0.23 and 0.32, the total environmental water contribution in a hypothetical (i)

secondary consumer (TL = 3) and (ii) top predator (TL = 4) are (i) 26%, 41%, 54% and (ii) 36%, 53%, 69%, respectively. Thus, the modelled allochthony values of the consumers, especially those occupying higher trophic levels, are significantly different when using omega value of 0.14 or 0.32, but the allochthony trends observed across the landscape gradient remain the same for most of the consumer groups (Supplementary Fig. 5). Additionally, due to the cumulative effect of omega-corrections onto $\delta^2H$ values over multiple trophic levels, the allochthony estimates of higher trophic level organisms are more uncertain than estimates for lower trophic level organisms. However, we did not observe positive trends between the PC1 and consumer trophic levels (Supplementary Fig. 7), demonstrating that our trophic level estimations are not driving the observed decreasing consumer allochthony patterns across the environmental gradient.

Applying a bacterial trophic step to consumer allochthony models could make the consumer allochthony models more realistic as a 'microbial link' is a key pathway between terrestrial organic matter and aquatic food webs. To our knowledge, there are no data regarding the environmental water contribution to bacterial $\delta^2H$ values in natural environments (only an amino acid-specific environmental water contribution from laboratory experiments with *E. coli*[71]), so attempting to incorporate this additional step at present would make the models unnecessarily speculative. Nevertheless, we did run a model scenario in which we modelled bacterial $\delta^2H$ values ($\delta^2H_{bact} = \delta^2H_{inlet\ DOM} \times (1 - \omega) + \delta^2H_{lake\ water} \times \omega$) with a relatively high omega value (0.17) and used these values as the terrestrial source for consumers (Supplementary Fig. S8). This bacterial step scenario produced slightly lower allochthony estimates than those from the main model, but most importantly the trends between consumer allochthony and the environmental gradient remained the same.

As the collected benthic algal samples might have contained different amounts of impurities among lakes, we used the average difference between the $\delta^2H$ values of benthic algae and lake water to obtain the photosynthetic $\delta^2H$ fractionation value. Because the photosynthetic $\delta^2H$ fractionation might differ temporally and among lakes, we also ran the Bayesian consumer allochthony models with a scenario in which we used the individual sampled benthic algae values as a lake-specific aquatic hydrogen source (Supplementary Fig. S9). This test provided slightly more variance to the model output parameters but the overall consumer allochthony trends along the environmental gradient again remained essentially the same. In the main model (Fig. 3) we implemented a relatively high standard deviation to the lake-specific source values derived from spatiotemporal sampling with concurrent low variance in lake water $\delta^2H$ values. Therefore, we are confident that the applied spatiotemporal variance is taking the possible variance in photosynthetic $\delta^2H$ fractionation into account in the main model. Based on the multiple different omega scenarios and scenarios when applying different source $\delta^2H$ values and their similarity to the model results shown in the main article, we are confident that our overall findings are robust and that consumer allochthony is highest in lakes with predominantly forested catchments and receiving high inputs of t-OM, and declines towards more eutrophic lakes with predominantly agricultural catchments.

We were able to evaluate consumer allochthony in lakes with contrasting land use types in the catchment, but we did not have measurements of land use intensity. The impact of anthropogenic activities in forests is especially difficult to determine as the regeneration cycles of the forests are long. In our study landscapes, industrialisation and mechanised agriculture have increased nutrient (nitrogen and phosphorus) loading from fields to freshwater ecosystems[72] while forest management, such as ditching and clear cutting, have been suggested to be a major factor behind the recorded browning of boreal lakes[49,50,73]. Especially in southern and central Finland, only a minority of the forests is protected, and the majority is under intensive forestry, which includes extensive ditch networks covering most of the forests grown on former peatlands and wetlands[50]. Therefore, the combined environmental gradient (PC1) incorporates the forestry activities (Fig. 4) but disentangling their effect from other variables is difficult. Future studies pinpointing the anthropogenic impacts (especially ditching) on consumer allochthony should include pristine control lakes for quantifying the strength of anthropogenic actions in different parts of the catchments.

### Reporting summary

Further information on research design is available in the Nature Portfolio Reporting Summary linked to this article.

## Data availability

All datasets used in this study are publicly available. Geospatial shape files used for map producing are available through Natural Earth (www.naturalearthdata.com). Water chemistry data are available at the HERTTA open database maintained by the Finnish Environment Institute (www.syke.fi/avointieto). Corine land use level 4 data are available from the Finnish Environment Institute (www.syke.fi/avointieto). A compiled version of these environmental variables is presented in Supplementary Table 1. The stable isotope data are available through a GitHub repository[74] [https://doi.org/10.5281/zenodo.14792582]. Source data are provided with this paper.

## Code availability

R scripts used in consumer allochthony modelling are available through a GitHub repository[74] [https://doi.org/10.5281/zenodo.14792582].

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

## Acknowledgements

We thank the staff of the Natural Research Institute of Finland (LUKE) for help with lake selection and sampling, especially of fish. We thank the students and technical staff at the University of Jyväskylä who assisted with field and laboratory work. Krista Rantamo and Salla Ahonen provided the CDOM spectral absorption data. This research was funded by the Research Council of Finland research grants (285619 to R.I.J. & H.H., 340901 to A.P.E. and 355717 to Dr. Jan Weckström). This research was funded through the 2020-2021 Biodiversa+ and Water JPI joint call for research projects, under the BiodivRestore ERA-NET Cofund (GA N°101003777), with the EU and the funding organisation the Research Council of Finland (project nr 351860).

## Author contributions

O.K. conducted the data analysis and drafted the first version of the manuscript. M.C. participated in data analysis. M.K. and J.S. collected the data and conducted stable isotope analyses. All authors, including A.P.E. and H.H., contributed to interpretation of the results and editing of the text. R.I.J. planned the study and was the principal investigator for the core grant funding. Authors between O.K. and R.I.J. are ordered alphabetically in the author list.

## Competing interests

The authors declare no competing interests.
