## [Transparent Peer Review file · Nature Communications]

The role of land use in terrestrial support of boreal lake food webs

Corresponding Author: Dr Ossi Keva

Version 0:

Reviewer comments:

Reviewer #1

(Remarks to the Author)

This study tests how cross-ecosystem subsidies support aquatic food webs. This is an important area of research in ecology and conservation, though it strikes me that the issue of how allochthony influences productivity rather than resource use is a timelier question. The authors also claim that this research question is a controversial topic (lines 41-43), but I disagree, and I do not believe it is helpful to our science to continue to portray this question as controversial. Ref 25 resolved this “debate” about the importance of allochthony once and for all by demonstrating there is massive variation in the degree of consumer allochthony in a larger global (though northern temperate focused) dataset, and identified the factors both within lakes and catchments that influence this variation. The novelty of this current study is communicated much better on line 45, where I agree that past focus has been on zooplankton and small pelagic, though also littoral fish.

The main result of this study is that allochthony varies along a land use gradient from more forest to agricultural-dominated catchments. I think that this is an important result that is largely novel, though does overlap somewhat with past literature, e.g. ref 25. The main distinction with this study is the thoroughness with which this main result is demonstrated across habitat zones (pelagic and littoral) and across trophic levels (in 19 taxa). However, I do feel that the study largely lacks a hypothesis-driven, mechanistic focus for the results unlike ref 25, and this opinion resonates strongly in the Discussion. I have highlighted specific lines where this point could be addressed in the Discussion in my Specific Comments.

The work appears largely technically sound, meeting expected standards in the field. Nonetheless, it will be important to answer how well the Bayesian mixing models fit the data, such as on line 131. Some of this information is provided from the credible intervals but something like a R2 for the MixSIAR models must be added. More philosophically, I'm not a fan of indirect ordination-based approaches to define covariates. The principal components analysis is particularly messy because forest and agricultural cover are associated with different axes, and DOC seems orthogonal to both. Many of the environmental vectors appear weakly aligned with PC1, and PC1 itself anyway only explains a quarter of the variation in the environmental datasets. I would have preferred testing the importance of forests directly by including it as a covariate along with other variables. How robust are the results to this, i.e. what happens if you replace PC1 with %forest or %agriculture in the catchment?

Specific comments:

Line 51: This was a synthesis of 140+ sites not a meta-analysis in the strict definition, which would be aggregating existing statistical estimates. This study generated all the statistical analyses fresh using one approach.

Line 63-65: I think the novelty could be better stated here given ref 25. I think this has to do with the 19 consumer taxa spanning four trophic levels within benthic and pelagic habitat (line 67).

Line 88: The reason for this decline to be expected should be explained. How might forests differ from agricultural land in transport of t-OM and nutrients? Also, aren't the transport routes and their proximity to the water key here? I can imagine that this isn't a problem in the strictly forested or agriculture landscapes, but in the mixed landscapes. In a mixed landscape, if forests are more hydrologically connected than agricultural lands, then they can dominate the signal or pattern you're hoping to detect.

Line 95: What is a 0.2 loading? Is that a Pearson correlation coefficient?

Line 115: The loadings should be reported somewhere to support this conclusion.

Line 117: SR ratio should be explained before it introduced in this I-R-D-M article format.

Line 133: Figure 3 is outstanding, but again, it's really unfortunate that we have PC1 on the x-axis because it is so non-intuitive, e.g. what does -2.5% mean vs 2.5 for the % of forest in the landscape?

Line 142: So these pie charts are averaged across all individuals collected across PC1?

Lines 179 and 224: What is it specifically about the composition of OM that influences allochthony? There could be much more focus on mechanisms and explanations here for why different materials result in different levels of allochthony. What is it about these different organic matter sources? Likewise, why is t-OM more important for zoobenthos on line 196?

Line 219: See also ref 25.

Lines 224-229: I think this text belongs in the Methods and could be expanded to explain why a POM N:C gradient reflects catchment characteristics and especially PC1.

Line 235: I don't think this conclusion is supported by the data. Surely, hydrology and delivery of that OM is what matters. Afforestation at the margins of a large catchment may have a negligible effect. I wonder how much of the results would change if these effects were considered, e.g. distance to lakes. A good way to measure this would be the R² value or AIC of the MixSIAR models with different covariates.

Line 401: PC1 explained "more of the total variance" but in all fairness, this was still small (27.3%).

Line 481: Why use trophic level 2 as the isotopic baseline? This estimation doesn't include any uncertainty as far as I can tell, i.e. the TL estimates are fixed. Omega (line 489) seems to suffer from a similar problem, though the sensitivity analysis in Expanded Data Fig 4. alleviates this concern. Should the same thing be done for TL?

Line 498: Given how essential this number is to the entire analysis, there should be much more detail on how this number was derived, how it was modelled, how good the model fit the data, and why this difference between measured benthic algae and pelagic water is an appropriate assumption for phytoplankton 2H, including how this assumption has been validated.

Line 504: What does "individually" mean here? You fitted MixSIAR twice?

Line 505: How appropriate is it to extrapolate error from 4 lakes to the entire dataset? Presumably variability in DOM depends on landscape characteristics and lake morphometry, so were these four lakes representative?

Reviewer #2

(Remarks to the Author)
see attached

Reviewer #3

(Remarks to the Author)

I have followed the literature on terrestrial contributions to aquatic consumers (allochthony) closely the last 15 years. The research topic has partly stagnated over time, but the ms by Keve et al. (NCOMMS-23-44093-T) takes a great leap forward by its unprecedented analysis of multi-taxa allochthony at the landscape level. Although the study is limited to a snapshot picture of 35 lakes, the inclusion of a large number of taxa and organism groups makes it one of the largest existing SIA-based assessment of allochthony. The ms presents the first systematic evidence for strong broad-scale patterns in allochthony across gradients in boreal forested and agricultural landscapes, hence it has high news value and should be considered for publication in a high-impact journal. However, I have a few relatively minor comments, mainly about the overall statistical approach and the methods choices.

General comments

1. Patterns in allochthony showed strong links to the first axis of a PCA run on a range of environmental variables. However, I see a risk of basing the landscape analysis only on links to this single ordination variable (PC1). The variable explained only 27% of the environmental data, out of which the catchment composition variables % Forest, % Agriculture and % Urban contributed partly. If being cynical, one can argue that a PC variable can be made to behave the way we want, by playing around with what variables to include and not include in the PCA. There are more sophisticated methods to disentangle the influence from different landscape and environmental variables on a dependent variable. I can see the elegance of the simplicity in the authors' analysis, and I am not proposing that a change is critical, but a PCA has limitations. Since the paper makes a big deal about the importance of forest, the authors could at least show that there are direct links between % Forest as a single variable and allochthony, in my humble opinion. Such an analysis could be included in supplementary information.

2. The method used to assess allochthony from stably hydrogen isotopes and a Bayesian model is state-of-the-art and based on standard assumptions. Nonetheless, uncertainties of such a model are inherently very large given the variability in algal vs terrestrial isotope signatures and contributions to consumer hydrogen from dietary water. The ms does a decent job at addressing these uncertainties in the "Caveats" section, and there is a sensitivity analysis for the possibly most uncertain parameter "omega" (per-trophic-level H contribution from dietary water). Authors make a good point about the overall patterns in allochthony being qualitatively the same regardless of model assumptions, even if the absolute allochthony values vary widely with the assumptions. However, there is one view on uncertainty that I find lacking: the way the model works, propagation of uncertainties is disproportionately increasing (in an exponential manner) with trophic level. This means that the allochthony results for herbivorous zooplankton are much better confined than those of, e.g., TL-4 fish or zoobenthos. At some trophic level it can be discussed if it is meaningful at all to model allochthony from hydrogen. This is something that needs to be addressed.

3. Whereas the Bayesian allochthony modelling is well explained, I have a hard time following how the trend analyses were

carried out. There is literally no statistics provided for the trend analyses in Fig. 3 and elsewhere, and little explanation provided in the methods section. I suspect that this is based on some kind of curve fitting within a Bayesian framework, but also with Bayesian statistics it can be shown how probable the hypothesis of a trend is. In, for example, the extended fig 4 it would really help to know which curves are “significant” or not, because some of those trends at alternative omega scenarios look very weak.

4. Something to remember about allochthony estimates is that they are relative to the biomass. This means that there, hypothetically, can be high “allochthonous biomass” and high rates of “allochthonous production” in a lake with low allochthony, or vice versa; a low percentage of a high biomass could represent high absolute incorporation of allochthonous carbon in the food web. It would be interesting to know if the authors have biomass data, such that it can be discussed whether or not the incorporation of terrestrially-derived organic matter was higher in the lakes with high allochthony than in lakes with low allochthony. It could also be that the lakes with most allochthony were very unproductive and that the incorporation of terrestrially-derived organic matter in those lakes was small in quantitative terms.

Specific remarks

1. Line 13 and elsewhere. Not sure the narrative is needed about the topic being controversial. Indeed, there was a hot debate about SIA-based allochthony 15 years with a couple of loud voices, but today allochthony is not controversial. Rather emphasize the knowledge gaps.

2. Line 16. Vague phrasing “variation related”. Should be explained how the trend looked like more specifically - not just that there was a relation.

3. Line 25. Urbanization is a process, not a land use

4. Line 29. Repeated word Considering ... considered

5. Line 43. See specific remark #1

6. Fig. 1 caption. The figure would be more readable if the lakes are numbered from 1 and onwards, instead of using the three-letter code which make the figure look messy.

7. Fig. 2 Consider adding all the isotope raw data (including 2H in water) as supplementary information. This increases transparency and makes it easier for others to repeat the analyses, or use the data as part of future meta-analyses of many data sets.

8. Paragraph beginning on line 199. Here or possibly in the preceding paragraph it could be discussed that allochthony estimates at higher trophic levels are relatively much more uncertain. See general comment #2.

9. Lines 250-256. I find the management implication a bit unclear. This is also not conclusion, but rather “outlook” so it appears misplaced under the conclusion heading.

10. Lines 418-419. Were the zooplankton gut-evacuated? If not, gut contents of terrestrial particles may have biased the results.

11. Lines 465-466. Arbitrary effects from exchangeable hydrogen is a classical problem in deuterium measurements. Can the authors comment on the advantage and disadvantage of the selected method (storing the samples open for 5-d equilibration in lab air), compared to steam equilibration?

12. Line 498. This assumption is probably fine, but the supplementary data shows that this discrimination factor varied a lot in the measurements. The uncertainty related to this should be commented on.

13. Extended Fig. 1. There is lack of statistical detail here. On what alpha level were the correlations significant? Was a correction for multiple comparisons carried out?

14. All figures with trend lines: It is difficult to follow exactly how the curve fitting was done and how the confidence intervals were extracted, and whether or not the relationships represent statistical significance.

Version 1:

Reviewer comments:

Reviewer #1

(Remarks to the Author)

Comments in response to Response

I thank the authors for replying to my previous review. They have mostly addressed my concerns, although a few points persist from their Response to Referees.

General comments:

The authors are incorrect about p-values and R2 values. These can entirely be estimated within a Bayesian framework. Many standard packages exist to do this in R (e.g. see `bayes_R2.brmsfit` in the `brms` package or `bayes_R2` package) and I think the authors must know about this if they are using LOOic (which comes from the same set of packages). R2 can even be calculated by hand from the MixSIAR output as per the equations in Gelman et al., 2019 American Statistician <https://doi.org/10.1080/00031305.2018.1549100>. Claiming that the CIs are analogous to R2 values is not really quantitatively comparable.

More problematically, the authors state the precise problem in their own words: "We want to highlight that currently it is possible to include only one covariate at the time in MixSIAR models" Which makes one wonder, is MixSIAR an appropriate tool? I think it is pretty obvious that the answer is no, MixSIAR is inappropriate. Although isotopic mixing models are the state-of-the-art and most appropriate way to address the specific research questions posed in this study, the authors should really be writing their own models. That said, I think Table R2 and Supplementary Fig. 4 help to address my concerns about the use of multiple coefficients, so perhaps the authors don't need to do anything more here beyond finding a way to report the R2 values.

Specific comments

Line 401: I'm sorry but perhaps there has been a misunderstanding. My entire point is that PC1 *does not* have a "clear ecological meaning". It is a non-intuitive composite variable. This revision here also doesn't address the fact that PC1 is kind of a poor predictor (explains only 27.3% of the total variance in the "multiple" parameters). Nonetheless, the new Table R2 helps address this concern and could be included in the Supplementary Information.

I also commend the authors in their measured and thorough reply to Reviewer #2, who has misportrayed both models like the MixSIAR approach used by the authors as well as the wider ecology (i.e. comment about consumption of t-POM rather than t-DOM). I also think the Reviewer's insinuation from their comment #6 is inappropriate and commend the authors in handling it so well.

** Comments on revised paper **

The revised paper is improved and I only have some minor points remaining, some of which have emerged from revision:

Line 29: Do you mean "organic" rather than "mineral"? Or "organic and mineral"?

Line 88: Reduce "the number of environmental dimensions" for what purpose? Add a few more words to explain the rationale.

Line 91: What are these numbers in parentheses? How does one interpret them? Is -0.31 large or small? I would consider if these numbers are really necessary to report. I think they might be Pearson correlation coefficients, in which case, interpretation would be straightforward if this point was simply reported.

Line 139: Why are these "not as conclusive"? Please add some words to explain why this is the case, especially in light of line 143.

Line 144: This is the first time omega has been mentioned so explain what it is.

Line 145: Can you summarise how these relationships were affected?

Reviewer #2

(Remarks to the Author)

Reviewer #3

(Remarks to the Author)

I was "Reviewer #3" in the assessment of Keva et al. original manuscript (NCOMMS-23-44093-T), which has now been revised. In general, the rebuttal letter does an excellent job at replying to all my concerns.

However, whereas the replies as such are OK, in some cases there is no explanation to how the manuscript was changed (or why it was not changed). Moreover, the authors provided no line number references or other means to make it easy to follow up on the changes. This being mentioned, I have carefully read the manuscript and found all changes that could be identified acceptable. It took some (unnecessary) time to identify the changes because the authors left no line number references or tracked changes to guide where to look. In the following cases I was not able to identify how the manuscript has been changed:

General comment 4 from the last review round: Even if the authors have no biomass data, it is still possible to discuss whether or not the allochthony reflects patterns in absolute masses or fluxes. This is critical when interpreting what the results mean for the ecosystems and how they function.

Specific remark 10 from the last review round: I don't see how the manuscript has been changed to clarify gut evacuation protocol and its implications.

Specific point 11 from last review: I don't see how the manuscript has been changed to justify the approach for dealing with exchangeable hydrogen and discuss its weaknesses relative to e.g. steam equilibration. The answer provided in the rebuttal is fine, but what does that matter do future readers of the paper, I mean if there was no change made in the actual manuscript?

Version 2:

Reviewer comments:

Reviewer #1

(Remarks to the Author)

I thank the authors for going beyond what was necessary (e.g. adding new analyses) to address my remaining concerns. I have no further comments. Congratulations on the excellent paper!

(Remarks on code availability)

I have skimmed the repo and it does seem like the code is a usable community resource (note I have not tried running it). The README file has suitable instructions but it could include more information on how to install and run the code for beginners.

Reviewer #2

(Remarks to the Author)

see attached

(Remarks on code availability)

Reviewer #3

(Remarks to the Author)

The authors have done an excellent job responding to all of my comments. I have no further concerns.

(Remarks on code availability)

Response Letter

Comments from Reviewers (Text in blue with normal font)

Response from authors (*Text in black with italics*)

Reviewer #1 (Remarks to the Author):

This study tests how cross-ecosystem subsidies support aquatic food webs. This is an important area of research in ecology and conservation, though it strikes me that the issue of how allochthony influences productivity rather than resource use is a timelier question. The authors also claim that this research question is a controversial topic (lines 41-43), but I disagree, and I do not believe it is helpful to our science to continue to portray this question as controversial. Ref 25 resolved this “debate” about the importance of allochthony once and for all by demonstrating there is massive variation in the degree of consumer allochthony in a larger global (though northern temperate focused) dataset, and identified the factors both within lakes and catchments that influence this variation. The novelty of this current study is communicated much better on line 45, where I agree that past focus has been on zooplankton and small pelagic, though also littoral fish.

RESPONSE: *We thank Reviewer 1 for the positive and constructive comments on our manuscript. We have now changed the word “controversial” to “underexplored”. Previous studies have focused on zooplankton and pelagic vertebrate consumers, whereas our study covers both benthic and pelagic habitats and food-web compartments. Moreover, as there are several uncertainties in allochthony modelling (e.g. omega value and trophic level estimation), the absolute degree of allochthony in consumers still remains largely underexplored. We have now added the suggested reference to our hypothesis section.*

The main result of this study is that allochthony varies along a land use gradient from more forest to agricultural-dominated catchments. I think that this is an important result that is largely novel, though does overlap somewhat with past literature, e.g. ref 25. The main distinction with this study is the thoroughness with which this main result is demonstrated across habitat zones (pelagic and littoral) and across trophic levels (in 19 taxa). However, I do feel that the study largely lacks a hypothesis-driven, mechanistic focus for the results unlike ref 25, and this opinion resonates strongly in the Discussion. I have highlighted specific lines where this point could be addressed in the Discussion in my Specific Comments.

RESPONSE: *Our study provides valuable insights into how catchment area characteristics and lake chemistry modify the degree of consumer allochthony, with the novel finding that this trend is consistent across different habitats and trophic levels, i.e. consumer allochthony is lowest in eutrophic lakes with agricultural catchments and is highest in brown water lakes with highly forested catchments. We have modified the wording of our study hypotheses to better highlight the study novelty and also to cover the important aspects related to responses across habitat zones and trophic levels.*

The work appears largely technically sound, meeting expected standards in the field. Nonetheless, it will be important to answer how well the Bayesian mixing models fit the data, such as on line 131. Some of this information is provided from the credible intervals but something like a R² for the MixSIAR models must be added. More philosophically, I'm not a fan of indirect ordination-based approaches to define covariates. The principal components analysis is particularly messy because forest and agricultural cover are associated with different axes, and DOC seems orthogonal to both. Many of the environmental vectors appear weakly aligned with PC1, and PC1 itself anyway only explains a quarter of the variation in the

environmental datasets. I would have preferred testing the importance of forests directly by including it as a covariate along with other variables. How robust are the results to this, i.e. what happens if you replace PC1 with %forest or %agriculture in the catchment?

RESPONSE: We are happy that Reviewer 1 agrees that our modelling meets the expected standards in the field. We would like to point out that the MixSIAR isotopic mixing model used for estimating consumer allochthony is based on a Bayesian framework and thus does not provide similar statistical metrics (e.g. R^2 or p-values) as conventional statistical tests. The R^2 value is a point estimate and does not account for uncertainty in the parameter estimates and thus does not fit to the Bayesian philosophy (e.g. MacElreath 2020 *Statistical Rethinking* 2nd edition <https://doi.org/10.1201/9780429029608>). However, the presented credible intervals are relatively analogous to R^2 values, where a narrow span of CI is analogous to a high R^2 value and a wide span of CI corresponds to a low R^2 value. We have also explained how the credible intervals in MixSIAR plots should be interpreted in the Figure 3 legend, i.e. that non-overlapping intervals indicate significant shifts in consumer allochthony along the covariate values. Moreover, in the MixSIAR model, linear coefficient of the covariate (PC1) in isometric log-ratio space (ILR space) was clearly non-zero (median [CI] = 0.541 [0.344–0.758], Fig. R2). As our model included only two sources, ILR space is one dimensional, and thus covariate slope differing from zero indicates that the covariate (PC1) had significant effect on consumer allochthony. We have now added this “significance test” also to the Methods section. With these, it is clear that the shown MixSIAR model predicts well consumer allochthony which decreases along the environmental gradient (PC1).

We disagree with Reviewer 1 that “forest and agricultural cover are associated with different axes”, because the two variables are on the same PC1 axis (see the variable loadings in Figure 1 and Table S2), Forest % has high negative loading on PC1 and Agriculture % has high positive loading on PC1. To partly meet the Reviewer’s request for modelling, we have now run the MixSIAR model with different covariate configurations to see which covariates improve the model fit most (Table R1). Based on the leave-one-out cross-validation criterion (LOOic), all the selected environmental factors (PC1, POM N:C ratio, Forest %) produced very similar posterior estimates as the SE for LOOic overlapped in all the models. We have also added a new figure (Extended Data Fig. 4) to show how consumer allochthony changes with increasing forest coverage in the catchment, because this relationship is likely among the most interesting ones for management of boreal landscapes. Our modelling results seem very robust: changing the covariate to e.g. Forest % (or Agriculture %) or POM N:C ratio does not change our results at all, as we believe that these factors are all partly linked, most likely because higher forestry activities in the catchment increase the available t-POM (seen in e.g. POM N:C ratio) and thus promote terrestrial support of lake food webs.

As we did not have an experimental setup where only one environmental factor was changed, we argue that using a combined environmental variable makes most sense and best captures variation across the study lakes and landscapes. We want to highlight that currently it is possible to include only one covariate at the time in MixSIAR models. Therefore, and based on the issues pointed out above, we strongly argue that it makes most sense to use a covariate (i.e., PC1) that captures multiple environmental drivers simultaneously.

Specific comments:

Line 51: This was a synthesis of 140+ sites not a meta-analysis in the strict definition, which would be aggregating existing statistical estimates. This study generated all the statistical analyses fresh using one approach.

RESPONSE: We agree and have now revised the sentence to read: “For example, a recent study of 147 lakes indicated zooplankton to rely on average 11% to 83% on t-OM sources²⁴.”

Line 63-65: I think the novelty could be better stated here given ref 25. I think this has to do with the 19 consumer taxa spanning four trophic levels within benthic and pelagic habitat (line 67).

RESPONSE: The sentence is now revised according to the reviewer’s suggestion

Line 88: The reason for this decline to be expected should be explained. How might forests differ from agricultural land in transport of t-OM and nutrients? Also, aren’t the transport routes and their proximity to the water key here? I can imagine that this isn’t a problem in the strictly forested or agriculture landscapes, but in the mixed landscapes. In a mixed landscape, if forests are more hydrologically connected than agricultural lands, then they can dominate the signal or pattern you’re hoping to detect.

RESPONSE: We have reworded the study hypotheses, which hopefully better clarify the drivers behind expected patterns. We agree that hydrological connectivity can influence the transport of t-OM and nutrients to boreal lakes, which sounds like a fruitful avenue for future research. Unfortunately, we do not have data that would allow detailed modelling of transport routes etc. and we believe our catchment-scale variables capture the main drivers of water quality and transport of t-OM. Our Finnish boreal study lakes typically have catchments that are clearly dominated by either forest or agricultural areas, meaning that very few lakes have a “mixed landscape” where the proximity of forest and agricultural areas to inflow rivers could contribute to variation in consumer allochthony. We would also like to clarify that most forests in Finland are heavily managed for forestry, including ditching and clear cutting, with only 3–5% of forests being protected (Korhonen et al. 2021, <https://www.silvafennica.fi/article/10662/author/20717>).

Line 95: What is a 0.2 loading? Is that a Pearson correlation coefficient?

RESPONSE: We have now clarified this in the Figure 1 legend. Loadings are the projections of the original variables to Principal components. So yes, the loadings are Pearson correlation coefficients between original environmental variables and Principal components. We have now removed the word “excluded” with “are not shown”, as it describes better what we did. The threshold of loading 0.2 was used simply to make the figure more readable as stated in the figure legend. With this procedure we excluded the least important environmental variables for the PCs from the figure. The full set of environmental variable loadings on PCs can be found in Supplementary table 2 and Extended Data Figure 2.

Line 115: The loadings should be reported somewhere to support this conclusion.

RESPONSE: We have now added the loading values to the parentheses for each variable and added a reference to Table S2. These values were originally provided only in the supplementary material.

Line 117: SR ratio should be explained before it introduced in this I-R-D-M article format.

RESPONSE: SR ratio was already explained in line 114. We have now added a more comprehensive explanation of the parameter to the paragraph to improve clarity.

Line 133: Figure 3 is outstanding, but again, it's really unfortunate that we have PC1 on the x-axis because it is so non-intuitive, e.g. what does -2.5% mean vs 2.5 for the % of forest in the landscape?

RESPONSE: We appreciate the positive feedback on the figure. We have added a new supplementary figure (Extended Data Fig. 4) showing consumer allochthony along increasing forest coverage, because this relationship may indeed be highly relevant for the management of boreal landscapes and lakes and this was requested by Reviewers 1 and 3. We agree that the PC scores are probably not the most intuitive ones and therefore we decided to name the x-axis as “Forested catchment, high DOC \leftrightarrow Agricultural catchment high nutrients”. We want to stress here that it is possible to project any new lake on the used PC1 axis with the reported data (raw data + loadings) once all the used parameters are measured from the new. The question of how much PC1 score is altered if forest % changes a bit is relative, and to answer this one would need to know how other parameters such as other catchment land-use types and lake chemical properties change simultaneously.

Line 142: So these pie charts are averaged across all individuals collected across PC1?

RESPONSE: Yes, the pie charts show the main diets of each fish group, measured as average prey proportions of all individuals within each fish group obtained from the study lakes. They are presented only to justify the categorization of the fish to different feeding guilds. We have now clarified this in the figure legend.

Lines 179 and 224: What is it specifically about the composition of OM that influences allochthony? There could be much more focus on mechanisms and explanations here for why different materials result in different levels of allochthony. What is it about these different organic matter sources? Likewise, why is t-OM more important for zoobenthos on line 196?

RESPONSE: We have revised the first paragraph to better clarify the main study findings and the potential mechanisms behind observed patterns. With the OM pool, we are referring to the ratio between autochthonous and allochthonous material. OM in forested lakes consists more of t-OM and thus allochthony of consumers is likely driven by the availability of t-OM in lakes, and it is most likely that microbial link channels this available energy source to higher consumer levels in the food web. We have now added following paragraphs to discussion: “The settlement of allochthonous material to lake bottom before assimilation by microbiota should make allochthonous material more bioavailable in benthic habitats compared to pelagic habitats, for which allochthonous material necessarily needs to stay in either suspension or dissolution to be accessible. This likely explains why we recorded higher allochthony in benthic than in pelagic consumers“

Line 219: See also ref 25.

RESPONSE: Thank you for the suggestion, we have now added the suggested reference. Indeed, it seems that our observed consumer allochthony trends are well in line with previous work for zooplankton.

Lines 224-229: I think this text belongs in the Methods and could be expanded to explain why a POM N:C gradient reflects catchment characteristics and especially PC1.

RESPONSE: Thank you for the comment. We have now revised this part and moved the additional modelling details and the reasoning to the Methods section. We have also clarified our deduction in the Discussion section. The reason why POM N:C gradient reflects the available t-OM in lakes was already clarified in the Methods: “POM N:C-ratio can be used to trace the origin of POM, where low values indicate more terrestrial material and higher values more autochthonous material in POM”. Based on the MixSIAR model results (with PC1, Forest % and POM N:C) and the strong negative correlation between POM N:C ratio and Forest% ($R=-0.77$), it is evident that the high Forest % in the catchment increases t-OM in lakes. As explained in the Discussion, we thus believe that the high availability of terrestrial material in lakes increases the allochthony of consumers across trophic levels.

Line 235: I don't think this conclusion is supported by the data. Surely, hydrology and delivery of that OM is what matters. Afforestation at the margins of a large catchment may have a negligible effect. I wonder how much of the results would change if these effects were considered, e.g. distance to lakes. A good way to measure this would be the R² value or AIC of the MixSIAR models with different covariates.

RESPONSE: Thank you for the suggestion. We have rerun MixSIAR with different configurations to see how different covariates affect the model fit (Table R2). The result of the model comparison suggests that all the models with differing environmental variables (PC1, Forest % and POM N:C ratios) provide almost identical posterior estimates as explained earlier (and seen when comparing figures 3, Ext 3-4). However, we do not have variables that would measure land-use type distance to lakes or the shape of the land use areas nearest to lakes, yet we believe that including such catchment area related weighting factors would describe in a more realistic manner how catchment affects freshwaters. We have revised the concluding sentence to better fit with our data and study findings.

Line 401: PC1 explained "more of the total variance" but in all fairness, this was still small (27.3%).

RESPONSE: Thank you for this comment. We have now removed this sentence and added a clarification for why PC1 was used as a covariate in the MixSIAR model: "PC1 was used in subsequent statistical analyses (MixSIAR) as it has a clear ecological meaning and at the studied lake gradient multiple different parameters are changing simultaneously."

Line 481: Why use trophic level 2 as the isotopic baseline? This estimation doesn't include any uncertainty as far as I can tell, i.e. the TL estimates are fixed. Omega (line 489) seems to suffer from a similar problem, though the sensitivity analysis in Expanded Data Fig 4. alleviates this concern. Should the same thing be done for TL?

RESPONSE: Thank you for the comment. Long-lived primary consumers are typically used as isotopic baselines in food web and dietary studies, because their isotope values integrate temporal and spatial isotopic variation unlike those of rapidly changing primary producers (e.g. Layman et al. 2011 and references therein; <https://doi.org/10.1111/j.1469-185X.2011.00208.x>). Moreover, obtaining sufficient samples of pure phytoplankton or benthic algae for isotopic baselines is notoriously difficult in oligotrophic and dystrophic lakes, whereas cladoceran zooplankton are easily collected with replicate hauls with a plankton net.

We have now added additional uncertainty ($\pm 13\%$) to omega corrected $\delta^2\text{H}$ values of consumers that account variability in the trophic level baseline (cladocera $TP=2.0\pm 0.1$) (Tanenztap et al., 2017) and to TDF of nitrogen (3.4 ± 1.0) (Post 2002). The additional uncertainty value was derived from the mean difference of consumer omega corrected $\delta^2\text{H}$ values from scenarios i) clado $TP=1.9$, $\delta^{15}\text{N}$ TDF=4.4 and ii) clado $TL=2.1$, $\delta^{15}\text{N}$ TDF=2.4. In MixSIAR package, there are limited possibilities to add this kind of variation to the model, but it is possible to do through the tracer trophic discrimination matrix. Here we followed the procedure in a recent paper that added also additional uncertainty to consumer isotope values (Vane et al. 2023).

Vane, K. et al. (2023) Tracing basal resource use across sea-ice, pelagic, and benthic habitats in the early Arctic spring food web with essential amino acid carbon isotopes. *Limnol. Oceanogr.* 68, 862–877.

Line 498: Given how essential this number is to the entire analysis, there should be much more detail on how this number was derived, how it was modelled, how good the model fit the data, and why this difference between measured benthic algae and pelagic water is an appropriate assumption for phytoplankton 2H, including how this assumption has been validated.

RESPONSE: Thank you for the feedback. We calculated the mean difference between $\delta^2\text{H}$ values of lake water and the sampled “pure benthic algae” across the study lakes. The value we obtained is very similar to those based on the best available literature review (Brett et al. 2018) after the re-evaluation of phytoplankton $\delta^2\text{H}$ values due to the found errors in previous standard values based on Soto et al. (2017). The reason for taking the average discrimination factor across all study lakes was related to the problems with sampling of pure algae as mentioned in the Methods section. The variability of this discrimination factor or algal source $\delta^2\text{H}$ was based on spatiotemporal sampling of four lakes (Table S3) and is larger than that suggested only based on the discrimination factor distribution reported by Brett et al. (2018).

Line 504: What does "individually" mean here? You fitted MixSIAR twice?

RESPONSE: Sorry for the confusion here. We have now modified the sentence: “We used two lake specific sources...”.

Line 505: How appropriate is it to extrapolate error from 4 lakes to the entire dataset? Presumably variability in DOM depends on landscape characteristics and lake morphometry, so were these four lakes representative?

RESPONSE: It would have been best to sample all the lakes with spatio-temporal resolution, but unfortunately, we did not have time and resources to do so. The four lakes selected for the intensive sampling were, however, representative for the study region as they included small and large lakes with differing water chemistry and catchment characteristics (Table S1 and S3). Given also that these lakes were sampled on many sites in the lake and once each season for 1–3 consecutive years, we believe the error variation we found in this exercise provides a credible estimate of the isotopic error variation in producers and consumers in space and time one might encounter in all the lakes we studied. In the revised manuscript, we have elaborated this to improve clarity. Moreover, the measured spatiotemporal variability of algal $\delta^2\text{H}$ values ($SD = 23.6\%$) corresponds well with those previously reported by Brett et al. (2018) and is now recalibrated (based on Soto et al. 2017) with photosynthesis fractionation values ($SD = 17.9\%$). We would like to stress that the used terrestrial source in the models were sampled from the inlets, thus the catchment characteristics can be expected to be the main driver for $\delta^2\text{H}$ values of terrestrial sources.

Reviewer #2

Review of NCOMMS-23-44093-T "Land use drives terrestrial support of boreal lake food webs"

When reviewing this paper, I focused almost entirely on the methods these authors employed. I will also note that in 2018 I published a paper on how the results of deuterium-based food web analyses are very strongly influenced by several poorly constrained assumptions (Brett et al. 2018). Nearly all of the problems noted in that analysis are apparent in this manuscript. I will detail my main concerns below.

1) As noted in Brett et al. (2018), deuterium-based food web analyses are extremely sensitive to the assumed environmental water (ω) value. Environmental water is the proportion of a consumer’s hydrogen content obtained directly from water, as opposed to its diet. Brett et al. (2018) showed that when the environmental water contribution to consumers was directly measured in laboratory experiments (where it can be easily quantified), it averaged 0.27 ± 0.11 (± 1 St.Dev). The authors of NCOMMS-23-44093-T used an environmental water assumption of 0.22. In the MS they state “We used an omega value of 0.22 based on the best literature assessments (Wilkinson et al. 2015)”. This claim is highly dubious because the analysis by Brett et al. (2018) showed the Wilkinson et al. (2015) study had a series of egregious errors in their compilation of omega values (see pages 1081-1082 of Brett et al.). Most importantly, more

than half of the estimated omega values reported by Wilkinson et al. were obtained from open systems where it is impossible to actually quantify omega because you cannot know what they actually consumed in an open system. Omega can only be quantified in closed systems where only a single diet item is used (and therefore the diet, water and consumer $\delta^2\text{H}$ values are exactly known). Secondly, Wilkinson et al. (2015) reported their summary of omega estimates had a standard deviation of 0.02. However, even the most cursory examination of the data these authors presented in their Table 2 and Figure 1 make it immediately obvious that the true standard deviation of these data is much larger, i.e., ± 0.11 . Thirdly, Wilkinson et al. claimed that the average omega estimates from open and closed systems were not significantly different, when clearly they were (unpaired t-test; $t = 2.93$, $df = 22$, $P < 0.01$). Finally, Wilkinson et al. (2015) also included two estimates of omega from a study that didn't even measure this parameter (Jardine et al. 2009). Based on this remarkable series of errors, one cannot possibly claim that the estimated omega values from Wilkinson et al. (2015) represent the "best literature assessments". Ironically, if the numerous errors in Wilkinson et al. are corrected, the omega values they summarized from closed systems had a mean \pm SD of 0.28 ± 0.10 , which is nearly identical to the results from Brett et al. (2018), i.e., 0.27 ± 0.11 . Since the authors of NCOMMS-23-44093-Cited Brett et al. (2018) when they noted "model outputs of consumer allochthony are known to be sensitive to the selected omega (ω) values", I also suspect they are well aware of the raft of errors in Wilkinson et al. (2015).

RESPONSE: We thank Reviewer 2 for the comprehensive and detailed comments on our manuscript. We agree that deuterium-based food web studies are sensitive to the used omega values. This is reflected by our sensitivity analysis, where the used omega values encompass those used in numerous previous studies, but the overall patterns seen in consumer allochthony remain essentially the same. We want to emphasize that the main aim of our study is NOT to provide absolute estimates of the degree of consumer allochthony, but instead show how reliance on t-OM changes along a gradient in catchment properties of boreal lakes. Thus, we believe the selection of "correct" omega value is not as crucial in our study as in studies aiming to truly quantify consumer allochthony. In reality, there probably is no "one true omega value" because how the hydrogen isotopes are incorporated from diet and water into consumer tissues likely depends on numerous physiological and environmental factors, such as water temperature, an individual's physiological status and composition of consumed prey. Such issues may eventually be resolved in the future, but for the present modelling we had to make use of the best available knowledge related to omega values.

We suspect that at the time the review paper by Brett et al. (2018) was written, the authors may have been unaware of the errors in hydrogen isotope standards that had only recently been reported by Soto et al. (2017). This means that almost all the experimentally determined omega values reviewed by Brett et al. (2008) were incorrect. Of the articles reviewed by Brett et al. (2018), the only ones that did not suffer from wrong $\delta^2\text{H}$ standard values were Maco et al. (1983) and Newsome et al. (2017). We have now corrected the omega values reported by Brett et al. (2018) (Table R1). The experiments controlling environmental water $\delta^2\text{H}$ values that have used wrong hydrogen isotope standards decreased ca. 4% points after recalibrating the data with equations from Soto et al. 2017. Retaining the omega values reported by Brett et al. (2018) that did not suffer from the standard bias results into mean omega value of 0.231 ± 0.087 (Mean \pm SD). Please find the corrected omega values in Table R1 at the end of this letter. We have now used an omega value of $0.23 (\pm 0.09)$ in all our consumer allochthony models.

As our calculations are preliminary and we had to mine most data from figures in the papers reviewed by Brett et al. (2018), we believe that our used omega value of 0.23 is very close to the true average omega values. Soto et al. (2019) showed that after the recalibration of the consumer $\delta^2\text{H}$, omega values decrease by 7% for Chironomidae and 5% for guppy. Therefore, we are confident that the recalibrated $\delta^2\text{H}$ values and recalculated omega values in Table R1 are very close to the truth. We accept that the omega values reported by Wilkinson et al. (2015) and Brett et al. (2018) have the same bias as they refer largely to the

same publications. We have now modified the sentence referring to the selected omega values and cited to the publications by Wilkinson et al. (2015), Brett et al. (2018), Soto et al. (2017) and Soto et al. (2019).

Soto et al. (2017). Re-evaluation of the hydrogen stable isotopic composition of keratin calibration standards for wildlife and forensic science applications. *Rapid Communications in Mass Spectrometry*, 31, 1193–1203. <https://doi.org/10.1002/rcm.7893>

Soto et al. (2019). Terrestrial contributions to Afrotropical aquatic food webs: The Congo River case. *Ecology and Evolution*, 9, 10746–10757. <https://doi.org/10.1002/ece3.5594>

Why do the numerous errors in Wilkinson et al. (2015) even matter? By claiming that the Wilkinson et al. (2015) environmental water estimate compilation represents the “best literature assessments” the authors of NCOMMS-23-44093-T can use a substantially lower and less variable assumed omega value than indicated by the actual published literature. Specifically, the authors of NCOMMS-23-44093-T used a point estimate of omega = 0.22 for their main conclusions and a range of omega = 0.10 to 0.30 values for their sensitivity analysis.

It is also noteworthy that in their sensitivity analysis reduced the point estimate of omega by 0.12 but only increased it by 0.08. As expected, their reported sensitivity analysis showed their estimated terrestrial contributions to the consumer hydrogen content was very dependent on the environmental water assumption used for their analyses. I believe very strongly that the only defensible point estimate and range for omega at this time is that based on the results summarized in Brett et al. (2018), i.e., omega = 0.27 with a range of 0.16 to 0.38 for the sensitivity analysis.

RESPONSE: We have now corrected the statement that the best available literature research of omega values is Brett et al. (2018). Moreover, we have now used symmetrical omega ranges in the sensitivity analysis: 0.14-0.32 (Fig Ext. 4). This range was derived from $\pm 1SD$ of our recalibrated omega values (Table R1).

2. The use of Bayesian mixing models is not warranted, and could in fact greatly obscure problems in the underlying dataset. People in the scientific community are often in awe of complex models, and the Bayesian dietary mixing model MixSIAR certainly fits this profile. The problem with any of the popular Bayesian mixing models is that they will always provide a solution within the 0-1 domain no matter how inappropriate the underlying data are. For example, the consumer being analyzed could have stable isotope values much higher or lower than either of the terrestrial or aquatic resources considered in the model. In these cases, the underlying model is obviously mis-specified, for example, the consumer may be utilizing a resource not included in the model, the stable isotopes values used to represent the consumer and resources may be wrong, or the assumed omega value could be incorrect. These sorts of problems get entirely brushed under the rug by the popular Bayesian mixing models.

RESPONSE: Resource utilization of a consumer cannot be negative nor over 100%. As reported, in our case most samples were between the sources $\pm 2SD$. We have conducted a previous study with the suggested simple arithmetic model (Keva et al., 2022) estimating consumer allochthony, including a small fraction of the data presented in this study. We observed exactly similar trends in consumer allochthony along similar environmental gradient with the individual based arithmetic modelling and the population level Bayesian dietary modelling. The aim of this study is not to determine exactly the proportional allochthony of the consumers, it is rather what happens to consumer allochthony along the studied environmental gradient. We accept that the allochthony estimations include various biases, but as we deal with only the data we have gathered from many lakes, all the samples can be assumed to have similar biases (source, water and consumer samples collected and prepared in similar manner in all lakes). Therefore, any possible biases in the allochthony estimations will not influence the direction of how environmental gradients affect consumer allochthony.

In fact, the calculations needed to estimate terrestrial contributions to aquatic consumers are very simple algebraic mass balance expressions (see equations 1 and 2 in Brett et al. 2018), thus there is no justification for not dealing with these in their original unadulterated form. If you analyze the data using the original equations you will readily see cases where the model is mis-specified as calculated outcomes that are substantially more than 100% or less than 0% for either the terrestrial or aquatic resources.

RESPONSE: *We agree that the referred equations are simple. In fact, the equations presented in Brett et al. (2018) are essentially the same as presented in our Methods section but with one difference: the equations in Brett et al. (2018) assume the trophic level difference between zooplankton and sources to be exactly 1. Thus, the presentation we do in this manuscript is more comprehensive and generalized. We would also like to point out the fundamental differences between the Bayesian and the simple linear mixing models: the former gives population-level dietary probabilities, whereas the latter gives individual-level estimates of dietary proportions. As our paper focuses on population-level responses, we do not see any reason to change our statistical approach.*

The authors justify their Bayesian analyses on lines 75-77 when they state “Most (~99%) of the sampled consumer $\delta^2\text{H}$ values fell inside the lake-specific range of $\delta^2\text{H}$ values of basal resources (inlet dissolved t-DOM $\pm 2\text{SD}$ & algae $\pm 2\text{SD}$; Fig. 2) as required by the model.” This means the observed consumers (after environmental water correction) could be as much as 22-48 $\delta^2\text{H}$ outside of the envelope described by the resource stable isotope values. I don’t have access to these authors raw data, but that seems like a very large disparity and seems to set the bar very low for model misspecification. I was curious how much error this level of uncertainty could introduce to their calculations. Since I don’t have access to the raw data from NCOMMS-23-44093-T, I based my sensitivity analysis on the mean $\delta^2\text{H}$ values for zooplankton, water, t-POM, and phytoplankton from the Tanentzap et al. (2017) study as reported in Table 6 of Brett et al. (2018). I also used the mean zooplankton trophic level from Tanentzap et al. I then used the assumed omega value (= 0.22) from NCOMMS-23-44093-T, as well as the $\pm 2\text{SD}$ values for DOM and benthic algae from their Table S3. My cursory sensitivity analysis indicated adding $\pm 2\text{SD}$ to the resource stable isotope values could introduce tremendous uncertainty to the estimated allochthony values. For the conditions described above, I calculated allochthony values from as high as +48% to as low as -223% using the algebraic approach.

RESPONSE: *We believe there must be some confusion with the simple individual-level arithmetic dietary estimations and population-level dietary estimations with the Bayesian approach. As clearly the majority (ca. 99%) of the samples are inside the source envelope, the model is not clearly mis-specified. The MixSIAR model estimates the most probable dietary proportions of the consumer population, including the variability in consumer and source tracer values. As there is always more than one sample per consumer population, it is not a concern if one consumer sample is slightly outside the mixing polygon depicted by the source values (mean $\pm 2\text{SD}$). We do understand that it was hard to read the data from Figure 2 and therefore we have now provided the original consumer $\delta^2\text{H}$ data in the supplementary material repository. We are concerned about the arithmetic calculations and the values reported by Reviewer 2. For example, for littoral chironomids (see Fig 3 d3), the black points to the furthest left of the axis are clearly on top of the allochthonous source, and there is huge distance to autochthonous source, thus they have to get higher values than +48% of allochthony in algebraic dietary estimations. Therefore, we believe that there is some sort of error in the algebraic calculations conducted by Reviewer 2 and thus do not fully understand what the point of this comment is.*

3. Another closely related problem with Bayesian mixing models is that messy (or highly variable) data generally give you the uninformed prior assumption as the model output (Brett 2014), which in the case of NCOMMS-23-44093-T would be 50% terrestrial and 50% aquatic resource utilization. Since the original algebraic assumptions don’t assume a prior, they are not subject to this bias towards all resources

being equally important. When adding a lot of uncertainty to the potential resource stable isotope values, you are also greatly increasing the likelihood that the Bayesian model will just return the prior assumption.

RESPONSE: *We would like to emphasize that no informative priors that could have influenced the model outputs were used in the MixSIAR model. We are slightly confused by what Reviewer 2 is referring to in this comment. We have now added the lake and species-specific posterior distributions in the repository, where one can clearly see that our model did not produce much overlapping dietary estimates and are not messy at all. In addition, overlapping dietary estimates between two sources are not wrong by default, but instead can be ecologically very well explained and justified.*

4. One of the most concerning things about this study is just how little of it is based on actual field data for the variables of interest. For example, on lines 503-504 the authors state “For each lake, we had individually analysed two sources: inlet t-DOM and modelled algae (aquatic).” This is problematic because zooplankton, benthic invertebrates and the other consumers in their lakes do not in fact consume t-DOM! They directly consume t-POM. That is the invertebrate and vertebrate consumers in lakes directly utilize particulate organic matter NOT dissolved organic matter. Why didn’t the authors directly determine the stable isotopic values for the t-POM? Are these authors assuming t-POM and t-DOM have same hydrogen stable isotope values? If so, they need to provide some support for this assumption. Similarly, lines 503-504 of the MS state they used “modelled algae” for their analysis. This part is confusing because elsewhere in the MS I got the impression that they actually used observed benthic algae values and modeled phytoplankton values. This also needs to be explained more clearly. Overall, it is problematic that apparently only two of the five variables needed for the authors’ analyses were directly determined. Specifically, the $\delta^2\text{H}$ values for the lake water and consumers were directly determined, but the $\delta^2\text{H}$ values for the terrestrial and aquatic resources were estimated in some way, and the omega values were assumed. Similarly, the resource standard deviation values used in the Bayesian modeling are not based on observations from the actual lakes, rather they are based on observations for a subset of 4 of the 35 sampled lakes. Given the dearth of real data in this study, it is hard to imagine the estimated allochthony values for these lakes are even remotely similar to the true values.

RESPONSE: *We disagree with Reviewer 2 stating that zooplankton use t-POM not t-DOM. We would argue that most evidence (field studies and laboratory experiments) indicates that zooplankton in lakes do NOT make significant use of t-POM and that t-DOM is actually the main source for allochthony and is channeled to zooplankton via a microbial link (e.g. Jones et al. 1992, Tanenzap et al., 2017, Hiltunen et al., 2017). Also, benthic grazers would not use t-POM, and even shredders probably actually gain nutrition from the microbes associated with terrestrial leaf litter rather than the leaf matter itself, and hence also exploit a microbial link similar to that leading to zooplankton. Hence, our choice of inlet t-DOM as the allochthony source value was deliberate and justified. Further, our use of benthic algae as the source value for aquatic resources is based on real values and, while obviously contains some level of uncertainty, can hardly be said to be not determined directly, and is probably also a realistic surrogate to provide values for photosynthetic discrimination of $\delta^2\text{H}$ by phytoplankton (which cannot normally be obtained directly from field material).*

In the Methods section, we have explained that the t-DOM values were measured from samples collected from the inlet streams of each lake. Moreover, we have sampled benthic algae from each study lake. Thus, we took the average discrimination factor from the analysed samples and used that with the lake water to model algae $\delta^2\text{H}$ values. The resulting discrimination factor (-106.7‰) is actually very close to that reported by Brett et al. (2018) after recalibration (-111.1±17.9‰) according to Soto et al. (2017). We do not fully understand why Reviewer 2 argues that we have used little field data for the variables of interest, because that is evidently not the case in our study.

Jones, R. I. The influence of humic substances on lacustrine planktonic food chains. *Hydrobiologia* **229**, 73–91 (1992).

Tanentzap, A. J. et al. Terrestrial support of lake food webs: Synthesis reveals controls over cross-ecosystem resource use. *Sci. Adv.* **3**, e1601765 (2017).

Hiltunen et al. Trophic upgrading via the microbial food web may link terrestrial dissolved organic matter to *Daphnia*. *Journal of Plankton Research*, **39**, 861-869 (2017).

Minor points:

5. Why were cladoceran zooplankton assumed to have a trophic level of 2 (which implies they are completely herbivorous)? It well established in the zooplankton feeding literature that cladocerans also feed on microzooplankton. It is conceivable that cladocera have a slightly higher trophic level (e.g., 2.1)? The strict herbivory assumption needs to be justified or modified.

RESPONSE: *It is true that even small cladocerans such as Daphnia and Bosmina filter also protozoans and thus the assumption of strict herbivory is incorrect. We have now added additional uncertainty ($\pm 13\%$) to consumer omega corrected $\delta^2\text{H}$ values to account for variability in the trophic level baseline (cladocera $TP=2.0\pm 0.1$) (Tanentzap et al., 2017) and in the TDF of $\delta^{15}\text{N}$ (3.4 ± 1.0) (Post 2002). In MixSIAR package, there are limited possibilities to add this kind of variation to the model, but it is possible to do so through the tracer trophic discrimination matrix as done in a recent paper by Vane et al. (2023).*

6. The expectations outlined in lines 82-89 of the MS line up almost perfectly with the reported results from this study. This begs the questions, were the authors of this study very good at predicting its outcome, or where the expectations modified after the results were known?

RESPONSE: *Given the long experience of several of the authors in this field, we think we probably were rather good at predicting the likely outcome of the study. We have conducted a previous study using partly the same (one consumer: large perch) data in arithmetic mixing calculations (Keva et al. 2022). Therefore, we indeed had some expectations for what may happen to allochthony of large perch along the studied environmental gradient. Corresponding consumer allochthony patterns across catchment land-use gradients have been reported also by previous studies cited in the manuscript. It should be quite obvious for the reader why we expected to see lower allochthony in pelagic zooplankton compared to littoral or profundal benthos: zooplankton is harvesting mainly pelagic algae, whereas benthos consume mainly biofilm and allochthonous and autochthonous detritus. We have now modified the sentences referring to our expectations and believe they are well justified based on previous literature and ecological theories.*

Keva, O. et al. Allochthony, fatty acid and mercury trends in muscle of Eurasian perch (*Perca fluviatilis*) along boreal environmental gradients. *Sci. Tot. Environ.* **838**, 155982 (2022).

7. Lines 526-529, as noted previously using “high variance in source values” won’t actually account for seasonal fluctuations (since these are unknowable based on a single sampling date), but it will make it much more likely that a Bayesian mixing model will just give you the prior assumption (Brett 2014).

RESPONSE: *We would like to highlight that our source isotope values are based on samples collected in different seasons and sites (Table S3), thus incorporating both seasonal and spatial variation. We would also like to remind that we have used uninformative priors in MixSIAR and therefore we are slightly confused of what the Reviewer 2 is referring to in this comment.”*

8. I disagree with the authors' conclusion on lines 531-533, i.e., "As our consumer data were collected during the summer growing season when autochthonous algal production is highest, our zooplankton allochthony values are likely underestimating the annual allochthony of these pelagic taxa." In fact, temperate or boreal lakes the vast majority of zooplankton production occurs during the summer because this is when zooplankton community biomass is greatest and the water is warmest and therefore metabolic rates are the highest (Zhang et al. 2022).

RESPONSE: *We disagree with Reviewer 2 on this issue and would like to note that the study by Zhang et al. (2022) is essentially a generalised model based on four lakes of quite different characteristics as compared to our 35 boreal study lakes. In fact, several previous studies have shown that the allochthony estimates of zooplankton in summer are indeed lower than in other seasons (e.g. Grey et al., 2001; Tanentzap et al. 2017), due to the high phytoplankton availability in summer. We have now added this reference to the sentence.*

Grey, J., Jones, R. I. & Sleep, D. Seasonal changes in the importance of the source of organic matter to the diet of zooplankton in Loch Ness, as indicated by stable isotope analysis. Limnol. Oceanogr. 46, 505–513 (2001).

9. Lines 538-539, why did the authors use an asymmetrical range of omega values for their sensitivity analysis. This seems like tilting the scales.

RESPONSE: *Thank you for highlighting this anomaly. We have now updated the figure with symmetrical omega values.*

Signed, Michael T. Brett

Brett, M.T. 2014. Resource polygon geometry predicts Bayesian stable isotope mixing model bias. *Marine Ecology Progress Series* 514, 1-12.

Brett, M.T., G.W. Holtgrieve, and D.E. Schindler. 2018. An assessment of assumptions and uncertainty in deuterium-based estimates of terrestrial subsidies to aquatic consumers. *Ecology* 99: 1073-1088.

Jardine, T. D., K. A. Kidd, and R. A. Cunjack. 2009. An evaluation of deuterium as a food source tracer in temperate streams of eastern Canada. *J. N. Am. Benthol. Soc.* 28: 885–893.

Tanentzap, A. J. et al. 2017. Terrestrial support of lake food webs: Synthesis reveals controls over cross ecosystem resource use. *Sci. Adv.* 3, e1601765.

Wilkinson, G. M., Cole, J. J. & Pace, M. L. 2015. Deuterium as a food source tracer: Sensitivity to environmental water, lipid content, and hydrogen exchange. *Limnol. Oceanogr. Meth.* 13, 213–223.

Zhang, C., M.T. Brett, J.M. Nielsen, G.B. Arhonditsis, A.P. Ballantyne, J.L. Carter, J. Kann, D.C. Müller-Navarra, D.E. Schindler, J.D. Stockwell, M. Winder, D.A. Beauchamp. 2022. Physiological and nutritional constraints on zooplankton productivity due to eutrophication and climate change predicted using a resource-based modeling approach. *Can. J. Fish. Aquat. Sci.* 79 (3), 472-486

Reviewer #3 (Remarks to the Author):

I have followed the literature on terrestrial contributions to aquatic consumers (allochthony) closely the last 15 years. The research topic has partly stagnated over time, but the ms by Keva et al. (NCOMMS-23-44093-T) takes a great leap forward by its unprecedented analysis of multi-taxa allochthony at the

landscape level. Although the study is limited to a snapshot picture of 35 lakes, the inclusion of a large number of taxa and organism groups makes it one of the largest existing SIA-based assessments of allochthony. The ms presents the first systematic evidence for strong broad-scale patterns in allochthony across gradients in boreal forested and agricultural landscapes, hence it has high news value and should be considered for publication in a high-impact journal. However, I have a few relatively minor comments, mainly about the overall statistical approach and the methods choices.

RESPONSE: *We thank Reviewer 3 for constructive and helpful feedback on our manuscript. We agree that this is one of the largest existing isotope datasets for modelling allochthony of different benthic and pelagic consumer taxa and trophic levels across landscape gradients. Please find below our responses to the specific comments, which we have aimed to incorporate when revising the manuscript.*

General comments

1. Patterns in allochthony showed strong links to the first axis of a PCA run on a range of environmental variables. However, I see a risk of basing the landscape analysis only on links to this single ordination variable (PC1). The variable explained only 27% of the environmental data, out of which the catchment composition variables % Forest, % Agriculture and % Urban contributed partly. If being cynical, one can argue that a PC variable can be made to behave the way we want, by playing around with what variables to include and not include in the PCA. There are more sophisticated methods to disentangle the influence from different landscape and environmental variables on a dependent variable. I can see the elegance of the simplicity in the authors' analysis, and I am not proposing that a change is critical, but a PCA has limitations. Since the paper makes a big deal about the importance of forest, the authors could at least show that there are direct links between % Forest as a single variable and allochthony, in my humble opinion. Such an analysis could be included in supplementary information.

RESPONSE: *We accept that PC axes can be hard to interpret. At the same time, in our study PC1 captures several environmental drivers that can influence lake productivity and food-web dynamics, including reliance of different consumers on t-OM. In our case, the PC1 forms a clear gradient from forested, acidic, brown water lakes towards eutrophic lakes. To address the concerns raised by Reviewer 1 and 3, we have now rerun MixSIAR models with different configurations to see how inclusion/exclusion of individual covariates (i.e., environmental variables) influence the model fit (Table R1). Moreover, as our main study findings indicate forest coverage to be a major driver, we have added a new supplementary figure (Extended Data Fig 4) to show the individual effect of this variable on consumer allochthony. However, we decided to retain the original figure with PC1 as the covariate in the main text, because this model and figure better reflect the large variation in different environmental drivers. We have now revised the discussion related to the mechanisms behind Forest % impacts on consumer allochthony and hope it reads better. Catchment and lake chemistry properties alter the availability of t-OM and this is channeled to consumer allochthony through a microbial link.*

2. The method used to assess allochthony from stably hydrogen isotopes and a Bayesian model is state-of-the-art and based on standard assumptions. Nonetheless, uncertainties of such a model are inherently very large given the variability in algal vs terrestrial isotope signatures and contributions to consumer hydrogen from dietary water. The ms does a decent job at addressing these uncertainties in the "Caveats" section, and there is a sensitivity analysis for the possibly most uncertain parameter "omega" (per-trophic-level H contribution from dietary water). Authors make a good point about the overall patterns in allochthony being qualitatively the same regardless of model assumptions, even if the absolute allochthony values vary widely with the assumptions. However, there is one view on uncertainty that I find lacking: the way the model works, propagation of uncertainties is disproportionately increasing (in an exponential manner) with trophic level. This means that the allochthony results for herbivorous zooplankton are much better confined than those of, e.g., TL-4 fish or zoobenthos. At some trophic level

it can be discussed if it is meaningful at all to model allochthony from hydrogen. This is something that needs to be addressed.

RESPONSE: *This is a very important point, and we agree that propagation of uncertainties increases with the trophic level of consumers. Originally, we had briefly explained this in the Discussion and in more detail in the Methods section, but we have now added more text about this issue to the main Discussion section.*

3. Whereas the Bayesian allochthony modelling is well explained, I have a hard time following how the trend analyses were carried out. There is literally no statistics provided for the trend analyses in Fig. 3 and elsewhere, and little explanation provided in the methods section. I suspect that this is based on some kind of curve fitting within a Bayesian framework, but also with Bayesian statistics it can be shown how probable the hypothesis of a trend is. In, for example, the extended fig 4 it would really help to know which curves are “significant” or not, because some of those trends at alternative omega scenarios look very weak.

RESPONSE: *The shadings accompanying the trend lines in Figure 3 show 95% Bayesian credible intervals. While the Bayesian models do not provide p - or R^2 -values like the conventional statistical tests of linear models do, one can interpret non-overlapping credible intervals as statistically significant changes or differences. In other words, if the credible intervals show no overlap at specific points or covariate values (here PC1 values), then the response variable (here consumer allochthony) differs significantly. In Figure 3, one can see that the credible intervals overlap across the PC1 axis for example in subplots “a5 Copepods” and “d2 Smelt”, meaning that these two consumers show no significant shifts in allochthony along the PC1 axis.*

MixSIAR does the covariate fitting (linear correlation) in isometric log-ratio space as it calculates the consumer dietary mixes there. The lines shown in Fig 3 are invILR-transformed (transformation back to p -space) estimates of covariate effects on consumer allochthony retrieved with `plot_continuous_var()` function in MixSIAR package. This back transformation causes the bending of the linear curve as explained in the MixSIAR manual: “MixSIAR fits a continuous covariate as a linear regression in ILR/transform-space. The plotted line uses the posterior median estimates of the intercept and slope, and the lines are curved because of the ILR-transform back into proportion-space. -- If the model contains both a continuous AND a categorical (factor) covariate, MixSIAR fits a different intercept term for each factor level and all levels share the same slope term”.

The 95% credible intervals shown in Fig 3 are derived from the model output as well with `plot_continuous_var()` function, which calculates the 95% CI from the model output by adding the continuous covariate effect (environmental variable) on consumer allochthony after accounting for the effects of lake and species factors.

We have now provided posterior distributions of the linear regression coefficient with PC1 and dietary mixtures in ILR space in Fig R1. The linear coefficient of PC1 in ILR space was clearly non-zero (median [CI] = 0.541 [0.344–0.758], Fig. R1). As our model included only two sources, ILR space is one dimensional Euclidean space, meaning that a slope differing from zero indicates that PC1 had significant effect on consumer allochthony. We have now added this “significance test” to the Methods section.

4. Something to remember about allochthony estimates is that they are relative to the biomass. This means that there, hypothetically, can be high “allochthonous biomass” and high rates of “allochthonous production” in a lake with low allochthony, or vice versa; a low percentage of a high biomass could represent high absolute incorporation of allochthonous carbon in the food web. It would be interesting to know if the authors have biomass data, such that it can be discussed whether or not the incorporation of

terrestrially-derived organic matter was higher in the lakes with high allochthony than in lakes with low allochthony. It could also be that the lakes with most allochthony were very unproductive and that the incorporation of terrestrially-derived organic matter in those lakes was small in quantitative terms.

RESPONSE: We agree that it would be very interesting to see how consumer allochthony is associated with consumer biomass, as done e.g. in the important study by Karlsson et al. (2009; <https://doi.org/10.1038/nature08179>). It would be very interesting to estimate the total allochthony of lake food webs with this approach. Unfortunately, we lack quantitative data for fish and invertebrate biomasses and thus are unable to conduct such statistical analyses in our study lakes. However, this topic is certainly an important avenue for future research.

Specific remarks

1. Line 13 and elsewhere. Not sure the narrative is needed about the topic being controversial. Indeed, there was a hot debate about SIA-based allochthony 15 years with a couple of loud voices, but today allochthony is not controversial. Rather emphasize the knowledge gaps.

RESPONSE: This is a valid comment raised also by Reviewer 1. We have replaced the word “controversial” with “unexplored” and also revised the Introduction section by emphasizing the knowledge gaps and the novelty of our study.

2. Line 16. Vague phrasing “variation related”. Should be explained how the trend looked like more specifically - not just that there was a relation.

RESPONSE: We have now added the direction of the relation to the sentence.

3. Line 25. Urbanization is a process, not a land use

RESPONSE: We have corrected this to “human settlements”.

4. Line 29. Repeated word Considering ... considered

RESPONSE: We have deleted the words “now considered”.

5. Line 43. See specific remark #1

RESPONSE: We have replaced “controversial” with “unexplored” and changed some of the references to indicate that previous studies have typically focused on a single trophic level and/or food-web compartment, such as pelagic zooplankton.

6. Fig. 1 caption. The figure would be more readable if the lakes are numbered from 1 and onwards, instead of using the three-letter code which make the figure look messy.

RESPONSE: We thank for this suggestion. Indeed, the three-letter code does not tell much for the first-time reader. However, once reader checks Table S1, it is much easier to identify the lakes than if there were only numbers, as the three letter codes are formed based on the three first letters of each lake. Yet we have now redrawn the figure according to Nature Communications’ guidelines. We do not feel that the map is currently too messy to understand and we note that neither of the other two reviewers expressed any concerns about this.

7. Fig. 2 Consider adding all the isotope raw data (including 2H in water) as supplementary information. This increases transparency and makes it easier for others to repeat the analyses, or use the data as part of future meta-analyses of many data sets.

RESPONSE: Thank you for the comment. The $\delta^2\text{H}$ data of consumers, lake water and sources will be published in an open GitHub repository once the manuscript has been accepted. Here is a link to the mirrored repository [<https://gitfront.io/r/osjomike/Aw9SMDpHgtri/Keva-et-al.-2024-NC/>] where you can find all the data and R scripts used in the MixSIAR runs and the corresponding figure production.

8. Paragraph beginning on line 199. Here or possibly in the preceding paragraph it could be discussed that allochthony estimates at higher trophic levels are relatively much more uncertain. See general comment #2.

RESPONSE: We have now added this to the Discussion section. This caveat was originally discussed in the Methods section.

9. Lines 250-256. I find the management implication a bit unclear. This is also not conclusion, but rather “outlook” so it appears misplaced under the conclusion heading.

RESPONSE: We have now moved this to the end of the Discussion section.

10. Lines 418-419. Were the zooplankton gut-evacuated? If not, gut contents of terrestrial particles may have biased the results.

RESPONSE: Zooplankton samples were kept in a jar for several hours before picking and freezing the samples, yet most likely zooplankton were frozen before they emptied their stomachs. However, the contribution of gut mass on e.g. cladoceran total mass is low (ca 5%; Feuchtmayr and Grey 2003) and thus also the confounding impacts can be considered minimal here. We agree that due to the sampling and handling procedures our zooplankton allochthony models might slightly overestimate their allochthony.

Feuchtmayr H., & Grey J. (2003) Effect of preparation and preservation procedures on carbon and nitrogen stable isotope determinations from zooplankton. *Rapid Communications in Mass Spectrometry*, 17, 2605–2610.

11. Lines 465-466. Arbitrary effects from exchangeable hydrogen is a classical problem in deuterium measurements. Can the authors comment on the advantage and disadvantage of the selected method (storing the samples open for 5-d equilibration in lab air), compared to steam equilibration?

RESPONSE: We appreciate this methodologically perceptive comment. Indeed, the Dual Vapor Equilibration (DVE) would have been a better and more precise method compared to a Comparative Equilibration (CE) method used here. At the time the samples were analyzed, Uniprep devices were still under development and to our knowledge the first version of the Uniprep had serious faults. The recent instrument (Uniprep2) is supposed to be more reliable, but it was released long after our data was collected. The only disadvantages of the DVE are practical: it is a more complex analysis and requires concurrent analyses. CE method used in this study is still widely applied, but we are aware of its weaknesses, as is the case with analysed organisms having different exchangeable hydrogen properties compared to used standards. This will probably produce some precision and accuracy issues with our data but due to large isotopic separation of the initial sources (aquatic versus terrestrial) we are confident that such uncertainties have not caused any major issues with our model results.

12. Line 498. This assumption is probably fine, but the supplementary data shows that this discrimination factor varied a lot in the measurements. The uncertainty related to this should be commented on.

RESPONSE: *The used value corresponds well with the recalibrated (Soto et al. 2017) photosynthesis hydrogen discrimination values ($-111.1 \pm 17.9\%$) reported in the best available review (Brett et al. 2018). We have now added this clarification also to the Methods section. The variability in the discrimination factor (or the algal source $\delta^2\text{H}$) was used in the MixSIAR model as described in the Methods section.*

13. Extended Fig. 1. There is lack of statistical detail here. On what alpha level were the correlations significant? Was a correction for multiple comparisons carried out?

RESPONSE: *We have now included the used alpha level (0.05) to the figure legend, this was only used to clarify the figure. We did not test any specific hypotheses with this and thus no Bonferroni correction was made.*

14. All figures with trend lines: It is difficult to follow exactly how the curve fitting was done and how the confidence intervals were extracted, and whether or not the relationships represent statistical significance.

RESPONSE: *MixSIAR uses JAGS to run MCMC and it does the covariate fitting (linear correlation) in isometric log-ratio space (ILR space) as it calculates the consumer dietary mixes there. The lines shown in Fig 3 are invILR-transformed (transformation back to p-space) estimates of covariate effect on consumer allochthony retrieved with `plot_continuous_var()` function (in `mixSIAR` package). This back transformation causes the bending of the linear curve. The 95% credibility intervals shown in Fig 3 are derived from the model output as well with `plot_continuous_var()` function, which calculates the 95% CI from the model output by adding the environmental variable effect on consumer allochthony after accounting for the effects of lake and species factors.*

*We have now provided posterior distribution of linear regression coefficient with PC1 and dietary mixtures in ILR space in Fig R1. The linear coefficient of PC1 in ILR space was clearly non-zero (median [CI] = 0.541 [0.344–0.758], Fig. R1). As our model included only two sources, ILR space is one dimensional and thus the non-zero coefficient values indicate that covariate (PC1) had significant effect on consumer allochthony. We have now added this “significance test” to the Methods section. In case the Reviewer 3 is interested, the following script is copied directly from the model definition: ... `ilr.tot[i,src] <- ilr.global[src] + ilr.fac1[Factor.1[i],src] + ilr.fac2[Factor.2[i],src] + ilr.cont1[src]*Cont.1[i] + ilr.ind[i,src]; # add all effects together for each individual (in ilr-space)...`*

Table R1. Recalculations of hydrogen isotope omega (ω) values based on Soto et al. (2017) suggestion to recalibrate $\delta^2\text{H}$ value of consumers due to bias in earlier standard values. The left side of the table (dashed vertical line) summarizes omega values provided in the best available review paper (Brett et al., 2018). On the right side of the dashed line, we have listed the standards used for $\delta^2\text{H}$ analyses of pulverized sample powder. ω calc. method = omega value calculation method where Arithmetic refers to formula: $\omega = (\Delta\delta^2\text{H}_{\text{consumer}}) / (\Delta\delta^2\text{H}_{\text{environmental water}})$ and linear model to linear regression slope in model $\delta^2\text{H}_{\text{consumer}} \sim \delta^2\text{H}_{\text{water}} \cdot \omega_{\text{recalc.}}$. $\omega_{\text{recalc.}}$ is recalculated omega values with the uncorrected data that were available, if multiple samples were available we used mean of the samples. $\omega_{\text{d2H cor. \& recalc.}}$ = corrected omega values with $\delta^2\text{H}$ corrected consumer values. $\Delta\omega$ express the difference between ω values reported by Brett et al. (2018) and the $\delta^2\text{H}$ corrected omega values ($\omega_{\text{d2H cor. \& recalc.}}$).

Study	Organism	Mean	SD	used standards	ω calc. method	$\omega_{\text{recalc.}}$	$\omega_{\text{d2H cor. \& recalc.}}$	$\Delta\omega$
^a M. W. O'Neill et al., unpublished data [†]	trout	0.23	0.03	NA	NA	NA	NA	NA
^b Macko et al. (1983)	marine amphipod	0.12		NA	NA	NA	(0.12)	NA
Solomon et al. (2009)	Daphnia	0.20	0.04	CFS, CHS, BWB	Arithmetic	0.206	0.175*	-0.030*
Solomon et al. (2009)	mosquito larva	0.39	0.04	CFS, CHS, BWB	Arithmetic	0.291	0.248*	-0.043*
Solomon et al. (2009)	salmonid fish	0.12	0.02	CFS, CHS, BWB	linear model	0.129	0.110*	-0.019*
Wang et al. (2009)	chironomids	0.31	0.03	BWB	linear model	NA	0.267	-0.043
Soto et al. (2013)	chironomids	0.47	0.04	CBS, KHS	linear model	NA	0.416	-0.054
Soto et al. (2013)	guppy	0.33	0.003	CBS, KHS	linear model	NA	0.283	-0.047
^c Graham et al. (2014)	Atlantic salmon	0.36	0.05	CBS, KHS	Arithmetic	NA	NA	NA
^c Graham et al. (2014)	Arctic charr	0.35	0.05	CBS, KHS	Arithmetic	NA	NA	NA
^d Hondula and Pace (2014)	estuarine clams	0.15	0.09	CFS, CHS, BWB	NA	NA	NA	NA
^e Newsome et al. (2017)	Nile tilapia ^f	0.24	0.02	Several	NA	NA	0.231 ^g	NA
Group average		0.27	0.11				(0.231 \pm 0.089)	-0.039

^a No access to this study. ^b $\delta^2\text{H}$ analyzed with the similar method as water. ^c This study did not estimate ω with water labelling experiments, thus it was excluded from the recalculations of omega. ^d No water, food or organism values presented and thus this study was excluded from recalculations of omega. ^e Several standards, correct values. ^f Possibly a mean of tilapia muscle and liver omega values. ^g Value for only Nile tilapia muscle. * Values for consumers $\delta^2\text{H}$ are copied from fig. 1-2 (Solomon et al., 2009), also the calculation of omega included some sort of bootstrapping, thus we recalculated the omega values with best we could and the omega value differs from their original report, therefore here $\Delta\omega$ refers to the difference between $\omega_{\text{d2H cor. \& recalc.}}$ and $\omega_{\text{recalc.}}$.

Macko, S. A., M. L. F. Estep, and W. Y. Lee. 1983. Stable hydrogen isotope analysis of food webs on laboratory and field populations of marine amphipods. *Journal of Experimental Marine Biology and Ecology* 72:243–249.

Solomon, C. T., J. J. Cole, R. R. Doucett, M. L. Pace, N. D. Preston, L. E. Smith, and B. C. Weidel. 2009. The influence of environmental water on the hydrogen stable isotope ratio in aquatic consumers. *Oecologia* 161:313–324.

Wang, Y. V., D. M. O'Brien, J. Jenson, D. Francis, and M. J. Wooller. 2009. The influence of diet and water on the stable oxygen and hydrogen isotope composition of Chironomidae (Diptera) with paleoecological implications. *Oecologia* 160:225–233.

Soto, D. X., L. I. Wassenaar, and K. A. Hobson. 2013. Stable hydrogen and oxygen isotopes in aquatic food webs are tracers of diet and provenance. *Functional Ecology* 27:535–543.

Graham, C. T., S. S. C. Harrison, and C. Harrod. 2014. Differences in the contributions of environmental water to the hydrogen stable isotope ratios of cultured Atlantic salmon and Arctic charr tissues. *Hydrobiologia* 721:45–55.

Hondula, K. L., and M. L. Pace. 2014. Macroalgal support of cultured hard clams in a low nitrogen coastal lagoon. *Marine Ecology Progress Series* 498:187–201

Newsome, S. D., N. Wolf, C. J. Bradley, and M. L. Fogel. 2017. Assimilation and isotopic discrimination of hydrogen in tilapia: implications for studying animal diet with $\delta^2\text{H}$. *Ecosphere* 8:e01616.

Table R2. Comparison of models including different continuous covariates. Leave-one-out cross-validation information criterion (LOOic, plus standard error, SE), LOOic difference and the corresponding standard error are resented in the following columns. Relative model support is given as Akaike weights that sum to one across nested models. Mean stretch errors (Multiplicative error term) for hydrogen isotope (standard deviation in parentheses) are presented in the last column (ξ_H). Based on the LOOic and their standard errors, all the models performed equally well and cannot be ruled out.

Model	LOOic	se_LOOic	dLOOic	se_dLOOic	weight	ξ_H (SD)
1 Lake+species+Forest %	2193.2	81.5	0	NA	0.60	0.494 (0.026)
1 Lake+species+POM N:C ratio	2195.3	81.7	2.1	3.5	0.21	0.493 (0.025)
1 Lake+species+PC1	2195.5	81.5	2.3	2.0	0.19	0.495 (0.025)

Fig. R1. Scaled density of linear regression coefficient with PC1 and dietary mixtures in ILR space. Median and 95% Bayesian Credible Intervals (CI) of the coefficient are shown in the figure. Bar width is set to 0.01. The coefficient distribution is clearly non-zero, indicating that PC1 had significant effect on consumer allochthony since with our model configuration ILR space is one dimensional.

Response Letter

We would like to thank the three Reviewers for their contributions to this second round of revisions. We have addressed each comment provided as highlighted below. In addition, we have improved clarity of language throughout the manuscript.

Comments from Reviewers (Text in blue with normal font)

Response from authors (*Text in black with italics*)

Reviewer #1 (Remarks to the Author):

****Comments in response to Response****

I thank the authors for replying to my previous review. They have mostly addressed my concerns, although a few points persist from their Response to Referees.

General comments:

The authors are incorrect about p-values and R2 values. These can entirely be estimated within a Bayesian framework. Many standard packages exist to do this in R (e.g. see `bayes_R2.brmsfit` in the `brms` package or `bayes_R2` package) and I think the authors must know about this if they are using `LOOic` (which comes from the same set of packages). R2 can even be calculated by hand from the `MixSIAR` output as per the equations in Gelman et al., 2019 *American Statistician* <https://doi.org/10.1080/00031305.2018.1549100>. Claiming that the CIs are analogous to R2 values is not really quantitatively comparable.

RESPONSE: *Thank you for the comment. We apologise for our poorly worded response and agree that width of CIs and R2 are not directly quantitatively comparable. Having gone through Gelman et al. (2019), we contest that R2 values can be calculated directly as suggested because the error structure employed here, and in many other studies using MixSIAR, is multiplicative of the process error generated by stochastic sampling of sources, and not an additive residual error term as seems required in the methods given in Gelman et al. (2019). This structure stems from the biological process of tissue turnover and integration, which leads to the expectation that variation arising from variable source isotope compositions are dampened in consumers (Stock et al. 2016). It has been pointed out elsewhere that, beyond identifying potential biases in data, R2 values and related statistical measures explaining variation in isotope compositions of consumers are not necessarily informative when using mixing models (Brown et al. 2018). This is because the main variable of interest, the dietary proportions, are unobserved latent variables being estimated from mixing models, not the isotope compositions of consumers. However, we appreciate that there can be value in estimating the ability of a model to reconstruct isotope compositions of consumers along gradients of other variables, particularly for model comparison. We have therefore estimated an approximate R2 posterior ($1 - SS_{\text{remaining}} / SS_{\text{total}}$) for each model based on the expected consumer value for each posterior draw, which has been added to Supplementary Table 8 (script provided in the GitHub repository).*

*Brown, C.J., Brett, M.T., Adame, M.F. et al. (2018). Quantifying learning in biotracer studies. *Oecologia* 187, 597–608*

*Stock, B.C. and Semmens, B.X. (2016), Unifying error structures in commonly used biotracer mixing models. *Ecology*, 97: 2562-2569.*

More problematically, the authors state the precise problem in their own words: “We want to highlight that currently it is possible to include only one covariate at the time in MixSIAR models” Which makes one wonder, is MixSIAR an appropriate tool? I think it is pretty obvious that the answer is no, MixSIAR is inappropriate. Although isotopic mixing models are the state-of-the-art and most appropriate way to address the specific research questions posed in this study, the authors should really be writing their own models. That said, I think Table R2 and Supplementary Fig. 4 help to address my concerns about the use of multiple coefficients, so perhaps the authors don't need to do anything more here beyond finding a way to report the R² values.

RESPONSE: Yes, we agree that the ideal situation would be to construct from scratch our own coded models, but this is something beyond the skill of the authors. As Reviewer #1 highlights, we are using what is considered the current state-of-the-art for mixing models, but we do agree that with complex data sets, the MixSIAR-framework might not be the optimal choice. Since the return of these comments, we highlight that the next generation of *simmr* mixing models, *cosimmr*, has been released by Govan, Parnell and colleagues (<https://emmaqovan.github.io/cosimmr/>) in May 2024. This new software does allow the implementation of multiple continuous covariates, however it does not yet facilitate the implementation of random effects structures in the data. For our dataset with lake-specific source values, this is important to be able to incorporate. We have, however, now customised and expanded upon the MixSIAR JAGS code to additionally incorporate a random slope effect of species on the single continuous variable (i.e. PC1), particularly in response to Reviewer #2's comment 6, the script of which is now in the GitHub repository. Models with a continuous effect have all now been rerun to include the random slope with species where appropriate. While the main results remain the same, we believe the model structure now better fits the structure of the data. As highlighted above, we have also calculated an approximated R² posterior for our models which is provided in Supplementary Table 8.

Specific comments

Line 401: I'm sorry but perhaps there has been a misunderstanding. My entire point is that PC1 *does not* have a “clear ecological meaning”. It is a non-intuitive composite variable. This revision here also doesn't address the fact that PC1 is kind of a poor predictor (explains only 27.3% of the total variance in the “multiple” parameters). Nonetheless, the new Table R2 helps address this concern and could be included in the Supplementary Information.

RESPONSE: As we used 18 different environmental variables in the PCA, we think that 27.3% of the explained variance in the environmental data set is decent. If we would have selected fewer variables to the PCA, the variance explained would have been trivially higher. In our case the most relevant thing is what kind of gradient PC1 forms, considering the loadings of variables, and how strongly PC1 and consumer allochthony are linked. Here PC1 had clear and intuitive ecological meaning from forested brown water lakes towards more eutrophic lakes and PC1 was clearly linked with consumer allochthony. We think this type of composition variable is a more realistic factor to use in the statistical analyses (when studying spatial gradients) than single environmental factors since PC-axes accounts changes of multiple different variables simultaneously. The suggestion of including Table R2 to supplementary material is rather strange as it was already included to the supplementary material (as Table S8) in the previous submission. We have now updated this table based on the new models (Table S8). With this update we have also included an empty model, a model with species as random factor and a model with species and lake as random factors to the table. Here it is now very clear that fitting

of consumer isotope values by the mixing model is much improved when including the continuous variable.

I also commend the authors in their measured and thorough reply to Reviewer #2, who has misportrayed both models like the MixSIAR approach used by the authors as well as the wider ecology (i.e. comment about consumption of t-POM rather than t-DOM). I also think the Reviewer's insinuation from their comment #6 is inappropriate and commend the authors in handling it so well.

RESPONSE: Thank you.

**** Comments on revised paper ****

The revised paper is improved and I only have some minor points remaining, some of which have emerged from revision:

Line 29: Do you mean "organic" rather than "mineral"? Or "organic and mineral"?

RESPONSE: We have now corrected this to "organic and mineral"

Line 88: Reduce "the number of environmental dimensions" for what purpose? Add a few more words to explain the rationale.

RESPONSE: Thank you for this comment. We have revised this sentence in LINES 92-98

Line 91: What are these numbers in parentheses? How does one interpret them? Is -0.31 large or small? I would consider if these numbers are really necessary to report. I think they might be Pearson correlation coefficients, in which case, interpretation would be straightforward if this point was simply reported.

RESPONSE: Yes, they are correlation coefficients and were provided as another Reviewer previously asked us to show them. We have now added a guide to reading these numbers where they are first presented LINE 100.

Line 139: Why are these "not as conclusive"? Please add some words to explain why this is the case, especially in light of line 143.

RESPONSE: You are correct that this wording was incorrect as the other models were almost as good as the model with PC1. We have now removed the words "not as conclusive". We want to stress that the model with PC1 was selected as the best model based on DIC, Akaike weights, and the approximate R^2 following your suggestion. However, the model with POM N:C performed equally well based on LOOic as the SE of LOOic overlapped with all these models. These points are highlighted in LINES 165-171

Line 144: This is the first time omega has been mentioned so explain what it is.

RESPONSE: Thank you for the comment. Indeed, the I-R-D-M structure required by Nature Communications can easily lead to this kind of issue. We have now added the explanation in LINES 171-172.

Line 145: Can you summarise how these relationships were affected?

RESPONSE: Yes, we can. Higher omega values result in lower allochthony estimates especially in high trophic level organisms due to multiplication effect in omega correction calculations. We have updated the omega sensitivity tests and they are almost identical to the previous ones. These results are summarised briefly in LINES 174-180 and more broadly in LINES 552-565.

Reviewer #2 (Remarks to the Author):

Main points

1. I want to start by thanking the authors for making their raw data available for the review process. I also want to thank them for pointing out that the true distribution of environmental water values may have a mean of 0.23 ± 0.09 , as opposed to the 0.27 ± 0.11 reported in Brett et al. (2018). I have not yet had the time to dig into the various papers by Soto et al. to verify this claim, but I will do that when I get the chance.

RESPONSE: Thank you for the comment. This is an interesting topic. We have now added the revised table presenting omega values to the supplementary material. We hope this might spark a published revision of omega values and more studies of taxon-specific omega values.

2. The authors disclosed an important detail about their analysis that was not previously apparent. In the MS that I first reviewed the authors stated they represented the d2H values of terrestrial resources with the d2H values of DOM in lake inputs. I assumed that they did this because they were using the deuterium values of the terrestrially derived dissolved organic matter (t-DOM) to represent the d2H values of terrestrially derived particulate organic matter (t-POM). But in their response to my comment the authors noted their approach assumed t-DOM was in fact the form of terrestrial organic matter that they hypothesized contributes to zooplankton and fish production in their lakes. This was not clearly explained in the methods of the initial draft that I reviewed (and it is still not explained in the methods section of the most recent draft). The authors also noted that some previous studies also hypothesized that t-DOM was the main source of terrestrial organic matter contributions to consumers in lakes. However, they failed to note that the highly cited paper by Cole et al. 2006 concluded the t-DOM pathway only accounted for ca. 2% of zooplankton production in the lakes they studied. More importantly, Keva et al. did not correctly take into account the impact of a t-DOM to bacteria to protozoa pathway on their deuterium based calculations. If the pathway truly does entail t-DOM to bacteria to protozoa, then the zooplankton in their lakes are three trophic levels above the terrestrial resources, and not only one trophic level as their modeling approach assumes. This has important implications for the assumed contributions from environmental water to zooplankton hydrogen content for this pathway. If you assume that the contribution of environmental water for an increase in one trophic level is 0.23, then the contribution of environmental water after an increase in three trophic levels is 0.54. By the time the terrestrial organic matter makes its way to piscivorous fish (2 trophic levels above zooplankton), the overall contribution of environmental water to the consumer's hydrogen content will be 0.73. Given the considerable uncertainty in current best estimates for δ (i.e., ± 1 SD = 0.09), the environmental water contribution to zooplankton could range from 0.36 to 0.69 for

this pathway, and for piscivorous fish they could range from 0.53 to 0.86. Given this very large range of uncertainty for environmental what contributions it is questionable whether you could get reasonable estimates for zooplankton and quite unlikely that estimates for piscivorous fish would be of any value whatsoever.

RESPONSE: Reviewer #2 is correct that we did not account for the microbial link trophic step in the $\delta^2\text{H}$ correction formulas. The only published way to do this would be correcting the different sources to consumer trophic level (Keva et al. 2022); however, with this setup applying MixSIAR this would be impossible. We do agree that both t-DOM and t-POM can be pathways for terrestrial matter to end up in aquatic food webs as described in the manuscript, and we do not advocate in our manuscript that the utilization of t-OM is wholly of one form or another in lakes. However, the Cole et al. 2006 paper mentioned by Reviewer #2 in support of the t-POM pathway is only one selected article. The same group working with the same two lakes also concluded that bacterial biomass comprised 45-75% allochthonous DOM (Kritzberg et al 2006, Ecosystems). It is true that some researchers in the USA have focused mainly on t-POM, but we would argue that the majority of researchers across Europe and Canada and elsewhere actually consider t-DOM to be the major pathway in consumer allochthony in lakes, and that the wider literature supports this argument. We question the omega assumption that Reviewer #2 has used for the microbes. To the best of our knowledge, there are no omega estimations for bacteria. Fractionation for C and N in microbes appears to be far lower (almost non-existent) compared to fractionation values of more biologically complex higher trophic level organism; thus, omega values for microbes may well also be lower compared to those measured from animals. At present it is simply not possible to resolve this. Moreover, it is correct that we measured t-DOM, which we used to quantify the isotopic composition of t-OM (i.e. both DOM and POM forms). Regardless of the preferential use of DOM vs POM among different trophic guilds within the food web, the use of t-DOM as representative of terrestrial inputs only becomes an issue if the two OM fractions are isotopically distinct from each other, which is not the case when using $\delta^2\text{H}$ (which we now show in supplementary table 10). As we are sure Reviewer #2 is aware, this is one of the main advantages for using $\delta^2\text{H}$ as a terrestrial isotope tracer in aquatic systems over e.g. $\delta^{13}\text{C}$. In conclusion, despite this Reviewer #2's comment, we believe that our results are robust. As mentioned in the previous and current revision round, this type of trophic level or omega modifications of organisms would not modify the slopes but only the intercept of our models. Therefore, we are confident that our main findings, that consumer allochthony broadly declines from brown dystrophic lakes to green eutrophic lakes, hold.

3. There is an even greater conundrum for the t-OM pathway to benthic invertebrates, which the authors have not adequately described. For example, on lines 38-39, they state "Concurrently, t-OM mobilized by bacteria and fungi can support detritivorous zoobenthos and benthivorous fish". But this statement does not indicate whether the pathway is based on dissolved or particulate OM. I contend that at least >> 90% of the aquatic ecology community believes the primary pathway for t-OM to be incorporated into the tissues of benthic invertebrates in streams and lakes is via the classic particulate pathway, i.e., the "peanut butter and crackers" analogy articulated by Cummins (1974). In line 160 of the MS, the authors suggest both particulate and dissolved OM is utilized by benthic invertebrates. In lines 217-218, the authors state "availability of t-OM via a microbial link is likely the key driver of consumer allochthony for both benthic and pelagic food webs in lakes". If the predominant pathway for t-OM to benthos is particulate, then there is still the large problem that the authors do not have an estimate for the $\delta^2\text{H}$ value for the particulate t-OM that is being assimilated by the benthic invertebrates in their lakes. Alternatively, if the prevailing t-OM pathway to benthos is microbial, then the authors have the similar large problem that they have not accounted for the additional trophic

levels between the t-OM and the benthos in the environmental water calculations. Reading the various passages in the MS relevant to this point, I get the impression that Keva et al. mostly believe benthos consume the biofilm that grows on particulate t-OM. For example, on lines 159-160 the authors mentioned that benthos feed “on detritus with its associated microbial communities as well as the surrounding biofilm which can incorporate both particulate and dissolved forms of t-OM”. If this is the case (which I also believe), then the authors need to have measured the $\delta^2\text{H}$ values of the particulate t-OM loaded to their lakes, as well as accounted for the microbial processing and additional trophic levels and incorporation of environmental water. I do not believe this was done in the current MS, or if it was done it was not explained adequately.

RESPONSE: As noted by Reviewer #1, we believe Reviewer #2’s opinion here is partly incorrect. We believe that DOM as a sample better reflects the inputs of terrestrial OM as compared to POM and therefore is our preferred proxy for allochthonous source. DOM represents the leaching of soil and broken-down plant material and therefore covers the full suite of biologically accessible t-OM. Conversely, t-OM in POM sampled from runoff into lakes likely over-represents leaf-litter (pers. obs. based on isotopic comparisons) which typically requires microbial degradation before it can be fully assimilated into animal tissues due to e.g. high lignin content. A further issue with using POM is that it also likely includes algal signatures as well as even the smallest creeks have some algal production and benthic algal suspension to the water column. $\delta^2\text{H}$ value of DOM usually lies between $\delta^2\text{H}$ values of soil and $\delta^2\text{H}$ values of needles+leaves, but $\delta^2\text{H}$ of POM samples can be lower and more variable, as POM from rivers likely includes some algal particles. We agree with the “peanut butter and crackers” analogy and effectively state this in the manuscript. We agree that t-POM and t-DOM can both contribute to consumer allochthony. Our argument is simply that microbes associated with the terrestrial particulate detritus are providing a link for transferring the carbon in that detritus to the benthic consumers, which is analogous to the microbial link that enables transfer of carbon in t-DOM to pelagic zooplankton consumers. We would also highlight that we have indeed measured $\delta^2\text{H}$ values of both t-DOM and t-POM from inlets of the study lakes. The t-POM samples collected in summer showed similar mean $\delta^2\text{H}$ values ($-123.7 \pm 14.3\text{‰}$, $n=35$) compared to those of t-DOM ($-120.5 \pm 8.9\text{‰}$, $n=35$), but the standard deviation was almost twice as large for t-POM, as would be expected due to the more variable composition mentioned above. Overall, we think that DOM from inlets is better reflecting the terrestrial signal of t-OM.

4. In the first round of review, I noted that it is widely known that common herbivorous zooplankton like Daphnia also consume some protozoa and other microzooplankton, which would put them at a trophic position somewhat more than a conventional herbivore, e.g., a trophic position in the food web of 2.1 relative to algae (with a trophic position of 1). In their response, the authors acknowledged my point and stated that in their revised analysis they have dealt with this issue by assigning herbivorous cladocera a trophic position of 2.0 ± 0.1 . This supposed correction is wrong! If you assume that zooplankton have a trophic position of 2.0 ± 0.1 , then you are stating that half the time these zooplankton will have a trophic position < 2.0 , which means they would have to be partially mixotrophic (i.e., capable of some photosynthesis themselves). There is no evidence in the cladocera literature to support such an assumption. Instead, the authors should have assumed that cladocera have a trophic position of 2.1, with an uncertainty of ± 0.1 and with the added caveat that they never have a trophic position below 2.0 because cladocerans, copepods, Chaoborus and benthic macroinvertebrates are not photosynthetic.

RESPONSE: Thank you for this comment. We have now used cladoceran trophic position of 2.1 ± 0.1 in our trophic level estimations of zooplankton, profundal zoobenthos and fish individuals, but the results

hardly changed. We have also added the consumer trophic level plots across PC1 to the Supplementary Figure 7. As observable, there are no clear positive trends between the trophic level and PC1. Thus, it is obvious that the trophic level estimations are not driving the observed decreasing consumer allochthony trend along the studied environmental gradient (PC1).

5. The authors also claimed that the “fundamental difference between the Bayesian and simple linear mixing models: the former gives population dietary probabilities, whereas the later gives individual-level estimates of dietary proportions”. I completely disagree with this claim. Firstly, if you have a single zooplankton sample from a particular lake, you are only going to obtain a diet composition estimate for THAT SAMPLE (not for the “population”).

RESPONSE: While we appreciate the difference of opinion, it seems that Reviewer #2 here misconstrues the implementation of Bayesian mixing models, as has been highlighted by Reviewer #1. The Bayesian implementation of mixing models, structured as in our manuscript, generates the posterior distribution of dietary proportions of the different populations that can reasonably give rise to the observed data, which is assumed to be fixed. This is in stark contrast to the philosophy of frequentist statistical methods, which it seems Reviewer #2 is applying to our approach. While it is true that for some lake-consumer combinations we only have a single replicate, the model still generates the posterior of population, which will incorporate the increased uncertainty due to a low sample size. We would also highlight that due to the crossed random effects structure of the populations we have implemented here, population posteriors are also informed by those from other lakes for the same species, and other species from the same lake. This results in shrinkage to the central tendency for populations with low sample sizes as those levels have less data to update their dietary proportions. These are only some of the many advantages that such Bayesian mixing models bring to analysing compositional data, far above and beyond what can be achieved using simple mass balancing linear equations that were the standard approach several decades ago.

To get an estimate for the actual population, you would have to collect multiple samples of the same taxa at multiple representative locations on multiple representative dates. The data shown in Supplementary Table 4 show the vast majority of zooplankton and benthos samples were represented by samples of 1 or 2 per lake (presumably collected on the same date within a lake). The fish samples were usually 5 per lake (again, presumably collected on the same date within each lake). With this very low level of sampling, it is inconceivable that one could obtain anything approaching “population dietary probabilities”.

RESPONSE: As pointed out above, this is simply not true, with the low sample size being inherently incorporated into Bayesian implementations as model priors are updated, with fewer data replicates resulting in less informed posteriors. This has been demonstrated in Bayesian mixing models by Brown et al. (2018), of which Reviewer #2 is a co-author. More recently, Heikkinen et al. (2022) concluded that small sample sizes in Bayesian dietary estimations are not that problematic. In fact, when there are small sample sizes a Bayesian framework performs much better than a frequentist approaches.

Brown, C.J., Brett, M.T., Adame, M.F. et al. 2018. Quantifying learning in biotracer studies. *Oecologia* 187, 597–608

Heikkinen, R., Hämäläinen, H., Kiljunen, M., Kärkkäinen, S., Schilder, J., & Jones, R. I. (2022). A Bayesian stable isotope mixing model for coping with multiple isotopes, multiple trophic steps and small sample sizes. *Methods in Ecology and Evolution*, 13(11), 2586-2602.

I recently sampled a lake for Daphnia and Chironomids at ten sampling stations each date over a five month period. We used a fatty acid based dietary mixing model to estimate their diets. This study showed diet composition was similar between stations within a date, but vastly different from one month to another. Both Daphnia and Chironomids were strongly reliant on diatoms in the spring and then showed a sharp transition to much lower absolute fatty acid content and reliance on a mix of cyanobacteria, diatoms and cryptophytes during the summer bloom period.

RESPONSE: As Reviewer #2 has not provided this data, it is difficult for us to comment on any specifics. However, we would point out that the use of fatty acids (FAs) as tracers of biomass assimilation from different production sources is highly problematic, far more so than the use of bulk stable isotopes. FAs trace a small suite of biomolecules synthesised by various diverse clades that can often be highly specific. Physiological modifications to these FAs during and post assimilation are plentiful, including saturation, shortening etc., the specific metabolisms of which are broadly unknown. Yet quantifying contributions from different FA sources requires these species-specific modifications to be known, which likely vary within individuals due to seasonal differences in energy mobilization physiology, and between individuals due to e.g. carry-over effects. It is not surprising that FA profiles in consumers vary of a five-month period that follows the successional progression of microalgal blooms. However, we would point out that successional changes in microalgal blooms, in and of themselves, are unlikely to impact the degree of allochthony in consumers. As our sampling was conducted towards the end of summer, our samples will be incorporating signals from the majority of the bloom cycle. Additionally, we are aware of the FA dietary libraries that some researchers use to trace the origin of the FAs, but these libraries are typically solely based on monoculture growth experiments with one or two temperature modifications. Thus, there are no fractionation values derived from the natural environment nor distributions for these individual FA fractionation estimates. What is even more concerning is that these libraries are not openly published. Therefore, dietary studies based on solely FAs are not convincing when inferring total assimilation within consumers. Bulk $\delta^2\text{H}$ isotope ratios however are more robust against such short-term fluctuations and differential fractionations as they are far less sensitive to compositional variations within the suites of biomolecules upon which consumers depend. Moreover, as a bulk tracer, $\delta^2\text{H}$ values correspond to assimilation across the breadth of the suites of biomolecules, whereas FAs constitute only one albeit an important part of production transfer.

A Bayesian approach will give you fancy plots with distributions for the dietary estimates, but these distributions just represent the uncertainty inherent in the input parameters. They are not in any way representative of the true uncertainty within the population, which as already noted could only be ascertained with a MUCH more rigorous sampling design. For example, based on the information presented in Supplementary Table 4 this could only be done for some of the zooplankton and benthos taxa collected from Jyväsjärvi and Sääksjärvi.

RESPONSE: The incorporation of known uncertainties associated with different parameters necessary to evaluate mixture compositions should always be incorporated into models. However, this does not mean that the breadth of posterior distributions directly correlates to the uncertainty of input parameters in all instances as Reviewer #2 is suggesting. Much depends on data geometry, model structure, and how informative the data are (which again does not necessarily correspond to sample size). Regardless, the posteriors produce the distribution of population-level values that could reasonably give rise to the sample data in the context of these known uncertainties. Authors that exclude known uncertainties (such as variability in source isotope ratios) conversely will present an artificially deflated and false view of certainty in their estimates. This is why Bayesian approaches have

become the standard approach over the past 20 years as compared to simple linear balancing which cannot incorporate uncertainties directly nor adequately represent the rich, complex, but necessary model structures reflective of ecosystems and sampling designs.

Also, if you want to quantify uncertainty associated with the input parameters for only one or two samples, this can also be easily done using classic mass balance calculations with a Monte Carlo approach. But I do not think this would be worth the trouble since the calculated uncertainty would only pertain to those very specific samples.

RESPONSE: Yet knowing the uncertainty in the dietary proportions for each individual sample using a simple linear balancing approach would be necessary as, presumably, it is those estimates which will be used to test for associations in diet with other parameters, such as differences between lakes, species, or environmental gradients. That uncertainty should always be carried forward to avoid misleading readers as to the strength of support the data provide for reported associations. Suitably running Monte Carlo simulations across a dataset of over 1700 individual consumers is unreasonable, computationally expensive, inefficient, and, more importantly, unnecessary when Bayesian mixing models allow users to define and test for rich structuring within the dataset directly whilst simultaneously incorporating known uncertainties.

You could also, relatively easily, use a classic mass balance and Monte Carlo approach to estimate the diet of the population, provided you had proper population level data. The main difference between the classic mass balance based approach and the Bayesian approach is that the mass balance approach is solely based on the Law of Mass Conservation, one of the foundational principles of science.

RESPONSE: Bayesian mixing models are based on the Law of Mass Conservation and indeed the very same linear balance equations as used in the classical mass balancing approach. There are a plethora of review papers (e.g. Phillips 2012) highlighting the development of the more sophisticated Bayesian mixing models from this foundational principle.

Phillips, D. L. (2012). Converting isotope values to diet composition: the use of mixing models. *Journal of Mammalogy*, 93(2), 342-352.

Additionally, the solution to a classic mass balance calculation will exactly match the observed consumer's $\delta^2\text{H}$ value. This will often not be the case for Bayesian results, which can be very different from the observed stable isotope values.

RESPONSE: The aim of most (reasonable) Bayesian mixing models is not to exactly reconstruct every individual datapoint, but to holistically analyse the dataset to parameterize the coefficients of the defined model structure, providing posterior estimates that may reasonably generate the observed data. This provides insight into underlying processes or associations that are influencing underlying diet as expressed through changing isotopic compositions of consumers. Users can then draw inference from these parameter estimates. It should be obvious that with the incorporation of any degree of uncertainty, one would not be able to exactly predict all datapoints with a single model. However, if the aim is truly to exactly predict consumer values (X_i) then one could construct a Bayesian mixing model with the structure $P_{ik} \sim 1 + (1|X_i)$. However, there would be no wider inference that could be gained from such a model, and hence why such models are not implemented as the aim is not to optimally predict measured consumer isotope values of a specific dataset. The

same is likewise true with individual mass balance calculations applied individually to every consumer.

Conversely, the MixSIAR Bayesian algorithm is extremely “malleable”, and you can tinker with it in ways that can strongly influence model outcomes. That is, it is susceptible to what is generally known as data-torturing aka ‘making the data fit the hypothesis’ (Mills 1994).

RESPONSE: *The MixSIAR algorithm (i.e. the execution of the Bayesian model software) is not extremely malleable as Reviewer #2 suggests. Presumably he is referring to different inputs that need to be supplied, namely the source data, trophic discrimination data, error structure and model structure. The source and discrimination data of course need to be adequately provisioned and justified, which is readily highlighted by experts in the field in various reviews and papers stating that the quality of mixing model results can only be as good as the quality of data used as inputs (e.g. Phillips et al. 2014). Likewise, the selection of the error and model structures provided by the users need to be justified and suitable for testing specific aims and hypotheses, which is true of all statistical analyses, not just mixing models. Because of the nature of complex mixing systems, it can be more difficult to directly assess whether model assumptions have been validated, requiring various checks and quality assurance practices, to which Reviewer #2 himself has contributed (Brett 2014). However, we sincerely hope that Reviewer #2 is not suggesting that, by collecting and analysing thousands of samples over three years, applying the best analytical data corrections according to current knowledge, including the use of other measured data to estimate consumer TPs, and constructing a mixing model that tests for the effect of a well-reasoned environmental gradient while concurrently accounting for the complex data structure, we are merely “making our data fit the hypothesis”.*

Brett, M. T. (2014). Resource polygon geometry predicts Bayesian stable isotope mixing model bias. *Marine Ecology Progress Series*, 514, 1-12.

Phillips, D. L., Inger, R., Bearhop, S., Jackson, A. L., Moore, J. W., Parnell, A. C., ... & Ward, E. J. (2014). Best practices for use of stable isotope mixing models in food-web studies. *Canadian Journal of Zoology*, 92(10), 823-835.

The biggest problem with the Bayesian estimates is that they always constrain the estimated contributions to the 0-1 domain, when the raw data may indicate different outcomes. In the case of classic mass balance analyses, it simply is not possible to tinker with the algorithm. One set of data (and an assumed environmental water value) will give one exact answer.

RESPONSE: *As mentioned above, users do not tinker with the algorithm in MixSIAR, but may specify different model structures to test different hypotheses. However, Reviewer #2 raises an important point that, fundamentally, Bayesian mixing models constrain contributions to the domain of 0–1. This constraint is indeed necessary and a fundamental principle in the Law of Mass Conservation invoked earlier. Values falling outside this domain are biologically impossible, and when they occur when classical mixing equations are applied, demonstrate that the mixing system is inadequately described and therefore invalid, which should lead the practitioner to consider what aspects of their system of interest are poorly resolved rather than accepting those results as reasonable. As for proving only one exact answer for a given dataset and assumed environmental water value, the presumption of this stance is that there is perfect knowledge of the study system with no variation or uncertainty, which is clearly unrealistic in real world settings. As highlighted several times, the inclusion of known (or even estimated) uncertainties are one of the key advantages of the application of Bayesian mixing model approaches.*

The most worrisome concern is that the outcomes provided by the Bayesian mixing model may be more a function of the attributes of the algorithm than the information contained in the raw data. For example, on lines 50-51 of the MS, Keva et al. state “a recent study of 147 lakes indicated zooplankton to rely on average 11% to 83% on t-OM sources (Tanentzap et al. 2017)”. Because I have all the original raw data from Tanentzap et al. 2017, which I have already reanalyzed (see Brett et al. 2018), I can compare zooplankton allochthony estimates from their Bayesian analysis to a direct mass balance analysis using exactly the same data. My reanalysis of the zooplankton d2H based data (n = 439) from Tanentzap et al. (2017) gave a median zooplankton allochthony estimate of 5% when using an ω value of 0.23. The 95% percentile range for these data was -123% to +55% allochthony. Overall, 44% of the zooplankton cases from the Tanentzap et al. (2017) dataset indicated negative allochthony when using d2H as the dietary tracer. While 44% negative estimates may seem paradoxical, if the true value was zero and the raw data used had some estimation, observational and measurement error, then 50% of the estimates should be negative. We know the classic mass balance analysis of these data is correct because we can use the individual zooplankton allochthony estimates and resource and lake water d2H values to perfectly reconstruct the original zooplankton raw d2H values. [This of course does not mean that the underlining allochthony estimates are absolutely correct, because these estimates will be no better than the data they are based on, and these data likely have substantial estimation error]. This begs the question, why does the correct mass balance based analysis of these exact data indicate negligible allochthony, when the Bayesian results indicate considerable allochthony (i.e., Tenentzap et al. reported a median estimate of 42%)?

RESPONSE: *The concern of Bayesian mixing model output not being informed by the data (first sentence) is a fair point and one that is generally overlooked in mixing models. However, the degree to which posteriors have been informed by the data can easily be estimated using e.g. Kullbeck-Leiber divergence (Brown et al. 2018). If data are completely uninformative, then the posterior distributions will be the same as the supplied priors, which in our case are proper flat priors as we have only two isotopic end-members, with a mean of 50%. Our posteriors demonstrate this is not the case (fig. 3).*

As we do not have access to the Tanentzap et al. 2017 data, it is difficult for us to provide specific comments. However, the main argument Reviewer #2 seems to be making is that the mass-balance estimates are correct (de facto) and because they provide different proportions to those estimated using a Bayesian mixing model, Bayesian mixing models are wrong. As we highlighted to Reviewer #1, all mixing model approaches, including classical mass balance, aim to quantify compositions or proportions that are latent variables, that is they are unobservable quantities and therefore unknown. It is therefore infeasible to verify and validate whether one modelling approach or another provides the ‘correct’ answer, especially in biological systems. Even under strict laboratory feeding experiments, uncertainties in digestibility, assimilation rates, tissue routing, physiological processing, wider metabolism, and their interactions with environmental variables means it is difficult to verify as estimates of proportional contributions to tissue formation need not and likely do not directly correspond to ingested diet composition. The development of mixing models over the past twenty years has therefore focused on incorporating natural uncertainties, complex data structures and biological realism to better reflect real world processing with strong justifications.

Coming back to the specific example of Reviewer #2, he points out that 44% of the proportions calculated using classical mass balance provide biologically nonsensical results (i.e. contributions outside the domain of 0–1, in this case negative values). Arguing that these negative values are true zeroes and should be treated as such is a very blunt instrument, and moreover points to a modelling approach that cannot reasonably describe the system, indicating inadequacies such as in the described source data, unknown and unaccounted fractionations, omega values that are not reflective of the

system of interest or other processes and uncertainties not included in the model. As mentioned above, Reviewer #2's underlying argument here and throughout this large comment is that the simple mass balance calculation is de facto correct, which we highlight is impossible to verify as the proportions are an unobservable variable. Reviewer #2 however argues this case because simple mass balance can perfectly reconstruct the observed $\delta^2\text{H}$ value of a consumer. In the very simple system of two perfectly defined sources (i.e. have no variability) and a single tracer, i.e. a fully determined system with no uncertainty, simple mass balance will always provide a perfect reconstruction because the algebraic function that estimates the diet proportions in such a system is injective, i.e. has one to one mapping of distinct elements in its domain (the consumer isotope values) to distinct elements of the codomain (the proportions of the two sources which are perfectly codependent). There is therefore a circular argument in this logic because the perfect back transformation of a specific dietary proportion is purely a mathematic artefact of the simple linear function. In fact, taking the argument of Reviewer #2 that all negative proportions are true zeros, in other words the codomain is bounded to biological feasibility, then you can no longer perfectly reconstruct all consumer isotope data as the inverse function is no longer injective – the value of 0% allochthony in their example would now map to many observed consumer $\delta^2\text{H}$ values. While Reviewer #2 suggests that this can be explained by the “data likely [having] substantial estimation error”, in this case negative proportions of over -100% as they estimate would suggest isotope values of some of the consumers being at least more negative of the aquatic source than the isotopic difference between the two sources! If the “estimation, observational and measurement error” of these data would be so great as to alone cause such discrepancies, then clearly the data are not suitable to be analysed in such an approach.

Reflecting on this comment more broadly, we feel that Reviewer #2 has considerable prejudice against the Bayesian mixing model framework we have implemented to analyse this considerable dataset, which, as noted by other Reviewers, to date may be the most extensive hydrogen isotope data set sampled in a small spatial scale by one research group using consistent protocols. The arguments put forward in this comment and elsewhere do not, in our opinion, stand up to rigorous scrutiny which we have point-by-point highlighted. Reviewer #2's own paper which they point to (Brett et al. 2018) in fact highlights considerable uncertainty in fractionations and proportion of environmental water contributions to consumer tissues which should be incorporated into any model seeking to disentangle source contributions. The Bayesian mixing model frameworks implemented here and in the 100's of studies elsewhere over the past decade do just that, providing valuable insights for the wider fields of ecology, archaeology, and others where the analysis of compositional data is important. For clarity, acknowledging this uncertainty does not invalidate our previous discussion in our previous response that omega values in Brett et al. (2018) are incorrect, the focus of which is omega value bias rather than uncertainty.

I also want to note that Keva et al. did not present model corroboration results! As noted previously, a proper mass balance will yield perfect corroboration results. For example, I was able to reconstruct the observed zooplankton d2H values from Tanentzap et al.'s raw data using the equation below:

$$(\delta^2\text{H}_{\text{terr}} * \theta + \delta^2\text{H}_{\text{phyto}} * (1 - \theta)) * (1 - (1 - (1 - \omega) \text{TL})) + (1 - (1 - \omega)) * \delta^2\text{H}_{\text{H}_2\text{O}} = \delta^2\text{H}_{\text{zoop}},$$

where θ represents the estimated terrestrial contribution, ω represents the assumed hydrogen contribution from environmental water, TL represents the number of trophic levels above primary producers, and $\delta^2\text{H}$ represents the hydrogen stable isotope values for terrestrial and aquatic organic matter, lake water and zooplankton.

RESPONSE: Thank you for the comment. We provided many corroboration statistics for each of the models in the previous revision. However, we have now calculated also pseudo R^2 values from the predicted tracer values for each of the models according to Gelmann (2018) and Stock et al. (2017). The results can be found in Table S8, including also other type of corroboration statistics. As highlighted in above however, one-to-one back calculations using simple linear equations with no uncertainty does not corroborate results as this behaviour is characteristic of injective functions.

6. I have read the original and revised MS several times and I am still very confused as to how the authors represented the algal d2H values in their Bayesian model. I asked this question in the previous round of review and it still has not been clearly explained in the response and more importantly in the MS itself. For example, in their response the authors stated “we have sampled benthic algae from each study lake. Thus, we took the average discrimination factor from the analysed samples and used that with the lake water to model algae δ^2H values. . . . We do not fully understand why Reviewer 2 argues that we have used little field data for the variables of interest, because that is evidently not the case in our study.” This response does not make sense and it is self-contradictory. Firstly, if as the MS states benthic algae were sampled multiple locations in each lake (see lines 302-305), why weren't the actual observed algal d2H values used to represent the autochthonous resources in the dietary mixing model? If you have access to real data, why would you use “modeled” data instead. For the raw data shared by the authors, it appears that only one algal d2H value was used for nearly all of the lakes, and this algal value was almost always -106.7 d2H different from the lake water value. On lines 403-405 of the MS, the authors state “observed spatiotemporal standard deviation [was] $\pm 23.6\%$ ”. However, the variation among all lakes for the algal d2H values actually used in the Bayesian model was only $\pm 8\%$. From the outside looking in, it appears that the authors have used a quite convoluted approach to greatly reduce the lake to lake variability in the d2H values for algae. In addition, this approach results in a situation where the algal and lake water d2H values are nearly perfectly aligned ($r^2 = 0.99$). This example is exactly why I am concerned that the authors are not using real data when apparently it exists. Does their decision influence the outcome of the Bayesian mixing model? Without seeing the observed algal d2H values from each lake, it is very difficult to understand why the authors used “modeled” data when they could have used directly observed data, this seems suspect. To reiterate my previous point, the dietary mixing model depends on five variables (i.e., the d2H values for t-OM, algal, water and zooplankton and the assumption for environmental water). On these five terms, only the water and zooplankton d2H values are based on actual observations.

RESPONSE: This issue has been discussed already in the manuscript and previous revision rounds: “For most lakes, it was impossible to obtain sufficient samples of pure phytoplankton for SIA. Therefore, the phytoplankton δ^2H values were modelled from lake water δ^2H values with a photosynthetic discrimination estimate of -106.7% derived from the mean difference between measured benthic algae and water δ^2H values across the sampling sites”. The claim that only one algal value was used is incorrect, because we have used benthic algal values from many lakes (see e.g. Supplementary table 3). Table S3 shows only a subset of all available benthic algal data, but still includes four lakes which is enough to discount Reviewer #2's point. We applied the mean benthic algal fractionation to the measured water values from individual lakes and these values were used to model phytoplankton δ^2H for each lake. We already discussed in the last revision round and in the manuscript that our estimation of phytoplankton δ^2H fraction from water δ^2H values is very similar to those reported in many previous studies, and that this has been a standard approach in the field (see for example the excellent studies by Karlsson et al.). Hence, there is obviously a misunderstanding from Reviewer #2's side – he missed that we have used d2H fractionation from the individual lake water samples to model the algal values. He also missed the reason why we did this. It was written in the manuscript “it was impossible to obtain sufficient samples of pure phytoplankton for SIA”. Once more we try to reason this to Reviewer #2:

Collection of relatively pure phytoplankton was logistically infeasible, particularly in non-oligotrophic lakes where POM samples will inevitably incorporate substantial amounts of t-POM, zooplankton and their faecal materials etc. Hence, we opted to use benthic algae sampling to represent the autochthonous lake production. The benthic algal samples we collected likely also include some variable amounts of non-algal “impurities”, although to a much lesser degree than POM, and some were missing from a couple of lakes from which no samples could be collected. Hence why we used the average fractionation from across all measured lakes and incorporated the fractionation variability into our modelling. This was the reason to use modelled algal $\delta^2\text{H}$ values. The water-algae fractionation values we used are well in line with previous studies (one written by Reviewer #2). We would like to point out that Bayesian framework does not use a single point source isotope value; Reviewer #2 can easily find the used distributions from the manuscript or the GitHub repository. We would like to also point out that we did measure DOC $\delta^2\text{H}$ values from the inlets of each lake (data provided in the repository), so the last claim is not true.

*e.g. Karlsson, J. et al. Terrestrial organic matter support of lake food webs: Evidence from lake metabolism and stable hydrogen isotopes of consumers. *Limnol. Oceanogr.* 57, 1042–1048 (2012).*

7. I looked at the raw data the authors posted for this submission. I am concerned by the calculated Trophic Level values for a few of the consumers. For example, the average trophic level for Asellus was 1.59. This means the authors are inferring that Asellus is 41% phototrophic and 59% heterotrophic. In > 80% of the cases, the calculated trophic level was < 2. 10% of the Asellus samples had estimated trophic positions below plants (i.e., < 1). A trophic position characteristic of a mixotroph could be true for a tropical coral which hosts zooxanthellae, but a trophic level < 2 cannot be true for any of the invertebrates sampled from Keva et al.'s lakes. I realize the authors used a superficially reasonable approach to calculate the consumer trophic levels, but this approach has considerable uncertainty ($3.4 \pm 1.0\%$) and it produced unreasonable results in many cases. The 1.59 average trophic level also means the average environmental water contribution assumed for Asellus is only 0.14. I also note that this is not inconsequential because the overall allochthony estimate is dependent on the trophic level assumption used in the mass balance analysis (see plot below where I plotted the estimated TL against the estimated allochthony for Keva et al.'s Asellus raw data). Because the environmental water assumption is directly dependent on the assumed trophic level, the environmental water assumption is similarly correlated with the estimated Asellus allochthony. There is a similar issue with the estimated trophic position for littoral Chironomids and to a lesser extent littoral benthic macroinvertebrates, which both have average trophic levels < 2. I am also skeptical of the estimated trophic position for pike, which is slightly less than that for smelt. Since Keva et al.'s stomach content data indicate the pike they sampled had exclusively consumed fish, and the fish that pike would likely to have consumed had an average trophic level of ≈ 3.5 , I believe that a trophic level of 4.5 is much more reasonable for pike. It appears that the authors estimated the trophic level of all consumers relative to the 15N values of Cladocera, but the authors do not explain how they estimated the trophic position of consumers in lakes where a cladoceran samples were not collected.

RESPONSE: Thank you for the comment. This is a fair and valid point. Of course, Asellus cannot have a TL <1! This arose as the Reviewer #2 suspected from our use of cladoceran $\delta^{15}\text{N}$ values as a reference for calculating all consumer TLs. We have now updated the littoral zoobenthos trophic level models by using littoral Asellus $\delta^{15}\text{N}$ as reference for littoral zoobenthos groups, so using an analogous approach for the littoral benthic food web as used for the pelagic food web compartment. With these modifications the trophic level of zoobenthos groups increased approximately 0.5 units and we trust the trophic level estimations are now more realistic. Please find the updated trophic levels in Fig. 2 or

from the raw data. We used these updated trophic level estimations to omega correct the $\delta^2\text{H}$ values and further in the subsequent mixing models. We do agree that the trophic level of smelt appears high (this has actually been reported from other studies), but this does not make the trophic position of pike wrong. The probable reason for the high smelt trophic level is briefly discussed in the manuscript, but Reviewer #2 may have missed this. We feel the pike trophic level of 4.0 ± 0.3 is entirely reasonable. As discussed already in the previous revision round, these types of systematic changes in the trophic position or omega values do not impact the allochthony trends across the studied environmental gradient as they modify mainly the intercept of the models. The last point of this comment 7 is completely incorrect; we explain in the manuscript: "In four lakes (Hirvijärvi, Niemisjärvi, Pankajärvi and Valkea-Kotinen), the cladoceran $\delta^{15}\text{N}$ values were lacking and thus they were derived from a linear model based on $\delta^{15}\text{N}$ values of cladoceran and bulk zooplankton samples from the other study lakes (adjusted $R^2 = 0.928$, $p < 0.0001$, $\delta^{15}\text{N}_{\text{cladocera}} = 0.872 \times \delta^{15}\text{N}_{\text{bulk zooplankton}} - 0.095$)." which Reviewer #2 may have missed while reviewing our manuscript.

8. If you look at the main plots in the Keva et al. MS, there are some strange trends that concern me. For example, in Fig. 3 the allochthony estimates for Asellus in panel b2 appear to be very similar to the estimates for littoral bulk zoobenthos in panel b1 and profundal zoobenthos in panel b4. Additionally, the results for ruffe in panel c5 appear to be very similar to the results of perch > 15 cm in c1. These similarities in the allochthony estimates for different consumers strongly suggests that the estimates depicted in these plots are mainly a function of the structure of the Bayesian mixing model and its underlying assumptions, rather than the consumer raw $\delta^2\text{H}$ values.

RESPONSE: We partly disagree with Reviewer #2 here. One really can see the allochthony trends even from the raw data (Fig 2), such as approximate co-alignment of Asellus and bulk zoobenthos samples in between source end members for different lakes (samples from the same lake will fall on the same vertical line between the two plots). Further, given the functional similarities of Asellus and bulk zoobenthos, and perch with ruffe, it should not be surprising that their estimated terrestrial contributions are similar on a lake-by-lake basis. We accept however that the model structure of our mixing models could be improved upon compared to the last revision round, which was also highlighted by Reviewer #1. As presented, the model structure then only estimated the global trend across the environmental gradient (PC1), and was therefore the same for all species (although species and lakes both had random intercepts). We have now extended the model structure of our mixing models by manually customizing the JAGS script, such that it now also estimates random slopes with PC1 for different consumer taxa. We have now rerun all the analyses and redrawn all the result figures. We trust our results are now more realistic as different species have separate slopes. However, we would also point out that the overall results of rates of ecosystem-wide changes in allochthony with PC1 did not change much.

9. Finally, perhaps the most important concern is that when using the raw data Keva et al. provided for this review, I was unable to re-create the main trends these authors reported for Asellus in their Fig. 3 panel b2 and Fig. 4S panel b2. In their response to my original comments, these authors stated that the actual allochthony estimates are not nearly as important as the trends they observed between the estimated consumer allochthony and various landcover measures. I used the raw data they provided for the review process to calculate Asellus allochthony using the classic balance approach. I then compared these estimates to the PC1 score and the % Forest land cover data for each of the lakes where Asellus data were available. This comparison showed no relationship between the two land

cover characterizations and the estimated allochthony, in stark contrast to the key results from the Keva et al. MS. This discrepancy must be resolved. To help with this, I have provided my allochthony calculations for the Asellus data. So far, I have only attempted to re-create the reported trends for Asellus and landcover, but if I am unable to re-create these trends for Asellus, I strongly suspect the same will be true for the other consumers.

RESPONSE: We thank Reviewer #2 for examining our data in detail. However, we would highlight several reservations regarding his approach. Firstly, as explained extensively in our response to comment 5, simple mass balance calculations do not account for the considerable variation and uncertainty in the data and therefore is not carried forward in their analyses, nor does it limit estimates to the strict domain of 0–1. Secondly, simple ordinary least squares linear regression is not a suitable statistical approach to analysing proportional data as residual errors are not normally distributed and the regression domain does not match the known data domain; therefore, derived coefficients from the Reviewer #2’s analysis of their derived proportion estimates should be treated with caution. Thirdly, briefly looking through the provided datafile, we note some source values for lakes are incorrect due to errors presumably in transferring the data and working in Excel (e.g. lakes Sääksjärvi and Sääskjärvi have been mixed and combined), which will result in inaccuracies. Fourthly, looking at the raw data alone in Fig.2 would suggest that Asellus, of all 19 taxa groups samples, would be the most likely to produce a flat trend with PC1. It is circumspect therefore that the reviewer sought only to analyse this specific group, rather than the many others that show decreasing trends in the raw data, but to subsequently cast dispersions upon potential trends in the other taxonomic groups. Fifthly, the approach taken does not in fact aim to reproduce our results – our model parameterized a single, global trend with PC1 averaging over all taxonomic groups that could best reproduce the observed trends in $\delta^2\text{H}$ data across 19 taxa covering the breadth of functional guilds that occur in lentic ecosystems sampled from 35 lakes. Because the Reviewer #2 sought only to analyse Asellus, he has not attempted to reproduce our results, nor does it seem he has even attempted to incorporate the nested sampling design within his analysis.

However, we appreciate that the specific model structure implemented in our paper may have been misconstrued given the plot layout in Fig.3 and that a more complex model structure that incorporates random slopes by taxa may better reflect ecosystem processes following comments from reviewer one. We have therefore modified our model structure by directly customizing the JAGS code to now include random slope effects by taxa in our new Fig.3, and indeed the trend for Asellus is now flatter. The posteriors of slopes for each consumer group are provided in the supplementary Fig.6 which demonstrates that all other groups have positive posterior mean slopes in ILR space, translating to a decreasing trend with PC1 (although the 95% credible intervals of bulk zoobenthos, copepods, and smelt 95% credible intervals bound zero), and disproving the Reviewer #2’s suspicions regarding trends for other consumers in our dataset. More broadly, the global trend with PC1 has remained near identical to that of the previous submission, and we trust that Reviewer #2 is now more accepting of the patterns observed in our data.

Minor point

10. In their response to one of my comments, the authors state that “the equations in Brett et al. (2018) assume the trophic level difference between zooplankton and sources to be exactly 1”. This is incorrect, Brett et al (2018) used the trophic levels indicated in the original raw data from Tanentzap et al. (2017). Tanentzap et al. (2017) assumed various cladocera had a trophic position of 1 (above algae), calanoid cyclopoids had trophic positions of 1.25 and 1.5, respectively, and Chaoborus had a

trophic position of 2.0. Mixed zooplankton samples had intermediate trophic positions. The environmental water contributions were all corrected for trophic positions as described by Solomon et al. (2009), although the Brett et al. (2018) paper mistakenly did not mention that detail.

RESPONSE: Thank you for the clarification

Conclusion

11. If the discrepancy between my estimated Asellus allochthony and the PC1 and Percent Forest landcover characterizations (compared to what was reported in the MS) cannot be resolved, this study should not be published in Nature Communications or any other journal. I am unable to replicate the core findings of the Keva et al. MS using their own raw data, which calls into question the entire premise of their analysis. My overarching general concern with this paper is that the results appear to be mainly a function of the structure and assumptions imbedded into the Bayesian mixing model rather than attributes of the underlying raw data. Additionally, in some cases “modeled data” are used rather than available sampled data (e.g., the algal d2H values). This creates a direct correspondence between the water and algal d2H values used in the modeling, which would not be the case with real data where the average uncertainty when compared primary producer to water d2H values is large $\pm 27\%$. The t-OM d2H values used in the Bayesian modeling are highly problematic because these values represent 2 trophic levels below what the zooplankton or benthos are consuming. For example, because it is envisioned that zooplankton obtain their t-OM along a bacteria to protozoan pathway, the zooplankton are 3 trophic levels above the t-OM but 1.1 trophic levels above algae which they directly consume. This means the environmental water contributions along the t-OM pathway will be much larger than along the algal pathway, (e.g., 0.54 vs 0.23, respectively). The trophic pathway for t-OM to benthos is even more muddled because the authors do not use d2H values for particulate t-OM in their modeling and they do not account for the larger contribution of environmental water along this pathway.

Michael T. Brett

Mills JL. 1993 Data torturing. N. Engl. J. Med. 329, 1196–1199.

Cummins, K. W. 1974. Structure and function of stream ecosystems. BioScience 24:631-641.

Solomon, C.T., J.J. Cole, R.R. Doucett, M.L. Pace, N.D. Preston, L.E. Smith, and B.C. Weidel. 2009.

The influence of environmental water on the hydrogen stable isotope ratio in aquatic consumers. Oecologia 161:313–324

RESPONSE: Thank you for the extensive comments on our manuscript. As we responded in comment 9 for several reasons, the analyses you performed on a small subset of the data do not aim to reproduce our results nor the main finding of changes in relative terrestrial contributions to lentic ecosystems along the environmental gradient. To question the entire premise of our analysis based on this is therefore unfounded. However, we have updated the model structure to better reflect such concerns that a single global trend along the environmental gradient may not be adequate when analysing 19 functionally distinct taxonomic groups, but we stress that the main inferences have remained the same regardless. The Reviewer #2's disdain for Bayesian mixing models has been apparent throughout, but we have found no reasonable argument against these statistical frameworks presented by the reviewer that stands up to scrutiny. We presume therefore that this stems from Reviewer #2's misunderstanding of such statistical frameworks and his preference for gross simplification of complex biological

processes. Indeed, the Reviewer #2 has not suggested a reasonable alternative modelling framework that would seek to test the same overall hypothesis as we present. We presume from his perspective however, that he would seek to calculate individual mass balances for each single data point, then use simple linear regressions to model each individual taxa in turn, which would substantially limit any inferences that could be drawn from the dataset considered as a whole. Further, to incorporate the variations in sources, and other uncertainties in e.g. trophic fractionations, the bootstrapping that would be necessary to cover over 1700 datapoints and the likely 1000's of replicative linear models involved would be, to say the least, unwieldy. Given that we have implemented several differing model structures in a Bayesian framework that all result in similar inferences being drawn, this should be evidence enough that it is not the model structure driving the results we present, but results that are well informed by our extensive dataset.

Regarding using the environmental water $\delta^2\text{H}$ values to approximate the $\delta^2\text{H}$ value of algal primary producers in lakes, this issue was already raised and addressed in the previous review round. To clarify, we used measured benthic algae from most of our lakes to estimate the $\delta^2\text{H}$ fractionation between algae and environmental water, with average values aligning very closely to those published by the Reviewer #2 themselves. We used this fractionation to approximate the average algal $\delta^2\text{H}$ value for each lake based on lake environmental water. We directly incorporated the uncertainty (variability observed across lakes in fractionations) into our models as part of the substantial standard deviations we supply along with the mean values of algal isotopic endmembers (s.d. of 23.6‰). There is therefore not a direct correspondence, but a distribution of algal values supplied for each lake from which individual consumers stochastically sample in the model. We would highlight that this direct incorporation is simply not possible using the Reviewer #2's preferred simple mass balancing approach.

With respect to the t-DOM versus t-POM utilization pathways, we feel that this has been both exaggerated and misconstrued, as has also been pointed out by Reviewer #1. The use of terrestrially derived POM and DOM is not a dichotomous choice as the Reviewer #2 seems to be suggesting. In our response to the previous round of revisions, we highlighted that in many instances it has been shown that t-DOM, via the microbial loop, can be the major contribution to zooplankton energy pathways. Arguing that zooplankton themselves only directly graze algae could also be considered a gross simplification, ignoring the importance of the microbial loop in remineralization processes that support continued algae production as well as potential heterotrophic uptake in phytoplankton that occurs in light limited environments such as brown lakes high in DOM. Regardless, the Reviewer #2's argument rests on the necessity of modifying environmental water contributions accordingly, suggesting at least two additional trophic steps from DOM to zooplankton. In his response Reviewer #2 assumes that environmental water contributions within microbial communities are the same as in metazoans, i.e. the omega values used here. Given that the value of approximately 0.23 is derived only from animals, this is a gross assumption to make given bacteria have considerably different metabolisms, physiologies, and biomass routing mechanisms. Osburn et al. (2016) demonstrated in *E. coli* that under heterotrophic culturing, environmental water contributions to bacterial amino acids are low, typically <20% except for alanine and no more than 12% for essential AAs (note that AAs make up the bulk of animal tissue biomass). This suggests that microbial trophic steps are less of an issue than the reviewer suggests, and we would caution against applying animal derived parameterisations to bacterial physiology and metabolism in general. Ultimately, we sampled and measured t-DOM and used it as a proxy for t-OM (i.e. both POM and DOM forms), because it is a sample that is far less likely to be contaminated by e.g. phytoplankton cells, aquatic detritus, faecal material and so on, and therefore better reflects terrestrial origin OM as it enters the lakes, and crucially because we have no prior expectation that there should be any isotopic differences between either dissolved or particulate forms of terrestrial organic matter as it is routed into animal tissues.

In conclusion, we believe that our manuscript presents solid and robust results, using analytical and statistical methodologies that are considered standard practice across the field. We acknowledge that in some instances the clarity of the manuscript could be improved, model structure tweaked, and sensitivities of the overarching inferences to different parameterisations tested, all of which we have adequately addressed and incorporated into our resubmission.

*Osburn, M. R., Dawson, K. S., Fogel, M. L., & Sessions, A. L. (2016). Fractionation of hydrogen isotopes by sulfate- and nitrate-reducing bacteria. *Frontiers in microbiology*, 7, 196727.*

Reviewer #3 (Remarks to the Author):

I was “Reviewer #3” in the assessment of Keva et al. original manuscript (NCOMMS-23-44093-T), which has now been revised. In general, the rebuttal letter does an excellent job at replying to all my concerns.

However, whereas the replies as such are OK, in some cases there is no explanation to how the manuscript was changed (or why it was not changed). Moreover, the authors provided no line number references or other means to make it easy to follow up on the changes. This being mentioned, I have carefully read the manuscript and found all changes that could be identified acceptable. It took some (unnecessary) time to identify the changes because the authors left no line number references or tracked changes to guide where to look. In the following cases I was not able to identify how the manuscript has been changed:

RESPONSE: *We are truly sorry for unintendedly causing problems for the review process. We provided all the asked documents (including track-changes version of the corrected manuscript) and tried to follow the author guidelines as precisely as possible. The exclusion of the line numbers in last revision round was based on misunderstanding of author guidelines and functioning of the PDF reconstruction in the online submission system. We hope in this revision it would be easier to follow the changes.*

General comment 4 from the last review round: Even if the authors have no biomass data, it is still possible to discuss whether or not the allochthony reflects patterns in absolute masses or fluxes. This is critical when interpreting what the results mean for the ecosystems and how they function.

RESPONSE: *Thank you for the comment. We believe that consumer allochthony and lake allochthonous standing biomass might not be very strictly correlated. It was hypothesized in the early 2000s that in very eutrophic conditions and on the other hand in very dystrophic conditions, suppressed essential fatty acid production (by specific algae groups e.g. diatoms) would hinder the growth of higher trophic level organisms. However, more recent papers (Hiltunen et al. 2019; Taipale et al. 2012) show that primary consumers are able to grow well with terrestrial derived matter or less essential fatty acid rich diet in circumstances where even some traces of essential fatty acids are available. Moreover, the biomass trends of primary consumers across landscape are most likely top down regulated as hypothesized many decades ago (Carpenter et al. 1985, 2001) and also shown later with environmental data (e.g. Keva et al. 2021). Furthermore, the primary producer, secondary consumers and tertiary consumers are most likely bottom-up controlled by the nutrient availability as they tend to increase with increasing nutrient availability. Therefore, it is likely that the highest lake standing biomass are found from the most eutrophic lakes (highest PC1 scores). Moreover, some studies have shown that phytoplankton productivity follows a hump-shaped pattern along the DOC gradient, where maximum productivity might occur around 15 mg DOC / L water (Bergström and Karlsson 2019). This could be*

caused by nutrient limitation below 15 mg DOC / L and light limitation over 15mg DOC / L. Yet with nutrient addition the phytoplankton production could be highest in lakes with lowest DOC. This nutrient addition scenario might represent our environmental gradient the best (in reverse PC1 scale). From the 35 sampled lakes, five had higher than 15mg DOC / L concentrations, and these lakes had systematically very high forest coverage and low PC1 scores. Therefore, the lowest biomass production could be found from the forested lakes (low PC1 scores). Thus, when combining these ideas of nutrient and light limitation, the overall biomass curve along our PC1 gradient could resemble a S-type curve increasing towards right side of the PC1 axis. As the global allochthony approach to zero along the PC1 axis, we believe that these factors would flatten the absolute allochthonous biomass trend across the environmental gradient (PC1). However, this idea remains to be tested in future studies.

We have now added this type of reasoning to the manuscript in LINES 262-275

Bergström, A. K. & Karlsson, J. Light and nutrient control phytoplankton biomass responses to global change in northern lakes. *Glob Chang Biol* 25, 2021–2029 (2019).

Carpenter, S. R., Cole, J. J., Hodgson, J. R., Kitchell, J. F., Pace, M. L., Bade, D., Cottingham, K. L., Essington, T. E., Houser, J. N., & Schindler, D.E. (2001). Trophic cascades, nutrients, and lake productivity: Whole lake experiments. *Ecological Monographs*, 71, 163–186.

Carpenter, S. R., Kitchell, J. F., & Hodgson, J. R. (1985). Fish predation and herbivory can regulate lake ecosystems. *BioScience*, 35, 634–639.

Keva, O. et al. Increasing temperature and productivity change biomass, trophic pyramids and community-level omega-3 fatty acid content in subarctic lake food webs. *Glob. Chang. Biol.* 27, 282–296 (2021).

Hiltunen, M. et al. Terrestrial organic matter quantity or decomposition state does not compensate for its poor nutritional quality for *Daphnia*. *Freshw. Biol.* 64, 1769–1786 (2019).

Taipale, S. J., Brett, M. T., Pulkkinen, K. & Kainz, M. J. The influence of bacteria-dominated diets on *Daphnia magna* somatic growth, reproduction, and lipid composition. *FEMS Microbiol. Ecol.* 82, 50–62 (2012).

Specific remark 10 from the last review round: I don't see how the manuscript has been changed to clarify gut evacuation protocol and its implications.

RESPONSE: Thank you for this comment. We have revised this whole sentence according to our previous response, see LINES 375-379

Specific point 11 from last review: I don't see how the manuscript has been changed to justify the approach for dealing with exchangeable hydrogen and discuss its weaknesses relative to e.g. steam equilibration. The answer provided in the rebuttal is fine, but what does that matter do future readers of the paper, I mean if there was no change made in the actual manuscript?

RESPONSE: Thank you for the comment. If we have understood correctly, the peer review process is made open for public if the manuscript is accepted. An interested reader could have found the asked information from the peer review reports. However, we have now included this in the 'caveats' section in LINES 527-535.

Response Letter: 3rd revision

We would like to thank the three Reviewers for their contributions to the third round of revisions. Based on Reviewer 2's thorough checking of our calculations and models, we have now corrected the lake water $\delta^2\text{H}$ values, recalculated photosynthetic fractionation of deuterium, updated the consumer omega-corrected values and run two different scenarios of the models using different source values (i.e., modelled bacterial $\delta^2\text{H}$ values as the terrestrial source and sampled benthic algal $\delta^2\text{H}$ values as the aquatic source) and updated the models in the manuscript accordingly. We have also added text to the caveats and discussion sections about the probable bacterial step in the food webs and how this may impact the model results. In fact, these modifications and different model scenarios produced very similar results to those from the initial model. Thus, we are now even more confident that the main findings of the study are robust. We thank all the Reviewers for their extensive comments during this and previous rounds which have improved the final version of our manuscript, and we have recorded our thanks in the Acknowledgements section.

Comments from Reviewers (Text in blue with normal font)

Response from authors (Text in black font)

Reviewer #1 (Remarks to the Author):

I thank the authors for going beyond what was necessary (e.g. adding new analyses) to address my remaining concerns. I have no further comments. Congratulations on the excellent paper!

Response: We thank Reviewer #1 for the positive feedback and insight provided here and throughout the previous revisions.

Reviewer #1 (Remarks on code availability):

I have skimmed the repo and it does seem like the code is a usable community resource (note I have not tried running it). The README file has suitable instructions but it could include more information on how to install and run the code for beginners.

Response: We have added comments to the R scripts and they should be now clear enough for beginners to understand what we are doing at each step.

Reviewer #3 (Remarks to the Author):

The authors have done an excellent job responding to all of my comments. I have no further concerns.

Response: We thank Reviewer #3 for all the helpful comments during the previous revision rounds and for recommending our manuscript to be accepted for publication.

Reviewer #2 (Remarks to the Author):

Main points

1. Is the algorithm making the data fit the hypothesis?

In their second reply, the authors stated “we sincerely hope that Reviewer #2 is not suggesting that, by collecting and analysing thousands of samples over three years, applying the best analytical data corrections according to current knowledge, including the use of other measured data to estimate consumer TPs, and constructing a mixing model that tests for the effect of a well-reasoned environmental gradient while concurrently accounting for the complex data structure, we are merely ‘making our data fit the hypothesis’”. In fact, I will state very clearly that given the way that the Bayesian algorithm has been set up, the authors can profoundly change the modeled outputs just by changing the structure of the algorithm or some of the assumptions imbedded within the algorithm. This is precisely my biggest concern with this analysis. For example, in the last round of review I reanalyzed the authors’ Asellus data and my analysis showed a very flat response between the estimated allochthony and the Environmental gradient (PC1). I also pointed out a discrepancy in how the authors calculated the trophic level for this and other consumers. In their revision Keva et al. reanalyzed their data for Asellus and now report a completely different response, albeit a response very similar to the one I previously found for these data. The change in these two plots was solely due to how the Bayesian model was structured and how some of the underlying data were interpreted!

Response: As stated in the previous response, this was a valuable comment and improved the manuscript greatly. We emphasise that we are not changing the algorithm used to fit the model structure but are changing the model structure to be parameterised. Our initial model included only a single average slope effect (β_0) for the effect of PC1 (i.e. $\text{Allo.} \sim \beta_0 \cdot \text{PC1}$), as our aim was to quantify the general trend across lake food webs. As Reviewer 2 highlighted during the first round of review, the assumption of a single slope effect across all species was poor when interrogating data at the species-level. This unsuitable model structure was also pointed out by Reviewer 1. We since made a custom jags-script to run the model to instead allow the slope effect of PC1 to vary with species around a global average slope (i.e. random slope effect as in a linear mixed effects model: $\text{Allo.} \sim (\beta_0 + \beta_{\text{ spp.}}) \cdot \text{PC1}$). We are grateful that Reviewer 2 provided these comments that led us to improve the model structure to better reflect our systems. We are confident the model structure is now correct, as does Reviewer 1 according to their comments. More generally, it should not be surprising that with different model structures come different interpretations across all facets considered. However, we highlight that even with this additional structuring within our model, the values of the average slope effect with PC1 (β_0) are near identical in both cases and therefore this does not change our overall interpretation of general trends in allochthony across lake ecosystems, even if we now also quantify the interspecies differences with this improved model structure.

Additionally, as we all agree, changing the assumed environmental water assumption profoundly changes the model outputs! In their response to my second round of comments the authors stated: “We would also highlight that due to the crossed random effects structure of the populations we have implemented here, population posteriors are also informed by those from other lakes for the same species, and other species from the same lake.” Firstly, I do not feel this detail is explained adequately in the MS. Perhaps this is what the authors are alluding to on lines 487-488 when they state: “in the MixSIAR model, we set lake and consumer taxon as random factors.” But unless you are fluent in Bayesian jargon, you would have no idea what they are getting at here. Secondly, it is evident that smearing the outcomes across “lakes for the same species, and across species from the

same lake” will tend cause all of the results to converge on a common outcome that is not consistent with the attributes of the original raw data for individual consumers. This is a very clear example of where the structure of the Bayesian model profoundly influences the outputs.

Response: We would like to point out that random and fixed factors are not “Bayesian jargon”, they are general terms related to statistics and model structure. They can and should be applied in frequentist statistical analysis when appropriate as they provide the levelling at which data cannot be assumed to be independent and identically distributed, a fundamental assumption in almost all data analysis. The application of fixed and random effects in mixing models is well described in the MixSIAR paper we cite in our manuscript (reference number 27). We also provided in the last round an excellent paper dealing with random and fixed effects in frequentist data analysis. Random effects do not smear the outcomes, but model factor level effects as a gaussian distribution around the average level effect (after accounting for fixed effects) rather than modelling differences between an arbitrary reference level and all other factor levels. This distribution accounts for uneven sampling across levels whereby levels with small sample sizes are weighted less, and therefore tend towards the centre of the distribution unless data strong evidence the contrary. This is wholly consistent with the data – if data strongly evidence a departure from the average, then that level is modelled as such. This random factor structuring is a realistic reflection of natural systems whereby lakes tend to be similar to each other to some degree, but individual data points are generated from stochastic processes. We emphasise that generally the crossed random effects structures do not ‘smear’ results, but provide insight as to whether there are consistent species differences that occur across lakes and likewise consistent whole lake differences that occur across species in the ecosystems. Generally, these model structures are crucial in both Bayesian and frequentist statistics. It should be obvious that changes in model structure influences the model outputs and therefore interpretations where structures diverge. We are confident the model structure is now correct as also indicated by Reviewer 1.

In the previous round of review, I asked why the some of the results for the different taxa appeared to show nearly identical patterns. For example, in the plot to the right I have highlighted in red several clusters of outcomes that are oddly very similar. Real data rarely show these sorts of patterns. Now in retrospect, it is apparent that these anomalous results are probably because the authors “crossed” the outcomes across “lakes for the same species, and across species from the same lake”. In their response, Keva et al. also stated “given the functional similarities of Asellus and bulk zoobenthos, and perch with ruffe, it should not be surprising that their estimated terrestrial contributions are similar on a lake-by-lake basis.” I believe it is misleading to imply that these anomalous patterns in the plots have some deeper ecological meaning when it is much more likely they are almost entirely due to the way the authors set up their Bayesian mixing model! It may seem stark, but I am very concerned that 1) by crossing lakes and species and species and lakes, 2) by changing how this crossing is done from one MS draft to another, 3) by modifying your definition of trophic level (so that some consumers have trophic levels < 2 or not), 4) using modeled data instead of real, 5) not fully accounting for environmental water contributions, and 6) by choosing one environmental water assumption over another there are a multitude of opportunities to make the data fit a particular hypothesis!

Response: The results Reviewer 2 is referring to are the plots in which we visualise for the reader the parameterisations of PC1 (the slopes), species (different panels), and lake (offsets from the slope) effects from the mixing model. This figure does not show our measured isotope data; they are plotted elsewhere. There is nothing anomalous about the posterior estimates Reviewer 2 is pointing to here. The highlighted areas showcasing the fact that whole lake intercepts are consistent across

each species as the effects are crossed, which Reviewer 2 has pointed out. It should be entirely expected from an ecological view, that lake wide effects on allochthony not captured by PC1 should manifest across all species that occur within the same lakes and it is prudent to account for this in models. With respect to the specific numbered points raised: 1) We already provided a detailed response to these comments on the model structure in the previous revision round. We do not feel any further response is necessary. 2) We are grateful to Reviewer 2 for noticing that species-specific slopes were not originally included in the models, but we already made appropriate modification based on your comments in the previous revision, so we do not feel that any further response is necessary. Further, the crossed random effects structure of lake and species intercepts to which Reviewer 2 alludes have in fact remained consistent across all model structures throughout the review process and have not changed from one draft to the next. As discussed before, we are certain that the current model structure is correct. Despite what Reviewer 2 appears to be suggesting, we know very well that, when possible, it is better to use random and fixed factors in Bayesian and frequentist data analysis according to the hypotheses. 3) We provided a detailed response on how we calculated the trophic position of each individual already in the previous response round and this is clearly explained in the manuscript. We think using $\delta^{15}\text{N}$ values to calculate a continuous trophic level estimate (a procedure that is widely accepted) is better than fixing the trophic level of consumers based on some fixed preconceptions about consumer diets and trophic levels, as many consumers have quite flexible diets and the food webs of different lakes might be rather different. The modification to which Reviewer 2 is referring was in response to their observation that some of our individual consumers had unreasonably low trophic positions. 4) We have now provided model results based on the sampled benthic algae $\delta^2\text{H}$ values (please see Fig R2), and we have now also included this figure in the supplementary material and a more detailed description of the selection of source values for the model in the Methods section. We emphasise that the values we use in the main manuscript are based on measured $\delta^2\text{H}$ values from lake water at all sites coupled with water-algal discrimination values determined from a substantial subset of lakes where suitable samples were available. We show that this empirical discrimination value aligns extremely well with those reported elsewhere in the literature. The constant reference to these end-members as 'modelled' data implies that we are ignoring empirical data in contrast to the reality. 5) We have followed Reviewer 2's suggestions for selecting the omega values and done our best in recalculating the earlier biased omega values reported in Brett et al. (2018). We also conducted sensitivity analysis on this aspect covering a range of omega values that covers reasonable uncertainty in water contributions across different consumer groups. 6) We would like to emphasise that all lakes and consumers were treated in the same way and thus there is no way that selecting the omega values would have modified the trends observed in this study. The comment suggests that perfect knowledge on water contributions is known and widely available in the field which is categorically not true. There remains much uncertainty and unexplained variations in omega values across studies, which we have incorporated into our analyses. We do not understand why Reviewer 2 is making such accusations of data manipulation, and we categorically refute them.

2. Why did I only reanalyze the *Asellus* data in the second round of review?

In their response, the authors questioned why I only reanalyzed the *Asellus* data. Specifically, Keva et al. state "looking at the raw data alone in Fig.2 would suggest that *Asellus*, of all 19 taxa groups samples, would be the most likely to produce a flat trend with PC1. It is circumspect therefore that the reviewer sought only to analyse this specific group, rather than the many others that show decreasing trends in the raw data, but to subsequently cast dispersions upon potential trends in the other taxonomic groups." I reanalyzed the results for *Asellus* first because this taxon was first alphabetically when I sorted the raw data file by taxa. I do not agree that the flat response for *Asellus*

was evident just by looking at the “raw data in Fig. 2”. In my opinion, Fig. 2 is a very difficult figure to decipher (and it should not be included in the MS). I also hope Keva et al. recognize that reanalyzing another author’s raw data is a very time-consuming task, so I only had time to reanalyze the Asellus data before submitting my second review (especially because Nat. Comm. was sending me daily reminders that my review was late).

Response: We thank Reviewer 2 for clarifying this. We believe that Figure 2 is the best way to illustrate the mixing envelope and shows that there are no trends in the source values along the environmental gradient. We note that the other reviewers did not appear to have any difficulty understanding this figure, and that similar figures have been used in other publications (e.g. Hayden et al. 2021; Fig 3). Finally, we appreciate the effort Reviewer 2 has made in reanalysing our results (but again with a dissimilar model structure; as we suggested previously, in the comparison it would have been better to use the same model structure as we used but with a frequentist approach). Regarding the pressure of time to complete a review, this of course is a matter between the reviewer and the editor, and is not something we have any say over.

But now that I have had more time to look into this, I will say that the allochthony responses for 5 of the 19 taxa were flat with regard to PC1 (see table below). The functional responses for the four consumers with the highest estimated allochthony (= 52-88%) were very flat ($r^2 = 0.00-0.08$) (e.g., profundal chironomids, Asellus, profundal BMI, and littoral BMI). The response for copepods was also very flat ($r^2 = 0.06$), but this group also had a negative estimated average allochthony (e.g., -22%). Paradoxically, many of the taxa that showed the strongest functional responses with PC1 ($r^2 = 0.40-0.63$), also had negligible or even quite negative average estimated allochthony (e.g., vendace, medium perch, bulk zooplankton, bleak, large roach, Chaoborus, and small perch). The overall average r^2 values between the estimated consumer allochthony with PC1, forest area and agricultural area were only 0.32, 0.21, and 0.23, respectively. In particular, the relations between forest and agricultural landcover and consumer allochthony were rather weak given the mechanisms hypothesized and broad conclusions from the Keva et al. MS.

Response: We appreciate that Reviewer 2 has further analysed the complete dataset from their viewpoint to estimate species-specific average allochthony contributions. The (presumed) five taxa in their table in the review suggests having flat responses with PC1 show correlation coefficients of $r = 0.03$ (Asellus), -0.157 (litt. BMI), -0.196 (prof. chiron.), -0.247 (copepod), and -0.28 (prof. BMI). We have already highlighted previously that our species-specific random slope effects indicate Asellus to have a posterior bounding zero, matching with the above correlation coefficient ($r=0.03$). It is unclear what definition of ‘flat’ Reviewer 2 uses; however, the other four taxa show negative, albeit weak, correlations with PC1. This matches our model outputs that indicated the posteriors of these slopes do not bound zero at the 95% credible intervals. It is important to emphasise however that correlations do not correspond to slope coefficients and therefore cannot be used as a measure of slope ‘flatness’ even if they inform on the sign of the slope. Reviewer 2 then switches to using R^2 values of these correlations, the reasoning of which is unclear, but again, the coefficient of determination does not provide a measure of ‘flatness’ in relationships between two variables. We would highlight however that the Reviewer 2’s estimates of species-specific average allochthony values correspond well to the outputs of the model as shown below, and therefore our model reproduces the results determined from their own analysis.

Secondly, we again highlight the strengths of the Bayesian modelling approach we use compared to their averaging across lakes at the species level, including accounting for the more complex data structuring into a single model, incorporation of all uncertainties, and many other advantages we have given throughout this review process.

Reviewer 2 then suggests that coefficient of determination values of our allochthony estimates with PC1 (0.323), forest area (0.214), and agricultural area (0.228) are low. Although perhaps low compared to values expected from more controlled experimental studies, given the many mechanisms and processes that contribute to and influence stable isotope values of terrestrial inputs, their influx rates, relative aquatic production rates, trophic modifications, spatial heterogeneity, and so on, we believe these values are worthy of recognition, and contribute to our understanding of the functioning of lake ecosystems.

3. Yes, I am arguing that the Law of Mass Conservation is de facto correct!

The Law of Mass Conservation is an immutable fact of the world we live in! [In more specific theoretical cases, this law needs to be modified to account for the mass-energy equivalence, for example in nuclear reactions. However, this exception does not apply to Keva et al.'s study]. So anytime the results of a simple mass balance analysis and a Bayesian analysis differ the presumption is that the Bayesian approach has distorted the outcomes in some way relative to the original raw data. I agree with Keva et al. that the Bayesian approach is partially based on mass balance, but by collapsing the outcomes to the 0-100% domain, it deviates markedly from the correct mass balance. For this reason, I think an algebraic approach should always be used to analyze simpler datasets like the one from Keva et al. The Bayesian approach is a useful approach in more complex cases (e.g., when there are many potential resources and/or many dietary tracers) where the underlying algebra is far more complex. But in any case, it is essential to ascertain to what extent are the model outputs a function of the algorithm itself or the underlying raw data. This is the key conundrum for any environmental modeling exercise, especially Bayesian dietary mixing models (Brett 2014).

Response: We disagree with Reviewer 2. We reiterate again that we are not disputing the law of mass conservation. However, we re-emphasise that in addition there is also the natural law of diet proportions being restricted to the domain of [0,1]. Both need to be adhered to if we are to adequately describe trophic systems as they occur in nature using tracers. Contribution values outside of this domain indicate that the system is inadequately described. This DOES NOT mean that the law of mass conservation is violated. However, it does indicate that all processes that influence diet AND tracer values are not suitably captured. This includes sources of natural variability in source, consumer, and TDF values that simply cannot be incorporated into the simple algebraic approach in a

realistic manner. Consumer values falling far outside of the mixing space will of course result in values collapsing to the domain boundary, and this is why care and attention needs to be given to adequately ensure the mixing system being modelled is suitable. In such instances the algebraic approach simply provides biologically infeasible solutions well outside of the dietary domain. The “correctness” of the mass balance algebra that Reviewer 2 consistently falls back to comes, as we extensively explained in our prior response, from the one-to-one mapping that occurs with such injective algebraic functions. This should never be used as proof of validation as it is purely a direct function of the underlying algebra or ‘algorithm’. As we are estimating latent (i.e. unknown) variables, properly validating results is essentially impossible in natural ecosystems. Therefore, robustness must come from the ability of models to realistically capture underlying system processes and our understanding of those processes. We believe that we have gone above and beyond in ensuring that our data and modelling approach is appropriate, and consistent with its use in a very large number of other papers in this field published by numerous other authors and experts. We also note that the other two reviewers have not identified any concerns about our approach.

4. Am I disdainful of Bayesian mixing models?

In their response, Keva et al. stated “The Reviewer #2’s disdain for Bayesian mixing models has been apparent throughout, but we have found no reasonable argument against these statistical frameworks presented by the reviewer that stands up to scrutiny.” I do not have disdain for Bayesian dietary mixing models, but I do think these models are often used in an uncritical way that leads to a bias towards everything is somewhat important outcomes (e.g., the classic “plug-and-chug” approach). The essential step that should always be followed is proper model corroboration. Do the model outputs match up with the original raw data? Unfortunately, this is very rarely done in the Bayesian mixing model literature. In their second response, Keva et al. state “we have now calculated also pseudo R² values from the predicted tracer values for each of the models according to Gelmann (2018) and Stock et al. (2017). The results can be found in Table S8 . . .” If I understand correctly, the corroboration results represent “a posterior approximation of the proportional isotope variance explained by the model”, in other words, can the model outputs be used to recreate the original raw data. An approximate R² value of 0.310 indicates the model outputs did a poor job of recreating the original raw data. I would hope for a corroboration R² of >80% for such a simple problem. For example, in my as of yet unpublished fatty acid based mixing model analysis of dietary pathways in the Upper Klamath Lake food web, the corroboration between modeled and observed consumer fatty acid composition was $r^2 = 0.93$. I believe the poor corroboration in Keva et al.’s study is due to the Bayesian algorithm rounding all outcomes to the 0-100% domain. Keva et al. should present the raw results for their corroboration in a much more detailed way, e.g., what is the observed and predicted consumer raw $\delta^2\text{H}$ value for each of their samples so that this could be properly assessed. Without this additional information, my concerns about the Bayesian model’s ability to recreate the original raw data are only reinforced.

To test my conjecture that the poor model corroboration results reported Keva et al. are related to attributes of the Bayesian mixing model framework, I corroborated the mass balance model’s predicted raw medium perch $\delta^2\text{H}$ values with the mass balance allochthony, after setting all < 0% and > 100% allochthony estimates equal to zero and one, respectively. This comparison showed the mass balance model outputs were perfectly corroborated when the allochthony estimates were within the 0-100% domain (as expected) and quite poorly correlated when the perch allochthony estimates were negative. I suspect Keva et al.’s corroboration results were even worse than this because of their choice to use a crossed random effects structure in their model which would smooth the outcomes across consumers and lakes.

Response: We have certainly critically evaluated our model at every stage of data analysis as well as during the process of review. Proper model corroboration however depends on the objective of the model. The purpose of our model is to provide inference on the role of the environment on the degree of allochthony in lake ecosystems. As allochthony contribution is a latent variable and cannot be measured, we use stable isotopes as tracers. As an inferential model, the number of parameters to be estimated is relatively small, given the sample size (in our case, $n=1737$). We invoke just 3 parameters, PC1 as a continuous effect and the main parameter of interest, and the factors species with 19 levels and lake with 35 levels which need to be accounted for. Given a global intercept plus PC1 random slopes with species, this means our model has a total 75 parameter level estimates. It is from these three parameters alone which we reconstruct stable isotope values to provide our pseudo- R^2 of 1737 consumer observations using source mean values. The use of mean values alone for sources was for simplicity rather than incorporating stochastic uncertainty, as source values drawn during MCMC procedures are discarded and hence not known. Given we are reconstructing isotope values of consumers from 19 species and 35 lakes based on an inferential model with only three parameters, we strongly contest the premise that an r^2 value of 0.31 is poor (please note that the updated model r^2 is 0.38). Of course, and as we have mentioned in previous responses, if the objective was to maximise isotope predictive power, then we could have fitted an individual based model to our data, involving 1737 parameter level estimates. Such a model however would provide no direct inference on the underlying processes that are of interest to us.

With respect to the Reviewer 2's unpublished study of a fatty acid based mixing model analysis of dietary estimations for a particular lake for which we have no details, it is hard for us to comment. However, the fact that it is a single lake already suggests a much simpler system than what we are dealing with, and a single lake would not provide inference on the environmental factors at the regional scale which we are addressing here. The Reviewer 2's statements about algebraic models performing poorly when forced to constrain to dietary domain [0,1] is something we already pointed out in the previous round of review and above; has little bearing here; and highlights the inadequacies of applying such an approach.

As we stated earlier, the crossed random effects structure accounts for the fact that individuals of the same species are expected to respond more similarly to each other than other species across lakes, and likewise individuals from the same lake regardless of species are expected to respond more similarly than individuals from other lakes due to whole lake processes. This is the natural data structure and is highlighted in the reviewers table showcasing average species level differences that corresponded well with over model outputs using a crossed random effects design.

5. Does the terrestrial d_2H ratio get modified during a microbial pathway?

In their response, Keva et al. argued that it was not necessary to account for multiple trophic levels and environmental water exchange in a microbial pathway from t-DOM through bacteria and protozoa to zooplankton and benthic macroinvertebrates. Keva et al. cited some details from Osburn et al. (2016) and concluded "Osburn et al. (2016) demonstrated in *E.coli* that under heterotrophic culturing, environmental water contributions to bacterial amino acids are low, typically <20% except for alanine and no more than 12% for essential AAs (note that AAs make up the bulk of animal tissue biomass). This suggests that microbial trophic steps are less of an issue than the reviewer suggests, and we would caution against applying animal derived parameterisations to bacterial physiology and metabolism in general." I just read Osburn et al. (2016) and I have a completely different assessment of that paper than Keva et al. Firstly, it does NOT appear to me that Osburn et al. examined alanine or any other amino acids as Keva et al. claimed. [Specifically, the words alanine and amino acid do not appear in the Osburn et al. (2016) paper]. Instead, the Osburn

et al. (2016) study focused on lipid metabolism, or more specifically fatty acids. Osburn et al. (2016) concluded “large variations in the magnitude of [hydrogen] fractionation are observed for many heterotrophic microbes utilizing different central metabolic pathways.” These authors noted that bacterial hydrogen fractionation depends on the bacterial strain, the electron donor (the substrate), and the electron acceptor (O₂, NO₃⁻, SO₄²⁻, etc.). Osburn et al. (2016) also showed the δ²H values of bacterial lipids were very strongly modified by the ambient water (see their Fig. 5). Perhaps Keva et al. confused the paper by Osburn et al. (2016) with a paper by Fogel et al. (2016) which did look at environmental water contributions to bacterial amino acids? Fogel et al. (2016) found that on average, the bacterium *E. coli* obtained 17% of its amino acid hydrogen from environmental water (see Fogel et al.’s Table 1).

Based on the results by Fogel et al. (2016), as well as Osburn et al. (2016), it is very evident that the original δ²H values of t-DOM will be extensively modified (perhaps in a difficult to predict manner) as terrestrial organic matter is transferred up through the food web via a microbial pathway (i.e., t-DOM to bacteria to protozoa to zooplankton). Keva et al. are still ignoring this modification in their Bayesian mixing model, which is directly contradicted by the paper I believe they intended to cite. I suggest Keva et al. assume an environmental water value of 0.17 for the t-DOM to bacteria step. I also suggest using a value of 0.20 (i.e., the mean of 0.17 and 0.23) for the bacteria to protozoa step. Ignoring environmental water contributions along the microbial pathway, as Keva et al. are currently doing, would be a major error.

Response: We thank Reviewer 2 for clarifying this issue. Yes, we were referencing the Fogel et al. (2016) study here and apologise for this genuine citation mistake. We are glad that Reviewer 2 was able to correctly identify the paper to which we were referring.

We agree that including the bacterial step could, in principle, refine the model. However, even with the provided reference (a laboratory study of fractionation by *E. coli* for some amino acids, which is of questionable relevance to our lake environments) and the reasoning from Reviewer 2, any selected omega values for bulk bacterial mass will be highly speculative, far above the uncertainties we already attributed at higher consumer levels. This is clearly an issue that can only be properly resolved after further research into bacterial fractionation. However, to provide some insight into this issue and its possible influence, we applied the suggested omega of 0.17 (mean of AA omega values from Fogel et al. 2016 Table 1) and modelled the bacterial δ²H values with the formula provided by Reviewer 2 and rerun the models. We have provided the results with this response letter (Fig R1) and also included them in the supplementary material (Supplementary Figure 8) and provided more discussion about this topic in the Discussion and Caveats section of the manuscript. It is apparent that the allochthony estimates for the consumers are then slightly decreased as expected, but the main result of the study was not impacted at all – consumer allochthony trends with PC1 remained negative and near identical with previous model structures. Given the current lack of knowledge about these microbial fractionation values, we feel it is best to leave the original results and figures in the main text and to present those modified figures which incorporate a putative microbial fractionation in the supplementary information. Readers can then judge for themselves how this affects the interpretation of results. We thank Reviewer 2 for his useful input on this issue.

In their reply to, Keva et al. stated “Reviewer #2 is correct that we did not account for the microbial link trophic step in the δ²H correction formulas. The only published way to do this would be correcting the different sources to consumer trophic level (Keva et al. 2022); however, with this setup applying MixSIAR this would be impossible.” This response makes no sense, of course it is possible to correct these data for environmental water exchange within the microbial pathway. The very simple mass balance equation to do this is: $Bact\ d2H = tOM\ d2H * (1 - \omega) + lakewater\ H2O\ d2H * (\omega)$.

Based on the average t-OM and lakewater d2H values for Keva et al.'s data (i.e., -120.6‰ and -73.9‰, respectively), the recalculated mean d2H values would be -112.6‰ and -104.9‰ for the bacterial and protozoan steps of the microbial pathway. In essence, 34% ($= 1 - 0.83 \times 0.80$) of the original hydrogen contained in the terrestrial organic matter would get exchanged with hydrogen in the lakewater along the microbial pathway. Unfortunately, these corrections won't completely resolve this conundrum as the results from Osburn et al. (2016) suggest bacterial metabolism may modify organic matter d2H values in complex ways regarding both fractionation and exchange. Keva et al. should acknowledge that they only have a very rough idea what the d2H values of the terrestrial-microbial pathway are in their lakes and this represents a very large source of uncertainty in their dietary modeling approach. But to be clear, not correcting for and ignoring this source of uncertainty (as Keva et al. advocate doing) does not make it go away.

Response: Our comment about being unable to incorporate multiple trophic steps directly into the MixSIAR framework still stands. Instead, Reviewer 2 is suggesting we transform our measured diet prior to applying mixing models. The suggested approach is slightly different compared to Keva et al. (2022) and can be applied with the MixSIAR model. We have now applied this approach and re-run the model with the suggested bacterial omega scenario, with the results presented at the end of this document (Fig R1). As one can see from the model result, the consumer allochthony trend along the environmental gradient from forested to agricultural catchments essentially remains the same and the model modification does not change at all the interpretations of the results. However, we would also highlight that this approach is unable to account for any uncertainty in the presumed water contributions which is far from ideal given their speculative nature. See also our response to the previous comment.

6. Can tDOM be used a proxy for tPOM and vice versa?

In their second response, Keva et al. suggested tDOM d2H values could be used a proxy for tPOM value and vice versa. I am mostly in agreement here, provided the tDOM and tPOM d2H values are strongly correlated across the 35 lakes. BUT so far, the author have only presented data showing the overall means for their tDOM and tPOM samples are similar. This is not sufficient data to decide whether one can be used as a proxy for the other.

Response: Reviewer 2 is incorrect on this point as we were not suggesting that tDOM $\delta^2\text{H}$ values could be used as a proxy for tPOM. We were arguing that inlet DOM $\delta^2\text{H}$ is a good proxy for terrestrial hydrogen and better than POM. We have now provided the requested details in the supplementary material. However, since we have now included the bacterial scenario model in the supplementary material, we feel this topic conflates with many other suggestions and comments posed by Reviewer 2.

7. Why weren't real algal d2H values used in the mixing-model?

As discussed in my previous reviews the paper implies that real data were used to represent the algal d2H values, but in the fine print of the methods section the authors mention that "modeled" algal data were actually used for the mixing-model analyses. As I have already noted this was explained in a misleading way, and I cannot envision a justification for using "modeled" data when real data are available! For example, on lines 69-72, the MS states "We used $\delta^2\text{H}$ to differentiate terrestrial and aquatic OM sources and used Bayesian mixing models to generate estimates of consumer allochthony by sampling terrestrial inlet dissolved OM (t-DOM) and aquatic algae to represent the basal allochthonous and autochthonous resources, respectively". Subsequently, on lines 469-473 the authors state "For most lakes, it was impossible to obtain sufficient samples of pure phytoplankton

for SIA. Therefore, the phytoplankton $\delta^2\text{H}$ values were modelled from lake water $\delta^2\text{H}$ values with a photosynthetic discrimination estimate of -106.7‰ derived from the mean difference between measured benthic algae and pelagic water $\delta^2\text{H}$ values across the sampling sites, assuming that photosynthetic fractionation of hydrogen is equivalent for phytoplankton and benthic algae". On lines 360-361 on the MS they state "Benthic algae were sampled by scraping visually green patches of algae from hard substrate surfaces from many locations (e.g. sublittoral rocks from <1 m depth) and pooled on site to gain enough sample material for SIA". In their response to my second round of comments the authors state "it was impossible to obtain sufficient samples of pure phytoplankton for SIA". Once more we try to reason this to Reviewer #2". The authors completely misunderstand my point here and I suspect they are employing a red herring. I recognize that it could be difficult to obtain a pure sample of phytoplankton for SIA. That is not my point. To reiterate, these authors do have benthic algal samples, and from these samples they calculated the average photosynthetic hydrogen discrimination for their lakes is -106.7‰. They also state that they assume that photosynthetic hydrogen fractionation is equivalent for phytoplankton and benthic algae. If you assume that phytoplankton and benthic algae have equivalent photosynthetic hydrogen fractionation, which averages -106.7‰, then you are in fact also assuming that within a particular lake the phytoplankton and benthic algae will have the same $\delta^2\text{H}$ values. This equivalency for phytoplankton and benthic algae $\delta^2\text{H}$ values is also reflected in the fact that the raw data provided by the authors for this MS only presents one $\delta^2\text{H}$ value for BOTH phytoplankton and benthic algae for each lake. My point is that since you do have actual real benthic algal $\delta^2\text{H}$ data, you should use that instead of "modeled" data. This seems like a very easy place for the authors to modify their approach so that their analysis is based as much as possible on real data. If repeating the analysis using the real observed $\delta^2\text{H}$ for the benthic algae does not change the outcomes of their analysis that would be good to know. But regardless of the circumstances (Bayesian, frequentist, mass balance), I will always advocate for the use of real data in lieu of "modeled" data. The authors should also report the raw $\delta^2\text{H}$ benthic algae data for each of their lakes.

Response: We do not have measured pure phytoplankton values from any lakes for reasons that we have explained and which Reviewer 2 understands. There are clearly different approaches to resolving this conundrum. We think we all agree that phytoplankton $\delta^2\text{H}$ values can legitimately be assumed to be the same as benthic algal values, as has been widely assumed in previous studies, on the basis that fractionation by unicellular algae is probably the same regardless of whether they are in suspension or associated with a surface. As we understand him, Reviewer 2 feels we should use the actual individual lake benthic algal values as the aquatic source for pelagic and benthic food webs in the model. We feel it is more consistent to use the calculated mean fractionation value (which actually corresponds closely to values reported in the literature) to calculate aquatic source $\delta^2\text{H}$ values from the measured water values for each of the lakes. This has the added merit of encompassing inherent variability in the individual lake measured algal values caused by, for example, spatial and temporal fluctuations in algal community composition or by small differences in the purity of the benthic algal samples. Of course, we agree with Reviewer 2 that it is generally preferable to use directly measured data as the input to models, but there can be occasions when it makes more sense not to apply this general principle, and we believe this is one such occasion. In practice this issue appears to be something of a "storm in a tea cup". We have run the model using the actual benthic algal $\delta^2\text{H}$ values (see Fig R2 at the end of this document) and it makes no difference to the model outputs and the conclusions. Hence, we prefer to retain our approach rather than following the Reviewer 2's preference.

The author's decision to use "modeled" algal $\delta^2\text{H}$ values instead of real data means that in all cases the ambient water and primary producer $\delta^2\text{H}$ values were nearly PERFECTLY CORRELATED (see left-

side plot below). This is a completely unrealistic condition. I was curious how strongly the ambient water and algal d2H values might be correlated if as Keva stated photosynthetic hydrogen discrimination averaged $-106.7 \pm 17.9\text{‰}$ (± 1 SD) and the ambient pelagic water in their lakes had an average d2H value of $-73.9 \pm 8.7\text{‰}$. I did this by using a random number generator to generate hypothetical photosynthetic hydrogen discrimination and ambient water d2H distributions with means of $-106.7 \pm 17.9\text{‰}$ and $-73.9 \pm 8.7\text{‰}$ ($n = 1,000$), respectively. I then added the photosynthetic hydrogen discrimination values to the ambient water value to generate “modeled” algal d2H values. I then regressed the ambient water d2H values against the modeled algal d2H values (see the right-side plot below). This Monte Carlo simulation showed that given the inherent variability in the photosynthetic hydrogen discrimination and ambient water d2H values, there is likely only a weak statistical association between ambient water and modeled algal d2H values (see right-side plot below). It is also worth pointing out that Keva et al. reported the spatiotemporal variation in within benthic algal d2H values was $\pm 23.6\text{‰}$ (± 1 SD), during a time frame when spatiotemporal variation in within lake water d2H values was only $\pm 2.2\text{‰}$. Data presented in Table 2 of Brett et al. (2018) similarly showed that when estimated multiple times within a system photosynthetic hydrogen discrimination varied by $\pm \pm 23.1\text{‰}$. Based on this evidence, I can say that it was unjustified for Keva et al. to assume that the d2H values for ambient water and the modeled algal in their lakes were perfectly correlated.

Response: It is correct for Reviewer 2 to highlight the uncertainty with source $\delta^2\text{H}$ values. We have now changed the term “modelled algal $\delta^2\text{H}$ values” to “photosynthetic fixation-adjusted water $\delta^2\text{H}$ values”, as this makes it more clear how we derived the aquatic sources values used in the model. As argued by Reviewer 2, the fractionation in lakes might differ spatially for various reasons. Our estimates are based on field samples (that might include impurities) and are not from controlled experiments. As we pointed out in the previous responses, the data from Brett et al. (2018) should not be used without recalibrating the original values when needed (Soto et al. 2017). And as Reviewer 2 points out, the spatiotemporal variance we used also incorporates these possible differences in the fractionation values. However, the suggestion of perfect correlation between water and algal values in our models is wrong as we directly incorporate these uncertainties into our Bayesian mixing models. In fact, the uncertainty we use (s.d. of $\pm 23.6\text{‰}$) is greater than that which Reviewer 2 suggests. Sum of random variables means that $\pm 17.9\text{‰}$ plus $\pm 8.7\text{‰}$ s.d. used by Reviewer 2 combines to give an s.d. of 19.9‰ , less than the 23.6‰ we include in our models. Thus, this situation is already accounted for and is further discussed in the caveats section of our manuscript (L603-612).

If the perfect correlation between the ambient water and modeled algal d2H values did not affect the mixing model outcomes, this point would matter less. But since ambient water and algal d2H values are two of the five variables needed to estimate allochthony in aquatic consumers (with the other variables being the terrestrial resource and consumer d2H values, and the assumed environmental water contribution), it is likely that this quirk of Keva’s et al.’s modeling approach also impacts the outcomes. During the process of reviewing the third permutation of this MS, I used a classic algebraic mass balance approach to reanalyze all of the Keva et al.’s consumer allochthony estimates. When doing this I noticed that the ambient lake water d2H values were one of the best predictors of allochthony in each of the 19 consumers that Keva et al. sampled and analyzed (see Table on page 3). Intuitively, this makes no sense. Ambient water d2H values were on average a 4% weaker predictor of allochthony than was PC1 ($r^2 = 0.282$ vs 0.323 , respectively) and a 7% better predictor than forest area ($r^2 = 0.282$ vs 0.214 , respectively). Ambient water d2H values were also a better predictor than agricultural area. This is quite meaningful because one of the main conclusions of the Keva et al. MS is that forest/agricultural land cover determines the degree to which consumers

in their lakes are supported by terrestrial organic matter. The statistical results summarized below suggest the $\delta^2\text{H}$ value of the ambient water is a substantially better predictor. But how could this be? Why would the $\delta^2\text{H}$ values of ambient water be related to the estimated allochthony in these lakes? Surely the deuterium ratio of ambient water would have no mechanistic influence on whether consumers utilize more or less t-OM in these lakes! Conversely, the $\delta^2\text{H}$ value of real algal in these lakes should be important because a more negative algal $\delta^2\text{H}$ value in these calculations will directly lead to a lower allochthony estimates in the consumers. By artificially creating a perfect correlation between the ambient water and algal $\delta^2\text{H}$ values in their mixing models, I believe the outcomes of these calculations are substantially influenced. Finally, because the ambient water $\delta^2\text{H}$ values are also moderately correlated with the forest and wetland areas, pH, etc. in these lakes, one could create a spurious relationship with the “modeled” algal $\delta^2\text{H}$ values. Perfectly correlating the $\delta^2\text{H}$ data for ambient water and algal data may be part of the cause of the environmental gradient response that Keva et al. reported for their lakes. This conclusion may simply be a result of the authors’ unorthodox approach for obtaining algal $\delta^2\text{H}$ values. The only way to resolve this conundrum is to rerun the mixing models with the directly determined benthic algae $\delta^2\text{H}$ values, and to provide the raw data for the algal $\delta^2\text{H}$ values.

Response: Far from making no sense, it should be obvious that in any mixing model system, a non-zero contribution of a source results in some negative correlation between the tracer value of that source and its proportional contribution due to the inverse relationships ($P_{S1} = \frac{Cons-S2}{S1-S2}$) in systems where the source values vary, in our case across lakes. This is the case in Reviewer 2’s table, which shows an average negative correlation of $r = -0.44$ across species. There should be no confusion that ambient water affects the omega-corrected consumer $\delta^2\text{H}$ values and thus also the allochthony, and the ambient water is included in the omega correction formula. Moreover, as we obtain the aquatic $\delta^2\text{H}$ signature for each lake by adding the photosynthesis fractionation value to the lake water value, the lake water $\delta^2\text{H}$ values have a direct link to the aquatic source $\delta^2\text{H}$ values we used in the models and thus to the allochthony estimates as well. Thus, Reviewer 2’s argument about correlated water $\delta^2\text{H}$ values and consumer allochthony is circular reasoning. However, we have now provided results from a model with actual measured benthic algal $\delta^2\text{H}$ values as the aquatic source (Fig R2, also Supplementary Figure 9), and the results are essentially the same (see earlier response to comments above).

8. The Bayesian algorithm distorts information contained in the raw data!

One of my most important points in the previous rounds of review is that the problem that Keva et al. are working on is a very simple mass balance calculation that could easily be taught to any undergraduate student in a STEM field. This mass balance problem is based on the Law of Mass Conservation, one of the fundamental principles of physics and chemistry. By using a Bayesian algorithm to analyze these data, the outcomes are often distorted in ways that contradict this law. Most importantly, this is completely unnecessary. During the process of reviewing the paper a third time, I reanalyzed all of the Keva et al. data using the correct algebraic approach. In the plot below, I plotted four cases that represented high (littoral benthic macroinvertebrates), intermediate-high (ruffe), intermediate-low (large roach), and low estimated allochthony (small perch). In all cases, I used the exact same assumptions as specified by Keva et al. So, the only difference between the outcomes I obtained and those reported by the authors should be solely due to using an algebraic versus Bayesian approach. As the plots show, the differences in the two approaches are often profound.

Response: As we have repeated frequently throughout this review process, the restriction of proportions to the domain of [0,1] is not a distortion, but a true reflection of reality. The very simple algebraic approach Reviewer 2 proposes does not adhere to this fundamental principle, and therefore gives results that are biologically infeasible. The Bayesian statistical framework merely returns a probability distribution of the parameters of the model that is compatible with the observed data. In our case, our model structure assumes (1) that mass is conserved during mixing of sources to consumer, (2) that all mass originates from either terrestrial or aquatic sources, (3) that mixing proportions must sum to unity and fall in the domain of [0,1], and (4) that sources, fractionations and drinking water contributions have values with quantified uncertainty. The contradiction that Reviewer 2 points to is reconciled by the fact that his approach also does not consider the various and often substantial variations in tracer values that may arise, mentioned throughout. This holds for his reanalysis and therefore it is incorrect for Reviewer 2 to state that what he shows are based on the exact same assumptions. As almost no details are provided, it is difficult for us to provide specific comments. However, it appears to us that Reviewer 2's reanalysis implies the same trend that is the main finding of our manuscript. While the consumer allochthony values might change to some extent, this is not the main thrust of our paper. In fact, the directions of the trends in relation to the key environmental gradient are the same, and this is the most important finding of our study.

I also created a histogram that plotted the estimated allochthony values for all 19 consumers ($n = 1,737$) (see below). When analyzing the data using an algebraic approach it is clear that a quite large portion (i.e., 44%) of the allochthony estimates were in the negative domain, and 2.3% of the estimates were in the greater than 100% terrestrial resource utilization domain. The overall median estimate for their dataset was 8.5% terrestrial resource utilization. This overall estimate would shift to zero if the environmental water assumption was only slightly increased.

Response: We thank Reviewer 2 for sharing this. As discussed many times in previous responses, we prefer using the Bayesian approach with this data set as it performs better with relatively low samples sizes across the many factor levels that need to be accounted for. It is worth noting that many of the species by lake populations have posterior allochthony estimates of near-zero. The last point is not true; we have provided the omega sensitivity analysis already in the previous rounds and from there one can see the median allochthony of consumers does not drop to zero if modifying the omega value to be slightly higher. However, we think that increasing the omega value by 9% units is rather larger than a slight modification.

9. What is the proper interpretation of a negative allochthony estimate?

In their response, Keva et al. stated "taking the argument of Reviewer #2 that all negative proportions are true zeros, in other words the codomain is bounded to biological feasibility, then you can no longer perfectly reconstruct all consumer isotope data as the inverse function is no longer injective – the value of 0% allochthony in their example would now map to many observed consumer δ^{2H} values." The authors have completely misunderstood my point. A small negative allochthony estimate (e.g., -5%) could easily arise due to sampling or measurement error. However, the large portion of strongly negative allochthony estimates indicates that there is a serious problem with Keva et al.'s dataset. Most likely the δ^{2H} data for the hypothetical resources are wrong. This is not too surprising given the very indirect way that these authors estimated the δ^{2H} values for the terrestrial and aquatic resources in their mixing model. As a famous quote goes "On two occasions I have been asked, 'Pray, Mr. Babbage, if you put into the machine wrong figures, will the right answers come out?' I am not able rightly to apprehend the kind of confusion of ideas that could provoke such a question" (Charles Babbage). This statement is as true today as it was 150+ years ago. Ironically, in

their reply Keva et al. made nearly the same point when they stated “If the “estimation, observational and measurement error” of these data would be so great as to alone cause such discrepancies, then clearly the data are not suitable to be analysed in such an approach.”

Response: As clearly explained in the Methods section (L492-498), most of the samples fell inside the mixing envelope and we trust the method is suitable for analysing consumer allochthony. We would like to reiterate that the models converged well and the epsilon values for $\delta^2\text{H}$ values were well below 1. We disagree with the assertion that the issue of highly negative values that Reviewer 2 estimates with his approach lies with our dataset, but which in fact stems from the inability of his overly simple algebraic approach to consider any uncertainty in sources or other parameters (e.g. omega values, trophic positions) that are well-known in natural systems. The implicit assumption of his point-estimate approach is of perfect knowledge of all system parameters and that any variation in consumer isotope values can only arise due to changes in proportional allochthony contributions. Clearly this is a gross assumption to make.

In relation to the quote by Charles Babbage given by Reviewer 2, we would highlight that the machine being referenced, Babbage’s difference engine (the earliest form of calculator), was based on “the unerring certainty of machinery” (Charles Babbage). These machines or models function solely on perfect knowledge of the system. Reviewer 2 insinuates that the biologically infeasible results he obtains using our data must be the result of our data being wrong. We would posit, as we have raised many times throughout this review process, that Reviewer 2 is simply using the wrong machine for the wrong job. Reviewer 2 has further misconstrued our point regarding observational and measurement error. We were highlighting that under their presumption of perfect knowledge, then the results would suggest the data would not be considered suitable using their approach. However, as we do not assume perfect knowledge with the Bayesian framework, we obtain estimates that reasonably hold with the natural law of dietary domain [0,1]. It is also worth highlighting here the very subjective nature of what can be considered a ‘small negative’ allochthony that would be acceptable under Reviewer 2’s approach.

The problem with a Bayesian approach is it completely obscures obvious problems with model misspecification by rounding all estimates to the 0-100% domain. In the plot below, I present the estimated allochthony values for the classic algebraic approach as well for an approach where all estimates are rounded to fall within the 0-100% range. These outcomes are presented for medium perch (see following plot) because this consumer had nearly the median overall estimated allochthony for the 19 consumers as well as a large sample size ($n = 174$). As this comparison shows, the original raw predictions (see left-side panel) show an overall distribution that hovers close to a mean of zero (i.e., $1 \pm 42\%$, ± 1 SD). Conversely, when the predictions are constrained to 0-100% (see right-side panel) the overall response seems compelling with a mean of $17 \pm 23\%$ and an interesting response pattern with the Environmental gradient represented by PC1. At low values of PC1, the estimated allochthony was 20-60% and at high values of PC1 the estimated allochthony was 5-10%. I will also note that the “adjusted” plot looks very similar to the outcome Keva et al. reported for medium perch. However, the scatter around the trend line and their within lake error bars are much smaller than in my adjusted plot, which is hard to reconcile since the uncertainty shown in my adjusted plot is only due to the within lake variation in point estimates. Since the Keva et al. estimates supposedly also took into account uncertainty the resource stable isotope values, the prediction error for their approach should have been substantially larger than for my approach. Most likely the small scatter around the trend line and smaller error bars for the Keva et al. plot are because these authors crossed the outcomes across “lakes for the same species, and across species from the same lake”.

My overall point is that the superficially convincing results presented in Keva et al.'s Figure 5 (see results for medium perch above) are largely due to a quirk of the Bayesian algorithm that misrepresents the true results of the mass balance calculations by forcing all of the outcomes into the 0-100% domain. It is also problematic that the uncertainty depicted in Keva et al.'s plots was much less than what I found when reanalyzing their data. The true relationship between the Environmental gradient (PC1) and the estimated consumer allochthony is much more ambiguous than they claim because of the large portion of negative estimates.

I want to use another example based on the Keva et al. data to clarify this point. Their data indicated very low allochthony estimates for five consumers (Chaoborus, Copepods, small roach, small perch, and smelt). The average estimate for these consumers combined is -28 ± 39 allochthony, $n = 405$. For these examples, the algebraic analysis is 100% correct, but it suggests a biological outcome that is not possible! We all agree that consumers cannot obtain -28% of their resources from terrestrial sources. What this outcome does show is that in > 70% of these cases, there is something seriously wrong with the data used to calculate allochthony for these consumers. Either the trophic levels assigned, or the environmental water value assumed, or the ambient water, consumer, and aquatic and terrestrial d2H values used in the calculations are wrong. I am guessing the ambient water and consumer d2H values are solid. I suspect the algal and terrestrial d2H values are the main source of this problem. Knowing that these data give unreasonable outcomes in many cases is very important information.

Response: The MixSIAR framework models proportions using a Dirichlet distribution, which satisfies the de facto requirements of proportions summing to unity plus values being constrained to the [0,1] domain, i.e. proportion vectors lie on the standard simplex and are modelled on that simplex in an appropriate manner (an isomorphism that satisfies linearity) due to the non-Euclidean geometry. This is not simply rounding as Reviewer 2 suggests. Regardless, we have taken care to ensure that within the bounds of variation and uncertainties, our consumer data fall within the mixing envelope as is standard practice. As pointed out in many previous revision rounds and in the Methods section, almost all of our consumer samples (>98%) fell inside the mixing envelope, therefore we do not believe this is a major issue.

Reviewer 2 does however point to the crux of the issue with regards to his approach: "problems with model misspecification". In his simple algebraic manner, he assume perfect knowledge of the system, all parameters (including sources, omega values etc.) have an exact single point estimate, and therefore any and all variations in isotope values arise from changes in allochthony proportion alone. Clearly, we do not have perfect knowledge of the system and single exact point estimates for parameters are a gross simplification. For example, trophic position estimates are going to be uncertain due to variations in diet and isotope fractionations between individuals that are known to occur in real systems, and the same is true for omega values. Likewise, source isotope values show spatio-temporal variation within lakes and therefore consumers are unlikely to have the same exact source values even within the same lake due to e.g. mobility of consumers. Reviewer 2 points to these uncertainties in his own comment: "Either the trophic levels assigned, or the environmental water value assumed, or the ambient water, consumer, and aquatic and terrestrial d2H values used in [my] calculations are wrong." Wrong in the sense that they are not and should not be considered single point estimates with perfect knowledge. The solution is to use a modelling framework that permits the specification of these uncertainties, which is why we use a Bayesian mixing model approach that provides probability distributions of parameters that could generate the observed data given the model structure. This is not the case for Reviewer 2's simple algebraic approach and hence why it cannot be taken for their approach to be considered providing the "true" values or

“100% correct”. As we have mentioned previously, as latent variables they are unknown and unmeasurable, and therefore cannot be directly verified or validated.

Reviewer 2 points to small scatter and errors around our proportion values as compared to his own analysis and suggests that, due to our model accounting for uncertainty, our outputs should be more uncertain than his own. This is not true. Firstly, he has not specified what his error bars represent so it is unclear whether they are at first directly comparable. Secondly, incorporating uncertainty within our models provides realistic mechanisms for the generation of variability within consumer isotope values even if allochthony proportions are fixed – e.g. stochastic feeding of consumers giving rise to variation in realised source isotope values at the individual level. The algebraic approach pushed by Reviewer 2 can only generate consumer isotopic variation by varying allochthony, hence his relationships are more ambiguous. We would like to point out that similar relationships have been found in many other systems and species, some using a frequentist and some using a Bayesian approach (Keva et al. 2022; Tanentzap et al. 2017). Thus, we are confident that the reported relationship between lake/catchment characteristics and consumer allochthony are true. Here, we would like to point out that we did not receive a script for the Reviewer 2’s calculations, so we cannot estimate whether the calculations are correct and did Reviewer 2 use the same assumptions as us.

Recalculated environmental water (ω) estimates

The authors also need to fully detail the protocol that they used to revise the published environmental water values for their analysis. It seems somewhat fortuitous that they originally assumed an environmental water value of 0.22 and after I noted that that value was based on the erroneous results from Wilkinson et al. (2015), Keva et al. revisited this topic and found a correction was necessary for the average of 0.27 ± 0.11 that I published in Brett et al. (2018) and that this correction resulted in a new estimated environmental water value (i.e., 0.23 ± 0.09) that was only slightly different from the value that they originally used. I am no expert on the detailed intricacies of how $\delta^2\text{H}$ is quantified with mass spectrometry, and how standards should be employed, etc. But if Keva et al. are going to argue that the best available estimate for environmental water is now based on their reanalysis of the original data (i.e., 0.23 ± 0.09), they need to very carefully explain why some studies were included in their reanalysis and others were excluded. This is done only briefly in the notes for Supplementary Table 9, but not nearly in sufficient detail for someone unfamiliar with this topic. For example, the inclusion of the estimate from Macko et al. (1983) which did not use standards, and exclusion of the estimates from Graham et al. (2014) which did use standards, both had the effect of lowering the recalculated environmental water estimate. If Macko et al. (1983) was excluded and Graham et al. (2014) was included the revised estimate would have been 0.27 ± 0.09 . Why is it necessary to use water labeling experiments to calculate ω ? I am admittedly not an expert on this, but it would seem it would only necessary to know the $\delta^2\text{H}$ values of a pure defined diet, the ambient water and the consumer to experimentally determine the environmental water contribution. Additionally, Keva et al. need to specify exactly which equations from Soto et al. (2017) they used to recalculate the previous environmental water estimates. Without a clearly described protocol, the recalibration process seems quite mysterious and even coincidental.

Response: We thank Reviewer 2 for this comment. The $\delta^2\text{H}$ water gradient in Graham et al. (2014) consists of only 2 points, and thus is not suitable for the updated omega table. A wider range in ambient $\delta^2\text{H}$ water values would make the omega estimations much more reliable; in the other omega studies water labelling with a wide range of water $\delta^2\text{H}$ was done. Moreover, it is clear that Macko et al. (1983) did not need these keratin standards as they analysed the sample $\delta^2\text{H}$ values directly against $\delta^2\text{H}_{\text{VSMOW}}$. In addition, based on Soto et al. (2017), it is trivial that the formula we used

for the $\delta^2\text{H}$ recalibrations is from Table 2 (correction equation a) in Soto et al. (2017), we have now added this information also to the Table S9. The protocol and reasoning for recalibration is well described in Soto et al. (2017) and we do not see why Reviewer 2 is claiming otherwise. But based on this, any $\delta^2\text{H}$ that were run with wrong standard values should not be used without recalibration; this is certainly an issue in past $\delta^2\text{H}$ studies, including the Brett et al. (2018) review paper.

Access to all raw data

I very much appreciate that the authors have provided access to most of their raw data during the review process. If this MS is finally published in Nature Communications, or elsewhere, I will officially request access the directly determined benthic algal d2H data for the 35 lakes sampled for this study. I will also officially request the original uncorrected consumer d2H data, as well as the predicted d2H values for these consumers from their validation analysis.

Response: The original uncorrected consumer $\delta^2\text{H}$ data were already uploaded to the repository and provided to the Reviewers. We have now added the benthic algal $\delta^2\text{H}$ values to the data repository. We would like to point out that the scripts for validation calculations were already stored to the repository. The original model objects will be made public through GitHub “releases” feature once the manuscript has been accepted for publication.

Minor points

8. The authors conceded my point that herbivorous zooplankton like Daphnia, as well as the BMI Asellus, most likely have a trophic position of 2.1. If that is true, then a zooplanktivorous fish like a smelt or a vendace that consumes Daphnia would have a trophic position of 3.1, and a pike that consumes smelt/vendace would have a trophic position of 4.1. The authors should be consistent in this regard.

Response: We used the trophic level estimation described in the manuscript, based on the $\delta^{15}\text{N}$ values of the primary producers and the consumers (L452-469). We are grateful for Reviewer 2 highlighting that some of these estimates produced biologically unrealistic, low TP values for a small subset of the consumers and we have now accounted for that. However, we disagree with Reviewer 2’s suggestion of fixed TPs for consumer groups. Intraspecific variation in trophic positions are very well documented, not least by having analysed the fish stomach contents within this study. We believe that the trophic level estimations based on $\delta^{15}\text{N}$ values provide an efficient tool to evaluate the long-term trophic position of individual consumers, which is rather a continuum than a fixed stepwise increasing feature. We think that using trophic position estimates based on $\delta^{15}\text{N}$ values is more appropriate than that the rigid presumed relationships suggested by Reviewer 2.

9. Fig. S7 appears to show some Cladocera are still assumed to have a trophic position somewhat below 2.0, which is not physiologically possible because Cladocera are not mixotrophic as discussed in my previous review. This appears to also be the case for copepods, profundal chironomids, bulk profundal BMI, littoral chironomids, bulk zooplankton, littoral Asellus, and bulk littoral BMI. Maybe this is just an optical illusion? As previously discussed, it is simply not physiologically possible for any of these consumers to truly have trophic positions below 2.0.

Response: The way we are defining the trophic level (i.e., as a continuous measure of trophic position) in relation to fixed pelagic and benthic baselines is clearly described in the Methods section. We acknowledge that some consumers may appear to be marginally below 2, which of course is theoretically impossible, and this may arise because some consumers straddle the pelagic

and benthic food webs. Nevertheless, we do not think this is a major issue as clearly most of the consumers trophic positions accord well with general ecological knowledge.

10. Keva et al. did not always use a photosynthetic hydrogen discrimination value of -106.7‰ when generating “modeled” algal $\delta^2\text{H}$ values as stated in the MS. In lake Jyväsjärvi they used a photosynthetic hydrogen discrimination value of -109.6‰. Jyväsjärvi was the most sampled lake in the dataset, with about 3X more observations than the other lakes on average.

Response: We thank Reviewer 2 for noticing this. For Jyväsjärvi and Sääksjärvi, the water $\delta^2\text{H}$ values in the lake water $\delta^2\text{H}$ data frame inadvertently represented the average of all analysed samples from different seasons rather than just summer values, and this is the origin of the observed difference in used fractionation values. Moreover, we noticed that we had unintentionally used in some cases average lake water and inlet water $\delta^2\text{H}$ values to build the lake water $\delta^2\text{H}$ data frame, clearly a mistake from our side. We have now corrected this and used only lake water samples taken from central part of the lakes from summer for obtaining the photosynthetic fractionation of $\delta^2\text{H}$. The corrected fractionation value is 109.7‰, which is not much different to the previous but does align better with the recalibrated fractionation values from Brett et al. 2018 (please note that in the main text we have rounded the cited value to 110‰). Moreover, we used the updated lake water $\delta^2\text{H}$ values in the consumer $\delta^2\text{H}$ omega correction calculations (updated in the repository). Although rerunning all the analyses did not change our results, we are happy that Reviewer 2 noticed this issue and enabled us to put it right.

11. In their response, Keva et al. stated “The last point of this comment 7 is completely incorrect; we explain in the manuscript: “In four lakes (Hirvijärvi, Niemisjärvi, Pankajärvi and Valkea-Kotinen), the cladoceran $\delta^{15}\text{N}$ values were lacking and thus they were derived from a linear model based on $\delta^{15}\text{N}$ values of cladoceran and bulk zooplankton samples from the other study lakes (adjusted $R^2 = 0.928$, $p < 0.0001$, $\delta^{15}\text{N}_{\text{cladocera}} = 0.872 \times \delta^{15}\text{N}_{\text{bulk zooplankton}} - 0.095$),” which Reviewer #2 may have missed while reviewing our manuscript.” Keva et al. are 100% correct here. I was mistaken, this was properly described in the MS.

RESPONSE: We thank Reviewer 2 for clarifying this, and for all his constructive and helpful comments during the revision rounds.

Brett, M.T., G.W. Holtgrieve, and D.E. Schindler. 2018. An assessment of assumptions and uncertainty in deuterium-based estimates of terrestrial subsidies to aquatic consumers. *Ecology* 99: 1073-1088.

Fogel, M. L., Griffin, P. L., & Newsome, S. D. (2016). Hydrogen isotopes in individual amino acids reflect differentiated pools of hydrogen from water and food in bacteria. *Proceedings of the National Academy of Sciences USA*, 113, E4648–E4653.

Osburn, M. R., Dawson, K. S., Fogel, M. L., & Sessions, A. L. (2016). Fractionation of hydrogen isotopes by sulfate-and nitrate-reducing bacteria. *Frontiers in microbiology*, 7, 196727.

Soto, D. X., Koehler, G., Wassenaar, L. I. & Hobson, K. A. 2017. Re-evaluation of the hydrogen stable isotopic composition of keratin calibration standards for wildlife and forensic science applications. *Rapid Commun. Mass Spectrom.* 31, 1193–1203.

Wilkinson, G. M., Cole, J. J. & Pace, M. L. 2015. Deuterium as a food source tracer: Sensitivity to environmental water, lipid content, and hydrogen exchange. *Limnol. Oceanogr. Meth.* 13, 213–223.

Fig R1 Consumer allochthony models with modelled bacterial $\delta^2\text{H}$ values as terrestrial source.

Modelled bacterial $\delta^2\text{H}_{\text{bact}}$ values were obtained with following equation: $(\delta^2\text{H}_{\text{bact}} = \delta^2\text{H}_{\text{inlet DOM}} \times (1 - \omega) + \delta^2\text{H}_{\text{lake water}} \times \omega)$ where the used omega value was 0.17. The modelled phytoplankton values were used as aquatic source for the consumers. The bold colored lines and the light ribbon areas indicate the median and 95% credible intervals, respectively, of the Bayesian estimates of consumer resource use across lake catchment forest coverage (%) gradient. Lake- and consumer-specific estimates of allochthony are marked with brown dots and shaded lines, respectively. Zooplankton (a), zoobenthos (b), and fish consumers (c-d) are divided into different columns. Consumer group names are presented for each subplot in the corresponding header, where text inside the brackets indicate habitats (a-b) or feeding guilds (c-d).

Fig R2. Consumer allochthony models with sampled benthic algae $\delta^2\text{H}$ values as aquatic source.

The bold colored lines and the light ribbon areas indicate the median and 95% credible intervals, respectively, of the Bayesian estimates of consumer resource use across lake catchment forest coverage (%) gradient. Lake- and consumer-specific estimates of allochthony are marked with brown dots and shaded lines, respectively. Zooplankton (a), zoobenthos (b), and fish consumers (c-d) are divided into different columns. Consumer group names are presented for each subplot in the corresponding header, where text inside the brackets indicate habitats (a-b) or feeding guilds (c-d).

Review of NCOMMS-23-44093-T "Land use drives terrestrial support of boreal lake food webs"

When reviewing this paper, I focused almost entirely on the methods these authors employed. I will also note that in 2018 I published a paper on how the results of deuterium-based food web analyses are very strongly influenced by several poorly constrained assumptions (Brett et al. 2018). Nearly all of the problems noted in that analysis are apparent in this manuscript. I will detail my main concerns below.

1) As noted in Brett et al. (2018), deuterium-based food web analyses are extremely sensitive to the assumed environmental water (ω) value. Environmental water is the proportion of a consumer's hydrogen content obtained directly from water, as opposed to its diet. Brett et al. (2018) showed that when the environmental water contribution to consumers was directly measured in laboratory experiments (where it can be easily quantified), it averaged 0.27 ± 0.11 (± 1 St. Dev). The authors of NCOMMS-23-44093-T used an environmental water assumption of 0.22. In the MS they state "*We used an omega value of 0.22 based on the best literature assessments (Wilkinson et al. 2015)*". This claim is highly dubious because the analysis by Brett et al. (2018) showed the Wilkinson et al. (2015) study had a series of egregious errors in their compilation of omega values (see pages 1081-1082 of Brett et al.). Most importantly, more than half of the estimated omega values reported by Wilkinson et al. were obtained from open systems where it is impossible to actually quantify omega because you cannot know what they actually consumed in an open system. Omega can only be quantified in closed systems where only a single diet item is used (and therefore the diet, water and consumer $\delta^2\text{H}$ values are exactly known). Secondly, Wilkinson et al. (2015) reported their summary of omega estimates had a standard deviation of 0.02. However, even the most cursory examination of the data these authors presented in their Table 2 and Figure 1 make it immediately obvious that the true standard deviation of these data is much larger, i.e., ± 0.11 . Thirdly, Wilkinson et al. claimed that the average omega estimates from open and closed systems were not significantly, when clearly they were (unpaired t-test; $t = 2.93$, $df = 22$, $P < 0.01$). Finally, Wilkinson et al. (2015) also included two estimates of omega from a study that didn't even measure this parameter (Jardine et al. 2009). Based on this remarkable series of errors, one cannot possibly claim that the estimated omega values from Wilkinson et al. (2015) represent the "*best literature assessments*". Ironically, if the numerous errors in Wilkinson et al. are corrected, the omega values they summarized from closed systems had a mean \pm SD of 0.28 ± 0.10 , which is nearly identical to the results from Brett et al. (2018), i.e., 0.27 ± 0.11 . Since the authors of NCOMMS-23-44093-T cited Brett et al. (2018) when they noted "model outputs of consumer allochthony are known to be sensitive to the selected omega (ω) values", I also suspect they are well aware of the raft of errors in Wilkinson et al. (2015).

Why do the numerous errors in Wilkinson et al. (2015) even matter? By claiming that the Wilkinson et al. (2015) environmental water estimate compilation represents the "*best literature assessments*" the authors of NCOMMS-23-44093-T can use a substantially lower and less variable assumed omega value than indicated by the actual published literature. Specifically, the authors of NCOMMS-23-44093-T used a point estimate of omega = 0.22 for their main conclusions and a range of omega = 0.10 to 0.30 values for their sensitivity analysis.

It is also noteworthy that in their sensitivity analysis reduced the point estimate of omega by 0.12 but only increased it by 0.08. As expected, their reported sensitivity analysis showed their estimated terrestrial contributions to the consumer hydrogen content was very dependent on the environmental water assumption used for their analyses. I believe very strongly that the only defensible point estimate and range for omega at this time is that based on the results summarized in Brett et al. (2018), i.e., $\omega = 0.27$ with a range of 0.16 to 0.38 for the sensitivity analysis.

2. The use of Bayesian mixing models is not warranted, and could in fact greatly obscure problems in the underlying dataset. People in the scientific community are often in awe of complex models, and the Bayesian dietary mixing model MixSIAR certainly fits this profile. The problem with any of the popular Bayesian mixing models is that they will always provide a solution within the 0-1 domain no matter how inappropriate the underlying data are. For example, the consumer being analyzed could have stable isotope values much higher or lower than either of the terrestrial or aquatic resources considered in the model. In these cases, the underlying model is obviously mis-specified, for example, the consumer may be utilizing a resource not included in the model, the stable isotopes values used to represent the consumer and resources may be wrong, or the assumed omega value could be incorrect. These sorts of problems get entirely brushed under the rug by the popular Bayesian mixing models.

In fact, the calculations needed to estimate terrestrial contributions to aquatic consumers are very simple algebraic mass balance expressions (see equations 1 and 2 in Brett et al. 2018), thus there is no justification for not dealing with these in their original unadulterated form. If you analyze the data using the original equations you will readily see cases where the model is mis-specified as calculated outcomes that are substantially more than 100% or less than 0% for either the terrestrial or aquatic resources.

The authors justify their Bayesian analyses on lines 75-77 when they state “Most (~99%) of the sampled consumer $\delta^2\text{H}$ values fell inside the lake-specific range of $\delta^2\text{H}$ values of basal resources (inlet dissolved t-DOM \pm 2SD & algae \pm 2SD: Fig. 2) as required by the model.” This means the observed consumers (after environmental water correction) could be as much as 22-48 $\delta^2\text{H}$ outside of the envelop described by the resource stable isotope values. I don’t have access to these authors raw data, but that seems like a very large disparity and seems to set the bar very low for model mis-specification.

I was curious how much error this level of uncertainty could introduce to their calculations. Since I don’t have access to the raw data from NCOMMS-23-44093-T, I based my sensitivity analysis on the mean $\delta^2\text{H}$ values for zooplankton, water, t-POM, and phytoplankton from the Tanentzap et al. (2017) study as reported in Table 6 of Brett et al. (2018). I also used the mean zooplankton trophic level from Tanentzap et al. I then used the assumed omega value (= 0.22) from NCOMMS-23-44093-T, as well as the \pm 2SD values for DOM and benthic algae from their Table S3. My cursory sensitivity analysis indicated adding \pm 2SD to the resource stable isotope values could introduce tremendous uncertainty to the estimated allochthony values. For the

conditions described above, I calculated allochthony values from as high as +48% to as low as -223% using the algebraic approach.

3. Another closely related problem with Bayesian mixing models is that messy (or highly variable) data generally give you the uninformed prior assumption as the model output (Brett 2014), which in the case of NCOMMS-23-44093-T would be 50% terrestrial and 50% aquatic resource utilization. Since the original algebraic assumptions don't assume a prior, they are not subject to this bias towards all resources being equally important. When adding a lot of uncertainty to the potential resource stable isotope values, you are also greatly increasing the likelihood that the Bayesian model will just return the prior assumption.

4. One of the most concerning things about this study is just how little of it is based on actual field data for the variables of interest. For example, on lines 503-504 the authors state "For each lake, we had individually analysed two sources: inlet t-DOM and modelled algae (aquatic)." This is problematic because zooplankton, benthic invertebrates and the other consumers in their lakes do not in fact consume t-DOM! They directly consume t-POM. That is the invertebrate and vertebrate consumers in lakes directly utilize particulate organic matter NOT dissolved organic matter. Why didn't the authors directly determine the stable isotopic values for the t-POM? Are these authors assuming t-POM and t-DOM have same hydrogen stable isotope values? If so, they need to provide some support for this assumption. Similarly, lines 503-504 of the MS state they used "modelled algae" for their analysis. This part is confusing because elsewhere in the MS I got the impression that they actually used observed benthic algae values and modeled phytoplankton values. This also needs to be explained more clearly. Overall, it is problematic that apparently only two of the five variables needed for the authors' analyses were directly determined. Specifically, the $\delta^2\text{H}$ values for the lake water and consumers were directly determined, but the $\delta^2\text{H}$ values for the terrestrial and aquatic resources were estimated in some way, and the omega values were assumed. Similarly, the resource standard deviation values used in the Bayesian modeling are not based on observations from the actual lakes, rather they are based on observations for a subset of 4 of the 35 sampled lakes. Given the dearth of real data in this study, it is hard to imagine the estimated allochthony values for these lakes are even remotely similar to the true values.

Minor points:

5. Why were cladoceran zooplankton assumed to have a trophic level of 2 (which implies they are completely herbivorous)? It well established in the zooplankton feeding literature that cladocerans also feed on microzooplankton. It is conceivable that cladocera have a slightly higher trophic level (e.g., 2.1)? The strict herbivory assumption needs to be justified or modified.

6. The expectations outlined in lines 82-89 of the MS line up almost perfectly with the reported results from this study. This begs the questions, were the authors of this study very good at predicting its outcome, or where the expectations modified after the results were known?

7. Lines 526-529, as noted previously using “high variance in source values” won’t actually account for seasonal fluctuations (since these are unknowable based on a single sampling date), but it will make it much more likely that a Bayesian mixing model will just give you the prior assumption (Brett 2014).

8. I disagree with the authors’ conclusion on lines 531-533, i.e., “As our consumer data were collected during the summer growing season when autochthonous algal production is highest, our zooplankton allochthony values are likely underestimating the annual allochthony of these pelagic taxa.” In fact, in temperate or boreal lakes the vast majority of zooplankton production occurs during the summer because this is when zooplankton community biomass is greatest and the water is warmest and therefore metabolic rates are the highest (Zhang et al. 2022).

9. Lines 538-539, why did the authors use an asymmetrical range of omega values for their sensitivity analysis. This seems like tilting the scales.

Signed, Michael T. Brett

Brett, M.T. 2014. Resource polygon geometry predicts Bayesian stable isotope mixing model bias. *Marine Ecology Progress Series* 514, 1-12.

Brett, M.T., G.W. Holtgrieve, and D.E. Schindler. 2018. An assessment of assumptions and uncertainty in deuterium-based estimates of terrestrial subsidies to aquatic consumers. *Ecology* 99: 1073-1088.

Jardine, T. D., K. A. Kidd, and R. A. Cunjack. 2009. An evaluation of deuterium as a food source tracer in temperate streams of eastern Canada. *J. N. Am. Benthol. Soc.* 28: 885–893.

Tanentzap, A. J. et al. 2017. Terrestrial support of lake food webs: Synthesis reveals controls over cross ecosystem resource use. *Sci. Adv.* 3, e1601765.

Wilkinson, G. M., Cole, J. J. & Pace, M. L. 2015. Deuterium as a food source tracer: Sensitivity to environmental water, lipid content, and hydrogen exchange. *Limnol. Oceanogr. Meth.* 13, 213–223.

Zhang, C., M.T. Brett, J.M. Nielsen, G.B. Arhonditsis, A.P. Ballantyne, J.L. Carter, J. Kann, D.C. Müller-Navarra, D.E. Schindler, J.D. Stockwell, M. Winder, D.A. Beauchamp. 2022. Physiological and nutritional constraints on zooplankton productivity due to eutrophication and climate change predicted using a resource-based modeling approach. *Can. J. Fish. Aquat. Sci.* 79 (3), 472-486.

Main points

1. I want to start by thanking the authors for making their raw data available for the review process. I also want to thank them for pointing out that the true distribution of environmental water values may have a mean of 0.23 ± 0.09 , as opposed to the 0.27 ± 0.11 reported in Brett et al. (2018). I have not yet had the time to dig into the various papers by Soto et al. to verify this claim, but I will do that when I get the chance.
2. The authors disclosed an important detail about their analysis that was not previously apparent. In the MS that I first reviewed the authors stated they represented the $\delta^2\text{H}$ values of terrestrial resources with the $\delta^2\text{H}$ values of DOM in lake inputs. I assumed that they did this because they were using the deuterium values of the terrestrially derived dissolved organic matter (t-DOM) to represent the $\delta^2\text{H}$ values of terrestrially derived particulate organic matter (t-POM). But in their response to my comment the authors noted their approach assumed t-DOM was in fact the form of terrestrial organic matter that they hypothesized contributes to zooplankton and fish production in their lakes. This was not clearly explained in the methods of the initial draft that I reviewed (and it is still not explained in the methods section of the most recent draft). The authors also noted that some previous studies also hypothesized that t-DOM was the main source of terrestrial organic matter contributions to consumers in lakes. However, they failed to note that the highly cited paper by Cole et al. 2006 concluded the t-DOM pathway only accounted for ca. 2% of zooplankton production in the lakes they studied.

More importantly, Keva et al. did not correctly take into account the impact of a t-DOM to bacteria to protozoa pathway on their deuterium based calculations. If the pathway truly does entail t-DOM to bacteria to protozoa, then the zooplankton in their lakes are three trophic levels above the terrestrial resources, and not only one trophic level as their modeling approach assumes. This has important implications for the assumed contributions from environmental water to zooplankton hydrogen content for this pathway. If you assume that the contribution of environmental water for an increase in one trophic level is 0.23, then the contribution of environmental water after an increase in three trophic levels is 0.54. By the time the terrestrial organic matter makes its way to piscivorous fish (2 trophic levels above zooplankton), the overall contribution of environmental water to the consumer's hydrogen content will be 0.73. Given the considerable uncertainty in current best estimates for ω (i.e., $\pm 1 \text{ SD} = 0.09$), the environmental water contribution to zooplankton could range from 0.36 to 0.69 for this pathway, and for piscivorous fish they could range from 0.53 to 0.86. Given this very large range of uncertainty for environmental water contributions it is questionable whether you could get reasonable estimates for zooplankton and quite unlikely that estimates for piscivorous fish would be of any value whatsoever.

3. There is an even greater conundrum for the t-OM pathway to benthic invertebrates, which the authors have not adequately described. For example, on lines 38-39, they state "Concurrently, t-OM mobilized by bacteria and fungi can support detritivorous zoobenthos and benthivorous fish". But this statement does not indicate whether the

pathway is based on dissolved or particulate OM. I contend that at least >> 90% of the aquatic ecology community believes the primary pathway for t-OM to be incorporated into the tissues of benthic invertebrates in streams and lakes is via the classic particulate pathway, i.e., the “peanut butter and crackers” analogy articulated by Cummins (1974). In line 160 of the MS, the authors suggest both particulate and dissolved OM is utilized by benthic invertebrates. In lines 217-218, the authors state “availability of t-OM via a microbial link is likely the key driver of consumer allochthony for both benthic and pelagic food webs in lakes”.

If the predominant pathway for t-OM to benthos is particulate, then there is still the large problem that the authors do not have an estimate for the d2H value for the particulate t-OM that is being assimilated by the benthic invertebrates in their lakes. Alternatively, if the prevailing t-OM pathway to benthos is microbial, then the authors have the similar large problem that they have not accounted for the additional trophic levels between the t-OM and the benthos in the environmental water calculations. Reading the various passages in the MS relevant to this point, I get the impression that Keva et al. mostly believe benthos consume the biofilm that grows on particulate t-OM. For example, on lines 159-160 the authors mentioned that benthos feed “on detritus with its associated microbial communities as well as the surrounding biofilm which can incorporate both particulate and dissolved forms of t-OM”. If this is the case (which I also believe), then the authors need to have measured the d2H values of the particulate t-OM loaded to their lakes, as well as accounted for the microbial processing and additional trophic levels and incorporation of environmental water. I do not believe this was done in the current MS, or if it was done it was not explained adequately.

4. In the first round of review, I noted that it is widely known that common herbivorous zooplankton like *Daphnia* also consume some protozoa and other microzooplankton, which would put them at a trophic position somewhat more than a conventional herbivore, e.g., a trophic position in the food web of 2.1 relative to algae (with a trophic position of 1). In their response, the authors acknowledged my point and stated that in their revised analysis they have dealt with this issue by assigning herbivorous cladocera a trophic position of 2.0 ± 0.1 . This supposed correction is wrong! If you assume that zooplankton have a trophic position of 2.0 ± 0.1 , then you are stating that half the time these zooplankton will have a trophic position < 2.0 , which means they would have to be partially mixotrophic (i.e., capable of some photosynthesis themselves). There is no evidence in the cladocera literature to support such an assumption. Instead, the authors should have assumed that cladocera have a trophic position of 2.1, with an uncertainty of ± 0.1 and with the added caveat that they never have a trophic position below 2.0 because cladocerans, copepods, Chaoborus and benthic macroinvertebrates are not photosynthetic.
5. The authors also claimed that the “fundamental difference between the Bayesian and simple linear mixing models: the former gives population dietary probabilities, whereas the latter gives individual-level estimates of dietary proportions”. I completely disagree

with this claim. Firstly, if you have a single zooplankton sample from a particular lake, you are only going to obtain a diet composition estimate for THAT SAMPLE (not for the “population”). To get an estimate for the actual population, you would have to collect multiple samples of the same taxa at multiple representative locations on multiple representative dates. The data shown in Supplementary Table 4 show the vast majority of zooplankton and benthos samples were represented by samples of 1 or 2 per lake (presumably collected on the same date within a lake). The fish samples were usually 5 per lake (again, presumably collected on the same date within each lake). With this very low level of sampling, it is inconceivable that one could obtain anything approaching “population dietary probabilities”. I recently sampled a lake for Daphnia and Chironomids at ten sampling stations each date over a five month period. We used a fatty acid based dietary mixing model to estimate their diets. This study showed diet composition was similar between stations within a date, but vastly different from one month to another. Both Daphnia and Chironomids were strongly reliant on diatoms in the spring and then showed a sharp transition to much lower absolute fatty acid content and reliance on a mix of cyanobacteria, diatoms and cryptophytes during the summer bloom period.

A Bayesian approach will give you fancy plots with distributions for the dietary estimates, but these distributions just represent the uncertainty inherent in the input parameters. They are not in any way representative of the true uncertainty within the population, which as already noted could only be ascertained with a MUCH more rigorous sampling design. For example, based on the information presented in Supplementary Table 4 this could only be done for some of the zooplankton and benthos taxa collected from Jyväsjärvi and Sääksjärvi. Also, if you want to quantify uncertainty associated with the input parameters for only one or two samples, this can also be easily done using classic mass balance calculations with a Monte Carlo approach. But I do not think this would be worth the trouble since the calculated uncertainty would only pertain to those very specific samples. You could also, relatively easily, use a classic mass balance and Monte Carlo approach to estimate the diet of the population, provided you had proper population level data.

The main difference between the classic mass balance based approach and the Bayesian approach is that the mass balance approach is solely based on the Law of Mass Conservation, one of the foundational principles of science. Additionally, the solution to a classic mass balance calculation will exactly match the observed consumer’s $\delta^2\text{H}$ value. This will often not be the case for Bayesian results, which can be very different from the observed stable isotope values. Conversely, the MixSIAR Bayesian algorithm is extremely “malleable”, and you can tinker with it in ways that can strongly influence model outcomes. That is, it is susceptible to what is generally known as data-torturing aka ‘making the data fit the hypothesis’ (Mills 1994). The biggest problem with the Bayesian estimates is that they always constrain the estimated contributions to the 0-1 domain, when the raw data may indicate different outcomes. In the case of classic mass balance analyses, it simply is not possible to tinker with the algorithm. One set of data

(and an assumed environmental water value) will give one exact answer. The most worrisome concern is that the outcomes provided by the Bayesian mixing model may be more a function of the attributes of the algorithm than the information contained in the raw data.

For example, on lines 50-51 of the MS, Keva et al. state “a recent study of 147 lakes indicated zooplankton to rely on average 11% to 83% on t-OM sources (Tanentzap et al. 2017)”. Because I have all the original raw data from Tanentzap et al. 2017, which I have already reanalyzed (see Brett et al. 2018), I can compare zooplankton allochthony estimates from their Bayesian analysis to a direct mass balance analysis using exactly the same data. My reanalysis of the zooplankton d2H based data (n = 439) from Tanentzap et al. (2017) gave a median zooplankton allochthony estimate of 5% when using an ω value of 0.23. The 95% percentile range for these data was -123% to +55% allochthony. Overall, 44% of the zooplankton cases from the Tanentzap et al. (2017) dataset indicated negative allochthony when using d2H as the dietary tracer. While 44% negative estimates may seem paradoxical, if the true value was zero and the raw data used had some estimation, observational and measurement error, then 50% of the estimates should be negative. We know the classic mass balance analysis of these data is correct because we can use the individual zooplankton allochthony estimates and resource and lake water d2H values to perfectly reconstruct the original zooplankton raw d2H values. [This of course does not mean that the underlining allochthony estimates are absolutely correct, because these estimates will be no better than the data they are based on, and these data likely have substantial estimation error]. This begs the question, why does the correct mass balance based analysis of these exact data indicate negligible allochthony, when the Bayesian results indicate considerable allochthony (i.e., Tenentzap et al. reported a median estimate of 42%)?

I also want to note that Keva et al. did not present model corroboration results! As noted previously, a proper mass balance will yield perfect corroboration results. For example, I was able to reconstruct the observed zooplankton d2H values from Tanentzap et al.'s raw data using the equation below:

$$(\delta^2 H_{terr} * \theta + \delta^2 H_{phyto} * (1 - \theta)) * (1 - (1 - (1 - \omega)^{TL})) + (1 - (1 - \omega)^{TL}) * \delta^2 H_{H2O} = \delta^2 H_{zoop},$$

where Θ represents the estimated terrestrial contribution, ω represents the assumed hydrogen contribution from environmental water, TL represents the number of trophic levels above primary producers, and $\delta^2\text{H}$ represents the hydrogen stable isotope values for terrestrial and aquatic organic matter, lake water and zooplankton.

6. I have read the original and revised MS several times and I am still very confused as to how the authors represented the algal $\text{d}2\text{H}$ values in their Bayesian model. I asked this question in the previous round of review and it still has not been clearly explained in the response and more importantly in the MS itself. For example, in their response the authors stated *“we have sampled benthic algae from each study lake. Thus, we took the average discrimination factor from the analysed samples and used that with the lake water to model algae $\delta^2\text{H}$ values. . . . We do not fully understand why Reviewer 2 argues that we have used little field data for the variables of interest, because that is evidently not the case in our study.”* This response does not make sense and it is self-contradictory. Firstly, if as the MS states benthic algae were sampled multiple locations in each lake (see lines 302-305), why weren't the actual observed algal $\text{d}2\text{H}$ values used to represent the autochthonous resources in the dietary mixing model? If you have access to real data, why would you use “modeled” data instead. For the raw data shared by the authors, it appears that only one algal $\text{d}2\text{H}$ value was used for nearly all of the lakes, and this algal value was almost always -106.7 $\text{d}2\text{H}$ different from the lake water value. On lines 403-405 of the MS, the authors state *“observed spatiotemporal standard deviation [was] $\pm 23.6\%$ ”*. However, the variation among all lakes for the algal $\text{d}2\text{H}$ values actually used in the Bayesian model was only $\pm 8\%$. From the outside looking in, it appears that the authors have used a quite convoluted approach to greatly reduce the lake to lake variability in the $\text{d}2\text{H}$ values for algae. In addition, this approach results in a situation where the algal and lake water $\text{d}2\text{H}$ values are nearly perfectly aligned ($r^2 = 0.99$). This example is exactly why I am concerned that the authors are not using real data when apparently it exists. Does their decision influence the outcome of the Bayesian mixing model? Without seeing the observed algal $\text{d}2\text{H}$ values from each lake, it is very difficult to understand why the authors used “modeled” data when they could have used directly observed data, this seems suspect.

To reiterate my previous point, the dietary mixing model depends on five variables (i.e., the $\text{d}2\text{H}$ values for t-OM, algal, water and zooplankton and the assumption for environmental water). On these five terms, only the water and zooplankton $\text{d}2\text{H}$ values are based on actual observations.

7. I looked at the raw data the authors posted for this submission. I am concerned by the calculated Trophic Level values for a few of the consumers. For example, the average trophic level for Asellus was 1.59. This means the authors are inferring that Asellus is 41% phototrophic and 59% heterotrophic. In > 80% of the cases, the calculated trophic level was < 2. 10% of the Asellus samples had estimated trophic positions below plants (i.e., < 1). A trophic position characteristic of a mixotroph could be true for a tropical coral which hosts zooxanthellae, but a trophic level < 2 cannot be true for any of the invertebrates sampled from Keva et al.'s lakes. I realize the authors used a superficially reasonable approach to calculate the consumer trophic levels, but this approach has considerable uncertainty ($3.4 \pm 1.0\%$) and it produced unreasonable results in many cases. The 1.59 average trophic level also means the average environmental water contribution assumed for Asellus is only 0.14. I also note that this is not inconsequential because the overall allochthony estimate is dependent on the trophic level assumption used in the mass balance analysis (see plot below where I plotted the estimated TL against the estimated allochthony for Keva et al.'s Asellus raw data). Because the environmental water assumption is directly dependent on the assumed trophic level, the environmental water assumption is similarly correlated with the estimated Asellus allochthony.

There is a similar issue with the estimated trophic position for littoral Chironomids and to a lesser extent littoral benthic macroinvertebrates, which both have average trophic levels < 2. I am also skeptical of the estimated trophic position for pike, which is slightly less than that for smelt. Since Keva et al.'s stomach content data indicate the pike they sampled had exclusively consumed fish, and the fish that pike would likely to have consumed had an average trophic level of ≈ 3.5 , I believe that a trophic level of 4.5 is much more reasonable for pike. It appears that the authors estimated the trophic level of all consumers relative to the ^{15}N values of Cladocera, but the authors do not explain how they estimated the trophic position of consumers in lakes where a cladoceran samples were not collected.

8. If you look at the main plots in the Keva et al. MS, there are some strange trends that concern me. For example, in Fig. 3 the allochthony estimates for Asellus in panel b2 appear to be very similar to the estimates for littoral bulk zoobenthos in panel b1 and profundal zoobenthos in panel b4. Additionally, the results for ruffe in panel c5 appear to be very similar to the results of perch > 15 cm in c1. These similarities in the allochthony estimates for different consumers strongly suggests that the estimates

depicted in these plots are mainly a function of the structure of the Bayesian mixing model and its underlying assumptions, rather than the consumer raw d2H values.

9. **Finally, perhaps the most important concern is that when using the raw data Keva et al. provided for this review, I was unable to re-create the main trends these authors reported for Asellus in their Fig. 3 panel b2 and Fig. 4S panel b2.** In their response to my original comments, these authors stated that the actual allochthony estimates are not nearly as important as the trends they observed between the estimated consumer allochthony and various landcover measures. I used the raw data they provided for the review process to calculate Asellus allochthony using the classic balance approach. I then compared these estimates to the PC1 score and the % Forest land cover data for each of the lakes where Asellus data were available. This comparison showed no relationship between the two land cover characterizations and the estimated allochthony, in stark contrast to the key results from the Keva et al. MS. This discrepancy must be resolved. To help with this, I have provided my allochthony calculations for the Asellus data. So far, I have only attempted to re-create the reported trends for Asellus and landcover, but if I am unable to re-create these trends for Asellus, I strongly suspect the same will be true for the other consumers.

Minor point

10. In their response to one of my comments, the authors state that “the equations in Brett et al. (2018) assume the trophic level difference between zooplankton and sources to be exactly 1”. This is incorrect, Brett et al (2018) used the trophic levels indicated in the original raw data from Tanentzap et al. (2017). Tanentzap et al. (2017) assumed various cladocera had a trophic position of 1 (above algae), calanoid cyclopoids had trophic positions of 1.25 and 1.5, respectively, and Chaoborus had a trophic position of 2.0.

Mixed zooplankton samples had intermediate trophic positions. The environmental water contributions were all corrected for trophic positions as described by Solomon et al. (2009), although the Brett et al. (2018) paper mistakenly did not mention that detail.

Conclusion

11. ***If the discrepancy between my estimated Asellus allochthony and the PC1 and Percent Forest landcover characterizations (compared to what was reported in the MS) cannot be resolved, this study should not be published in Nature Communications or any other journal.*** I am unable to replicate the core findings of the Keva et al. MS using their own raw data, which calls into question the entire premise of their analysis.

My overreaching general concern with this paper is that the results appear to be mainly a function of the structure and assumptions imbedded into the Bayesian mixing model rather than attributes of the underlying raw data. Additionally, in some cases “modeled data” are used rather than available sampled data (e.g., the algal d2H values). This creates a direct correspondence between the water and algal d2H values used in the modeling, which would not be the case with real data where the average uncertainty when compared primary producer to water d2H values is large $\pm 27\%$. The t-OM d2H values used in the Bayesian modeling are highly problematic because these values represent 2 trophic levels below what the zooplankton or benthos are consuming. For example, because it is envisioned that zooplankton obtain their t-OM along a bacteria to protozoan pathway, the zooplankton are 3 trophic levels above the t-OM but 1.1 trophic levels above algae which they directly consume. This means the environmental water contributions along the t-OM pathway will be much larger than along the algal pathway, (e.g., 0.54 vs 0.23, respectively). The trophic pathway for t-OM to benthos is even more muddled because the authors do not use d2H values for particulate t-OM in their modeling and they do not account for the larger contribution of environmental water along this pathway.

Michael T. Brett

Mills JL. 1993 Data torturing. N. Engl. J. Med. 329, 1196–1199.

Cummins, K. W. 1974. Structure and function of stream ecosystems. BioScience 24:631-641.

Solomon, C.T., J.J. Cole, R.R. Doucett, M.L. Pace, N.D. Preston, L.E. Smith, and B.C. Weidel. 2009. The influence of environmental water on the hydrogen stable isotope ratio in aquatic consumers. Oecologia 161:313–324.

1 Lake	species	For_area	PC1	Lake
2 Alajarvi	Asellus_littor.	77.5745898	-0.2396601	Alajärvi
3 Alajarvi	Asellus_littor.	77.5745898	-0.2396601	
4 Alanne	Asellus_littor.	82.0332268	-3.2491402	Älänne
5 Alanne	Asellus_littor.	82.0332268	-3.2491402	
6 Arkusjarvi	Asellus_littor.	49.1645516	1.10943832	Arkusjärvi
7 Arkusjarvi	Asellus_littor.	49.1645516	1.10943832	
8 Erajarvi	Asellus_littor.	64.787552	0.94097836	Eräjärvi
9 Erajarvi	Asellus_littor.	64.787552	0.94097836	
10 Haapajarvi	Asellus_littor.	75.0606475	-0.1619571	Haapajärvi
11 Haapajarvi	Asellus_littor.	75.0606475	-0.1619571	
12 Hameenjarvi	Asellus_littor.	82.7023717	-1.4895931	Hämeenjärvi
13 Hameenjarvi	Asellus_littor.	82.7023717	-1.4895931	
14 Hiidenvesi	Asellus_littor.	67.3993044	1.52578337	Hiidenvesi
15				
16 Hirvijarvi	Asellus_littor.	90.5908096	-2.887364	Hirvijärvi
17 Hirvijarvi	Asellus_littor.	90.5908096	-2.887364	
18 Horkajarvi	Asellus_littor.	97.5183824	-3.5808388	Horkajärvi
19 Horkajarvi	Asellus_littor.	97.5183824	-3.5808388	
20 Joroisselka	Asellus_littor.	71.4537196	-0.3355184	Joroisselkä
21 Joroisselka	Asellus_littor.	71.4537196	-0.3355184	
22 Jyvasjarvi	Asellus_littor.	71.2225824	-0.1201369	Jyväsjärvi
23 Jyvasjarvi	Asellus_littor.	71.2225824	-0.1201369	
24 Jyvasjarvi	Asellus_littor.	71.2225824	-0.1201369	
25 Jyvasjarvi	Asellus_littor.	71.2225824	-0.1201369	
26 Jyvasjarvi	Asellus_littor.	71.2225824	-0.1201369	
27 Jyvasjarvi	Asellus_littor.	71.2225824	-0.1201369	
28 Kakkisenjarvi	Asellus_littor.	83.6136836	-2.6026419	Kakkisenjärvi
29 Kaks Kerranja	Asellus_littor.	52.1847788	2.75921245	
30 Kaks Kerranja	Asellus_littor.	52.1847788	2.75921245	Kaks Kerranjä
31				
32 Koylionjarvi	Asellus_littor.	53.5948883	4.7724033	Köyliönjärvi
33 Koylionjarvi	Asellus_littor.	53.5948883	4.7724033	
34 Kuhajarvi	Asellus_littor.	57.3851272	0.54877771	Kuhajärvi
35 Kuhajarvi	Asellus_littor.	57.3851272	0.54877771	
36 Kuontijarvi	Asellus_littor.	69.3315741	-0.1132982	Kuontijärvi
37 Kuontijarvi	Asellus_littor.	69.3315741	-0.1132982	
38 Majajarvi	Asellus_littor.	96.4300714	-3.5610151	Majajärvi
39 Majajarvi	Asellus_littor.	96.4300714	-3.5610151	
40 Niemisjarvi	Asellus_littor.	73.1888604	-0.8259024	Niemisjärvi
41 Niemisjarvi	Asellus_littor.	73.1888604	-0.8259024	
42 Paajarvi	Asellus_littor.	73.5199268	0.23436892	Pääjärvi
43 Paajarvi	Asellus_littor.	73.5199268	0.23436892	
44 Pankajarvi	Asellus_littor.	71.0458345	-1.0339696	Pankajärvi

45					
46					Pesosjärvi
47					
48	Pusulanjarvi	Asellus_littor	69.8573478	0.89041957	Pusulanjärvi
49					
50	Pyhajarvi	Asellus_littor	59.0848509	3.97520278	Pyhäjärvi
51					
52	Ruokojarvi	Asellus_littor	76.0721469	-0.913896	Ruokojärvi
53					
54	Saaksjarvi	Asellus_littor	59.7695162	0.76912163	Sääksjärvi
55	Saaksjarvi	Asellus_littor	59.7695162	0.76912163	
56	Saaksjarvi	Asellus_littor	59.7695162	0.76912163	Sääskjärvi
57	Saaksjarvi	Asellus_littor	59.7695162	0.76912163	
58	Saaksjarvi	Asellus_littor	59.7695162	0.76912163	
59	Saaksjarvi	Asellus_littor	59.7695162	0.76912163	
60	Saaksjarvi	Asellus_littor	59.7695162	0.76912163	
61	Saaksjarvi	Asellus_littor	59.7695162	0.76912163	
62	Saaksjarvi	Asellus_littor	59.7695162	0.76912163	
63	Saaksjarvi	Asellus_littor	59.7695162	0.76912163	
64	Saaksjarvi	Asellus_littor	59.7695162	0.76912163	
65	Saaksjarvi	Asellus_littor	59.7695162	0.76912163	
66	Saaksjarvi	Asellus_littor	59.7695162	0.76912163	
67	Saaksjarvi	Asellus_littor	59.7695162	0.76912163	
68	Saaskjarvi	Asellus_littor	55.0902913	3.49133584	
69	Saaskjarvi	Asellus_littor	55.0902913	3.49133584	
70	Suuri-Vahvan	Asellus_littor	72.7403487	-1.4390464	Suuri-Vahvan
71	Suuri-Vahvan	Asellus_littor	72.7403487	-1.4390464	
72	Suuri Jukajan	Asellus_littor	77.0619887	-0.6272018	Suuri Jukajän
73	Suuri Jukajan	Asellus_littor	77.0619887	-0.6272018	
74	Tottijarvi	Asellus_littor	50.7366749	3.07918953	Tottijärvi
75					
76	Valkea-Kotine	Asellus_littor	80	-3.0580562	Valkea-Kotine
77	Valkea-Kotine	Asellus_littor	80	-3.0580562	
78	Vesijarvi	Asellus_littor	51.1470723	3.12949093	Vesijärvi
79					
80	Viippero	Asellus_littor	88.1761948	-2.1559338	Viippero
81					
82	Viitaanjarvi	Asellus_littor	82.2390506	-1.6602183	Viitaanjärvi
83					
84	Villikkalanjan	Asellus_littor	61.7398059	2.37286733	Villikkalanjän
85	Villikkalanjan	Asellus_littor	61.7398059	2.37286733	
86	Ylisjarvi	Asellus_littor	65.041819	2.10195131	Ylisjärvi
87	Ylisjarvi	Asellus_littor	65.041819	2.10195131	

Sääksjärvi
Sääskjärvi

Sääskjärvi

mean
median

d2H_water_c Aq	Terr	
-76.718013	-183.41801	-113.95252
-76.718013	-183.41801	-113.95252
-87.390731	-194.09073	-121.93307
-87.390731	-194.09073	-121.93307
-65.444578	-172.14458	-124.16955
-65.444578	-172.14458	-124.16955
-62.660834	-169.36083	-113.75776
-62.660834	-169.36083	-113.75776
-83.550158	-190.25016	-121.50852
-83.550158	-190.25016	-121.50852
-68.882249	-175.58225	-114.8009
-68.882249	-175.58225	-114.8009
-71.255103	-177.9551	-107.76078
-71.255103	-177.9551	-107.76078
-86.373158	-193.07316	-114.93992
-86.373158	-193.07316	-114.93992
-73.375195	-180.0752	-121.47008
-73.375195	-180.0752	-121.47008
-73.345007	-180.04501	-112.69943
-73.345007	-180.04501	-112.69943
-80.557733	-190.15085	-124.09401
-80.557733	-190.15085	-124.09401
-80.557733	-190.15085	-124.09401
-80.557733	-190.15085	-124.09401
-80.557733	-190.15085	-124.09401
-80.557733	-190.15085	-124.09401
-79.828814	-186.52881	-130.66291
-79.828814	-186.52881	-130.66291
-56.831307	-163.53131	-131.77752
-56.831307	-163.53131	-131.77752
-61.918246	-168.61825	-129.14445
-61.918246	-168.61825	-129.14445
-76.315645	-183.01565	-134.79843
-76.315645	-183.01565	-134.79843
-87.881118	-194.58112	-142.46341
-87.881118	-194.58112	-142.46341
-76.562523	-183.26252	-121.47008
-76.562523	-183.26252	-121.47008
-77.710123	-184.41012	-116.15234
-77.710123	-184.41012	-116.15234
-74.486603	-181.1866	-119.96635
-74.486603	-181.1866	-119.96635
-74.047259	-180.74726	-116.21428

Lake	species	d15N
Alajarvi	Asellus_littor	5.06341835
Alajarvi	Asellus_littor	6.48554025
Alanne	Asellus_littor	1.84783652
Alanne	Asellus_littor	1.93369226
Arkusjarvi	Asellus_littor	4.47780365
Arkusjarvi	Asellus_littor	2.89826816
Erajarvi	Asellus_littor	5.2907338
Erajarvi	Asellus_littor	6.05181894
Haapajarvi	Asellus_littor	4.31724699
Haapajarvi	Asellus_littor	5.44017842
Hameenjarvi	Asellus_littor	0.72330777
Hameenjarvi	Asellus_littor	0.50903991
Hiidenvesi	Asellus_littor	8.71150721
Hirvijarvi	Asellus_littor	1.1232043
Hirvijarvi	Asellus_littor	1.73333392
Horkajarvi	Asellus_littor	-0.3422249
Horkajarvi	Asellus_littor	0.62220085
Joroisselka	Asellus_littor	4.2274943
Joroisselka	Asellus_littor	5.3426004
Jyvasjarvi	Asellus_littor	2.88639543
Jyvasjarvi	Asellus_littor	7.79399626
Jyvasjarvi	Asellus_littor	8.6017088
Jyvasjarvi	Asellus_littor	7.67703879
Jyvasjarvi	Asellus_littor	8.4984407
Jyvasjarvi	Asellus_littor	9.79818218
Kakkisenjarvi	Asellus_littor	1.89090375
Kakkisenjarvi	Asellus_littor	4.43426493
Kakkisenjarvi	Asellus_littor	4.46281377
Koylionjarvi	Asellus_littor	7.96649709
Koylionjarvi	Asellus_littor	8.2762188
Kuhajarvi	Asellus_littor	3.33850326
Kuhajarvi	Asellus_littor	4.22792691
Kuontijarvi	Asellus_littor	3.08581753
Kuontijarvi	Asellus_littor	2.64850803
Majajarvi	Asellus_littor	2.77109088
Majajarvi	Asellus_littor	2.08538035
Niemisjarvi	Asellus_littor	7.35706349
Niemisjarvi	Asellus_littor	6.46206873
Paajarvi	Asellus_littor	5.3377689
Paajarvi	Asellus_littor	5.57866973
Pankajarvi	Asellus_littor	5.63817351

-74.047259	-180.74726	-116.21428			
-91.448726	-198.14873	-133.41737			
-91.448726	-198.14873	-133.41737			
-70.32907	-177.02907	-110.11118	Pusulanjarvi	Asellus_littor	7.55170681
-70.32907	-177.02907	-110.11118			
-74.961117	-181.66112	-113.65154	Pyhajarvi	Asellus_littor	7.66926994
-74.961117	-181.66112	-113.65154			
-68.820294	-175.52029	-135.34955	Ruokojarvi	Asellus_littor	2.12208587
-68.820294	-175.52029	-135.34955			
-76.008562	-182.62435	-113.02093	Saaksjarvi	Asellus_littor	7.5163058
-76.008562	-182.62435	-113.02093	Saaksjarvi	Asellus_littor	7.72906102
-76.008562	-182.62435	-113.02093	Saaksjarvi	Asellus_littor	5.9957441
-76.008562	-182.62435	-113.02093	Saaksjarvi	Asellus_littor	8.7259416
-76.008562	-182.62435	-113.02093	Saaksjarvi	Asellus_littor	6.62641071
-76.008562	-182.62435	-113.02093	Saaksjarvi	Asellus_littor	7.6924304
-76.008562	-182.62435	-113.02093	Saaksjarvi	Asellus_littor	8.09232262
-76.008562	-182.62435	-113.02093	Saaksjarvi	Asellus_littor	6.87753111
-76.008562	-182.62435	-113.02093	Saaksjarvi	Asellus_littor	7.41414729
-76.008562	-182.62435	-113.02093	Saaksjarvi	Asellus_littor	9.4458932
-76.008562	-182.62435	-113.02093	Saaksjarvi	Asellus_littor	5.13980103
-76.008562	-182.62435	-113.02093	Saaksjarvi	Asellus_littor	6.15367196
-76.008562	-182.62435	-113.02093	Saaksjarvi	Asellus_littor	6.13384679
-76.008562	-182.62435	-113.02093	Saaksjarvi	Asellus_littor	6.96879105
-76.008562	-182.62435	-113.02093	Saaskjarvi	Asellus_littor	8.15075296
-76.008562	-182.62435	-113.02093	Saaskjarvi	Asellus_littor	8.92268049
-66.042281	-172.74228	-114.95399	Suuri-Vahvan	Asellus_littor	1.51536535
-66.042281	-172.74228	-114.95399	Suuri-Vahvan	Asellus_littor	0.33375636
-72.599254	-179.29925	-124.16955	Suuri Jukajan	Asellus_littor	1.68101825
-72.599254	-179.29925	-124.16955	Suuri Jukajan	Asellus_littor	3.02528042
-62.240444	-168.94044	-109.67851	Tottijarvi	Asellus_littor	5.32814262
-62.240444	-168.94044	-109.67851			
-63.711149	-170.41115	-121.47008	Valkea-Kotin	Asellus_littor	2.09205191
-63.711149	-170.41115	-121.47008	Valkea-Kotin	Asellus_littor	1.56432191
-65.132131	-171.83213	-105.77849	Vesijarvi	Asellus_littor	5.11096845
-65.132131	-171.83213	-105.77849			
-78.985651	-185.68565	-126.36413	Viippero	Asellus_littor	1.03989418
-78.985651	-185.68565	-126.36413			
-89.035222	-195.73522	-127.75703	Viitaanjarvi	Asellus_littor	5.19458501
-89.035222	-195.73522	-127.75703			
-74.961117	-181.66112	-109.3025	Villikkalanjan	Asellus_littor	7.5646868
-74.961117	-181.66112	-109.3025	Villikkalanjan	Asellus_littor	8.51845198
-60.88686	-167.58686	-127.79869	Ylisjarvi	Asellus_littor	6.30419315
-60.88686	-167.58686	-127.79869	Ylisjarvi	Asellus_littor	6.98763518

-78.151934	-184.68351	-117.46594
-73.865191	-180.56519	-108.57592
mean	mean	mean
-76.008562	-182.62435	-113.02093

d2H_water_c	Aq	Terr	epsilon	Lake	species	d15N
-74.499341	-181.38552	-119.67196	-106.9		Asellus_littor	5.11436729
-76.008562	-182.62435	-116.21428	-106.6		Asellus_littor	5.34018465

TL	d2H(zoop,corr)	wcorr	w = 0.23	allo	percentile
1.4595722	-128.5531622	0.11		0.790	0.1
1.87784335	-158.9489538	0.21		0.352	0.2
2.23784984	-185.9666289	0.28		0.113	0.3
2.26310153	-167.2431536	0.28		0.372	0.4
1.42031261	-131.1425676	0.10		0.855	0.5
0.95574335	-115.5239471	-0.01		1.180	0.6
1.36511397	-133.8243279	0.09		0.639	0.7
1.58896254	-121.036544	0.14		0.869	0.8
1.24475416	-117.5514183	0.06		1.058	0.9
1.57502811	-133.8947834	0.14		0.820	
1.65450995	-124.8174499	0.16		0.835	
1.59148999	-139.4544805	0.14		0.594	
1.60423325	-145.1110281	0.15		0.468	
1.44023005	-149.7957818	0.11		0.554	
1.61967994	-142.7353292	0.15		0.644	
1.27835066	-115.9336946	0.07		1.094	
1.5620053	-119.6498841	0.14		1.031	
1.15260309	-114.6179108	0.04		0.972	
1.48057548	-130.1487527	0.12		0.741	
0.80490434	-113.0118346	-0.05		1.168	
2.24831635	-147.6261733	0.28		0.644	
2.48587886	-148.5100314	0.32		0.630	
2.21391709	-147.8092773	0.27		0.641	
2.45550589	-171.1259951	0.32		0.288	
2.83778279	-164.2971159	0.38		0.391	
1.70951185	-128.3307057	0.17		1.042	
1.3699198	-126.0161607	0.09		1.083	
1.37831652	-115.8335572	0.09		1.502	
1.81549737	-131.8377407	0.19		0.932	
1.90659199	-132.4416851	0.21		0.916	
1.05819261	-147.4376783	0.02		0.738	
1.3197878	-158.8697194	0.08		0.501	
1.0684307	-165.0781848	0.02		0.566	
0.93981026	-161.1545529	-0.02		0.641	
1.64444282	-128.8678034	0.16		0.880	
1.44276325	-123.1760129	0.11		0.972	
2.10969717	-134.6728723	0.25		0.729	
1.84646341	-133.2848665	0.20		0.749	
1.78329546	-152.3791976	0.19		0.471	
1.85414864	-147.563948	0.20		0.549	
1.52271776	-137.642728	0.13		0.668	

2.43884705	-174.9026804	0.31	0.032
1.75242537	-149.3846262	0.18	0.475
2.01018181	-155.5339893	0.23	0.498
1.30431357	-116.5128187	0.08	0.950
1.36688863	-125.0805842	0.09	0.827
0.85708954	-107.014183	-0.04	1.086
1.6600888	-184.0284697	0.16	-0.020
1.04257971	-130.6406321	0.01	0.747
1.35611492	-148.2509613	0.09	0.494
1.47373028	-151.3476629	0.12	0.449
1.11643866	-129.8341118	0.03	0.758
1.27426694	-143.6752098	0.07	0.560
1.87183927	-160.8979377	0.20	0.312
0.60534157	-121.923363	-0.11	0.872
0.90353891	-117.339933	-0.03	0.938
0.89770797	-128.8310272	-0.03	0.773
1.14327981	-125.7064154	0.04	0.818
1.53478818	-131.1409931	0.13	0.740
1.76182569	-135.3940597	0.18	0.679
1.90411095	-138.1409366	0.21	0.599
1.5565789	-121.9987702	0.14	0.878
2.26837357	-163.0006457	0.28	0.296
2.66374479	-184.6618745	0.35	-0.097
1.56883986	-127.7228457	0.14	0.696
1.59968387	-119.7994063	0.15	1.034
1.44446916	-108.4511589	0.11	1.266
1.41758033	-133.5626876	0.10	0.579
1.3508484	-138.172255	0.09	0.801
1.59364797	-153.4301671	0.14	0.622
2.02952736	-152.729395	0.24	0.400
2.31004653	-153.4912723	0.29	0.389
1.12000372	-125.2550405	0.03	1.064
1.32101608	-120.8227424	0.08	1.175

TL	d2H(zoop,corr)	wcorr	w = 0.23	allo
1.59	-138.8	0.137		0.69
1.56	-133.9	0.136		0.73

wcorr	TL	Allo
0.012	1.05	0.358
0.066	1.26	0.473
0.091	1.36	0.565
0.109	1.44	0.641
0.136	1.56	0.733
0.146	1.60	0.786
0.181	1.76	0.869
0.211	1.91	0.959
0.280	2.26	1.077

Main points

1. Is the algorithm making the data fit the hypothesis?

In their second reply, the authors stated “we sincerely hope that Reviewer #2 is not suggesting that, by collecting and analysing thousands of samples over three years, applying the best analytical data corrections according to current knowledge, including the use of other measured data to estimate consumer TPs, and constructing a mixing model that tests for the effect of a well-reasoned environmental gradient while concurrently accounting for the complex data structure, we are merely ‘making our data fit the hypothesis’”. In fact, I will state very clearly that given the way that the Bayesian algorithm has been set up, the authors can profoundly change the modeled outputs just by changing the structure of the algorithm or some of the assumptions imbedded within the algorithm. **This is precisely my biggest concern with this analysis.** For example, in the last round of review I reanalyzed the authors’ Asellus data and my analysis showed a very flat response between the estimated allochthony and the Environmental gradient (PC1). I also pointed out a discrepancy in how the authors calculated the trophic level for this and other consumers. In their revision Keva et al. reanalyzed their data for Asellus and now report a completely different response, albeit a response very similar to the one I previously found for these data. The change in these two plots was solely due to how the Bayesian model was structured and how some of the underlying data were interpreted!

Additionally, as we all agree, changing the assumed environmental water assumption profoundly changes the model outputs! In their response to my second round of comments the authors stated: “We would also highlight that due to the crossed random effects structure of the populations we have implemented here, **population posteriors are also informed by those from other lakes for the same species, and other species from the same lake.**” Firstly, I do not feel this detail is explained adequately in the MS. Perhaps this is what the authors are alluding to on lines 487-488 when they state: “in the MixSIAR model, we set lake and consumer taxon as random factors.” But unless you are fluent in Bayesian jargon, you would have no idea what they are getting at here. Secondly, it is evident that smearing the outcomes across “lakes for the same species, and across species from the same lake” will tend cause all of the results to converge on a

common outcome that is not consistent with the attributes of the original raw data for individual consumers. This is a very clear example of where the structure of the Bayesian model profoundly influences the outputs.

In the previous round of review, I asked why the some of the results for the different taxa appeared to show nearly identical patterns. For example, in the plot to the right I have highlighted in red several clusters of outcomes that are oddly very similar. Real data rarely show these sorts of patterns. Now in retrospect, it is apparent that these anomalous results are probably because the authors “crossed” the outcomes across “lakes for the same species, and across species from the same lake”. In their response, Keva et al. also stated “*given the functional similarities of Asellus and bulk zoobenthos, and perch with ruffe, it should not be surprising that their estimated terrestrial contributions are similar on a lake-by-lake basis.*” I believe it is misleading to imply that these anomalous patterns in the plots have some deeper ecological meaning when it is much more likely they are almost entirely due to the way the authors set up their Bayesian mixing model!

It may seem stark, but I am very concerned that 1) by crossing lakes and species and species and lakes, 2) by changing how this crossing is done from one MS draft to another, 3) by modifying your definition of trophic level (so that some consumers have trophic levels < 2 or not), 4) using modeled data instead of real, 5) not fully accounting for environmental water contributions, and 6) by choosing one environmental water assumption over another there are a multitude of opportunities to make the data fit a particular hypothesis!

2. Why did I only reanalyze the Asellus data in the second round of review?

In their response, the authors questioned why I only reanalyzed the Asellus data. Specifically, Keva et al. state “*looking at the raw data alone in Fig.2 would suggest that Asellus, of all 19 taxa groups samples, would be the most likely to produce a flat trend with PC1. It is circumspect therefore that the reviewer sought only to analyse this specific group, rather than the many others that show decreasing trends in the raw data, but to subsequently cast dispersions upon potential trends in the other taxonomic groups.*” I reanalyzed the results for Asellus first because this taxon was first alphabetically when I sorted the raw data file by taxa. I do not agree that the flat response for Asellus was evident just by looking at the “raw data in Fig. 2”. In my opinion, Fig. 2 is a very difficult figure to decipher (and it should not be included in the MS). I also hope Keva et al. recognize that reanalyzing another author’s raw data is a very time-consuming task, so I only had time to reanalyze the Asellus data before submitting my second review (especially because Nat. Comm. was sending me daily reminders that my review was late).

But now that I have had more time to look into this, I will say that the allochthony responses for 5 of the 19 taxa were flat with regard to PC1 (see table below). The functional responses for the four consumers with the highest estimated allochthony (= 52-88%) were very flat ($r^2 = 0.00-0.08$) (e.g., profundal chironomids, Asellus, profundal BMI, and littoral BMI). The response for copepods was also very flat ($r^2 = 0.06$), but this group also had a negative estimated average allochthony (e.g., -22%). Paradoxically, many of the taxa that showed the strongest functional responses with PC1 ($r^2 = 0.40-0.63$), also had negligible or even quite negative average estimated allochthony (e.g., vendace, medium perch, bulk zooplankton, bleak, large roach, Chaoborus, and small perch). The overall average r^2 values between the estimated consumer allochthony with PC1, forest area and agricultural area were only 0.32, 0.21, and 0.23, respectively. In particular, the relations between forest and agricultural landcover and consumer allochthony were rather weak given the mechanisms hypothesized and broad conclusions from the Keva et al. MS.

Table legend. The first 17 rows below report a correlation (r) matrix between various variables are the calculated allochthony for Keva et al.'s 19 consumers. The last row reports the overall average allochthony calculated for each consumer. The last column (labeled mean) is the average r² value for each of the variables from the correlation matrix.

variable	prof_chiron	Asellus	prof_BMI	litt_BMI	litt_chiron	pike	ruffe	lg_perch	clad	vendace	med_perch	bulk_zoop	bleak	lg_roach	Chaob	copepod	sm_roach	sm_perch	smelt	mean r ²
d15N	-0.489	-0.217	-0.499	-0.321	-0.519	-0.694	-0.550	-0.646	-0.349	-0.728	-0.607	-0.413	-0.559	-0.517	-0.632	-0.161	-0.535	-0.609	-0.376	0.269
TL	-0.362	-0.622	-0.540	-0.593	-0.293	-0.447	-0.017	-0.393	-0.216	-0.061	-0.395	-0.499	-0.296	-0.385	-0.645	-0.363	-0.487	-0.518	-0.145	0.177
For_area	0.060	-0.082	0.115	-0.062	0.388	0.466	0.364	0.642	0.521	0.592	0.630	0.520	0.489	0.534	0.640	0.103	0.379	0.625	0.597	0.214
Agr_area	-0.241	0.028	-0.241	-0.279	-0.396	-0.616	-0.601	-0.608	-0.412	-0.688	-0.616	-0.429	-0.531	-0.570	-0.422	-0.225	-0.536	-0.610	-0.404	0.228
pH	-0.089	-0.017	-0.274	-0.226	-0.580	-0.560	-0.598	-0.714	-0.650	-0.855	-0.690	-0.635	-0.685	-0.641	-0.660	-0.350	-0.635	-0.776	-0.653	0.345
PQM_NC_ratio	-0.288	-0.124	-0.255	-0.267	-0.510	-0.514	-0.537	-0.579	-0.374	-0.687	-0.563	-0.493	-0.601	-0.491	-0.666	-0.041	-0.449	-0.509	-0.403	0.222
PC1	-0.196	0.030	-0.280	-0.157	-0.527	-0.624	-0.558	-0.732	-0.572	-0.793	-0.727	-0.630	-0.661	-0.647	-0.661	-0.247	-0.598	-0.759	-0.555	0.323
PC2	0.115	0.027	-0.200	0.167	0.024	0.180	0.155	-0.012	-0.253	-0.357	0.044	-0.213	-0.009	-0.024	-0.416	-0.094	0.170	-0.074	-0.199	0.033
d2H_cons_corrected	0.887	0.923	0.962	0.848	0.951	0.916	0.938	0.937	0.745	0.966	0.896	0.851	0.905	0.897	0.908	0.825	0.790	0.832	0.568	0.767
aq13C	0.104	-0.022	-0.274	0.138	-0.159	-0.021	-0.038	-0.267	-0.470	-0.496	-0.177	-0.438	-0.115	-0.154	-0.453	-0.104	0.097	-0.321	-0.294	0.071
terr_13C	0.044	0.013	0.437	0.104	-0.197	-0.068	-0.043	-0.024	0.165	-0.241	0.038	0.059	-0.018	0.058	-0.090	0.187	0.088	0.000	0.252	0.024
aq15N	-0.006	-0.018	0.088	-0.098	-0.480	-0.575	-0.486	-0.423	-0.213	-0.559	-0.447	-0.260	-0.363	-0.281	-0.585	-0.017	-0.274	-0.375	-0.237	0.129
terr_15N	-0.073	-0.160	0.246	-0.195	-0.318	-0.255	-0.309	-0.121	0.069	-0.291	-0.074	0.079	-0.039	-0.034	-0.141	0.177	-0.043	0.042	0.405	0.038
aq2H	-0.135	0.347	-0.430	0.157	-0.147	-0.532	-0.457	-0.653	-0.491	-0.672	-0.651	-0.552	-0.490	-0.648	-0.380	-0.441	-0.591	-0.728	-0.760	0.272
terr_2H	-0.272	-0.213	-0.312	0.053	-0.196	-0.376	-0.311	-0.195	0.059	-0.373	-0.161	0.128	-0.182	-0.284	0.072	0.028	-0.129	0.041	0.274	0.049
d2H_water	-0.113	0.348	-0.370	0.146	-0.176	-0.566	-0.464	-0.664	-0.488	-0.672	-0.664	-0.564	-0.507	-0.662	-0.454	-0.456	-0.591	-0.744	-0.772	0.282
wcorr	-0.408	-0.614	-0.540	-0.543	-0.280	-0.444	-0.026	-0.399	-0.219	-0.038	-0.392	-0.484	-0.292	-0.383	-0.648	-0.372	-0.483	-0.516	-0.152	0.174
mean estimated allochthony	0.881	0.705	0.634	0.524	0.495	0.394	0.204	0.178	0.163	0.039	0.013	-0.013	-0.026	-0.042	-0.212	-0.219	-0.231	-0.247	-0.442	0.147

3. Yes, I am arguing that the Law of Mass Conservation is **de facto** correct!

The Law of Mass Conservation is an immutable fact of the world we live in! [In more specific theoretical cases, this law needs to be modified to account for the mass-energy equivalence, for example in nuclear reactions. However, this exception does not apply to Keva et al.'s study]. So anytime the results of a simple mass balance analysis and a Bayesian analysis differ the presumption is that the Bayesian approach has distorted the outcomes in some way relative to

the original raw data. I agree with Keva et al. that the Bayesian approach is partially based on mass balance, but by collapsing the outcomes to the 0-100% domain, it deviates markedly from the correct mass balance. For this reason, I think an algebraic approach should always be used to analyze simpler datasets like the one from Keva et al. The Bayesian approach is a useful approach in more complex cases (e.g., when there are many potential resources and/or many dietary tracers) where the underlying algebra is far more complex. But in any case, it is essential to ascertain to what extent are the model outputs a function of the algorithm itself or the underlying raw data. This is the key conundrum for any environmental modeling exercise, especially Bayesian dietary mixing models (Brett 2014).

4. Am I disdainful of Bayesian mixing models?

In their response, Keva et al. stated *“The Reviewer #2’s disdain for Bayesian mixing models has been apparent throughout, but we have found no reasonable argument against these statistical frameworks presented by the reviewer that stands up to scrutiny.”* I do not have distain for Bayesian dietary mixing models, but I do think these models are often used in a uncritical way that leads to a bias towards everything is somewhat important outcomes (e.g., the classic “plug-and-chug” approach). The essential step that should always be followed is proper model corroboration. **Do the model outputs match up with the original raw data?** Unfortunately, this is very rarely done in the Bayesian mixing model literature. In their second response, Keva et al. state *“we have now calculated also pseudo R^2 values from the predicted tracer values for each of the models according to Gelmann (2018) and Stock et al. (2017). The results can be found in Table S8 . . .”* If I understand correctly, the corroboration results represent *“a posterior approximation of the proportional isotope variance explained by the model”*, in other words, can the model outputs be used to recreate the original raw data. An approximate R^2 value of 0.310 indicates the model outputs did a poor job of recreating the original raw data. I would hope for a corroboration R^2 of >80% for such a simple problem. For example, in my as of yet unpublished fatty acid based mixing model analysis of dietary pathways in the Upper Klamath Lake food web, the corroboration between modeled and observed consumer fatty acid composition was $r^2 = 0.93$. I believe the poor corroboration in Keva et al.’s study is due to the Bayesian algorithm rounding all outcomes to the 0-100% domain. Keva et al. should present the raw results for their corroboration in a much more detailed way, e.g., what is the observed and predicted consumer raw $\delta^2\text{H}$ value for each of their samples so that this could be properly assessed. Without this additional information, my concerns about the Bayesian model’s ability to recreate the original raw data are only reinforced.

To test my conjecture that the poor model corroboration results reported Keva et al. are related to attributes of the Bayesian mixing model framework, I corroborated the mass balance model’s predicted raw medium perch $\delta^2\text{H}$ values with the mass balance allochthony, after setting all < 0% and > 100% allochthony estimates equal to zero and one, respectively. This comparison showed the mass balance model outputs were perfectly corroborated when the allochthony estimates were within the 0-100% domain (as expected) and quite poorly correlated when the perch allochthony estimates were negative. I suspect Keva et al.’s corroboration results were even worse than this because of their choice to use a crossed random effects structure in their model which would smooth the outcomes across consumers and lakes.

5. Does the terrestrial d2H ratio get modified during a microbial pathway?

In their response, Keva et al. argued that it was not necessary to account for multiple trophic levels and environmental water exchange in a microbial pathway from t-DOM through bacteria and protozoa to zooplankton and benthic macroinvertebrates. Keva et al. cited some details from Osburn et al. (2016) and concluded “*Osburn et al. (2016) demonstrated in E.coli that under heterotrophic culturing, environmental water contributions to bacterial amino acids are low, typically <20% except for alanine and no more than 12% for essential AAs (note that AAs make up the bulk of animal tissue biomass). This suggests that microbial trophic steps are less of an issue than the reviewer suggests, and we would caution against applying animal derived parameterisations to bacterial physiology and metabolism in general.*” I just read Osburn et al. (2016) and I have a completely different assessment of that paper than Keva et al. Firstly, it does NOT appear to me that Osburn et al. examined alanine or any other amino acids as Keva et al. claimed. [Specifically, the words alanine and amino acid do not appear in the Osburn et al. (2016) paper]. Instead, the Osburn et al. (2016) study focused on lipid metabolism, or more specifically fatty acids. Osburn et al. (2016) concluded “large variations in the magnitude of [hydrogen] fractionation are observed for many heterotrophic microbes utilizing different central metabolic pathways.” These authors noted that bacterial hydrogen fractionation depends on the bacterial strain, the electron donor (the substrate), and the electron acceptor (O_2 , NO_3^- , SO_4^{2-} , etc.). Osburn et al. (2016) also showed the d2H values of bacterial lipids were very strongly modified by the ambient water (see their Fig. 5). Perhaps Keva et al. confused the paper by Osburn et al. (2016) with a paper by Fogel et al. (2016) which did look at environmental water contributions to bacterial amino acids? Fogel et al. (2016) found that on average, the bacterium *E. coli* obtained 17% of its amino acid hydrogen from environmental water (see Fogel et al.’s Table 1).

Based on the results by Fogel et al. (2016), as well as Osburn et al. (2016), it is very evident that the original d2H values of t-DOM will be extensively modified (perhaps in a difficult to predict manner) as terrestrial organic matter is transferred up through the food web via a microbial

pathway (i.e., t-DOM to bacteria to protozoa to zooplankton). Keva et al. are still ignoring this modification in their Bayesian mixing model, which is directly contradicted by the paper I believe they intended to cite. I suggest Keva et al. assume an environmental water value of 0.17 for the t-DOM to bacteria step. I also suggest using a value of 0.20 (i.e., the mean of 0.17 and 0.23) for the bacteria to protozoa step. Ignoring environmental water contributions along the microbial pathway, as Keva et al. are currently doing, would be a major error.

In their reply to, Keva et al. stated “*Reviewer #2 is correct that we did not account for the microbial link trophic step in the δ^2H correction formulas. The only published way to do this would be correcting the different sources to consumer trophic level (Keva et al. 2022); however, with this setup applying MixSIAR this would be **impossible**.*” This response makes no sense, **of course it is possible** to correct these data for environmental water exchange within the microbial pathway. The very simple mass balance equation to do this is:

$$Bact\ d2H = tOM\ d2H * (1 - \omega) + lakewater\ H2O\ d2H * (\omega).$$

Based on the average t-OM and lakewater d2H values for Keva et al.’s data (i.e., -120.6‰ and -73.9‰, respectively), the recalculated mean d2H values would be -112.6‰ and -104.9‰ for the bacterial and protozoan steps of the microbial pathway. In essence, 34% (= 1 - 0.83*0.80) of the original hydrogen contained in the terrestrial organic matter would get exchanged with hydrogen in the lakewater along the microbial pathway. Unfortunately, these corrections won’t completely resolve this conundrum as the results from Osburn et al. (2016) suggest bacterial metabolism may modify organic matter d2H values in complex ways regarding both fractionation and exchange. Keva et al. should acknowledge that they only have a very rough idea what the d2H values of the terrestrial-microbial pathway are in their lakes and this represents a very large source of uncertainty in their dietary modeling approach. But to be clear, not correcting for and ignoring this source of uncertainty (as Keva et al. advocate doing) does not make it go away.

6. Can tDOM be used a proxy for tPOM and vice versa?

In their second response, Keva et al. suggested tDOM d2H values could be used a proxy for tPOM value and vice versa. I am mostly in agreement here, provided the tDOM and tPOM d2H values are strongly correlated across the 35 lakes. BUT so far, the author have only presented data showing the overall means for their tDOM and tPOM samples are similar. This is not sufficient data to decide whether one can be used as a proxy for the other.

7. Why weren’t real algal d2H values used in the mixing-model?

As discussed in my previous reviews the paper implies that real data were used to represent the algal d2H values, but in the fine print of the methods section the authors mention that “modeled” algal data were actually used for the mixing-model analyses. As I have already noted this was explained in a misleading way, and I cannot envision a justification for using “modeled” data when real data are available! For example, on lines 69-72, the MS states “We used δ^2H to differentiate terrestrial and aquatic OM sources and used Bayesian mixing models to generate estimates of consumer allochthony **by sampling** terrestrial inlet dissolved OM (t-DOM) and **aquatic algae to represent the basal** allochthonous and **autochthonous resources**, respectively”. Subsequently, on

lines 469-473 the authors state “For most lakes, it was impossible to obtain sufficient samples of pure phytoplankton for SIA. Therefore, **the phytoplankton $\delta^2\text{H}$ values were modelled from lake water $\delta^2\text{H}$ values with a photosynthetic discrimination estimate of -106.7‰** derived from the mean difference between measured benthic algae and pelagic water $\delta^2\text{H}$ values across the sampling sites, **assuming that photosynthetic fractionation of hydrogen is equivalent for phytoplankton and benthic algae**”. On lines 360-361 on the MS they state “Benthic algae were sampled by scraping visually green patches of algae from hard substrate surfaces from many locations (e.g. sublittoral rocks from <1 m depth) and pooled on site to gain enough sample material for SIA”.

In their response to my second round of comments the authors state “*it was impossible to obtain sufficient samples of pure phytoplankton for SIA*”. *Once more we try to reason this to Reviewer #2*”. The authors completely misunderstand my point here and I suspect they are employing a **red herring**. I recognize that it could be difficult to obtain a pure sample of phytoplankton for SIA. **That is not my point**. To reiterate, these authors do have benthic algal samples, and from these samples they calculated the average photosynthetic hydrogen discrimination for their lakes is -106.7‰ . They also state that they assume that photosynthetic hydrogen fractionation is equivalent for phytoplankton and benthic algae. If you assume that phytoplankton and benthic algae have equivalent photosynthetic hydrogen fractionation, which averages -106.7‰ , then you are in fact also assuming that within a particular lake the phytoplankton and benthic algae will have the same $\delta^2\text{H}$ values. This equivalency for phytoplankton and benthic algae $\delta^2\text{H}$ values is also reflected in the fact that the raw data provided by the authors for this MS only presents one $\delta^2\text{H}$ value for BOTH phytoplankton and benthic algae for each lake. **My point is that since you do have actual real benthic algal $\delta^2\text{H}$ data, you should use that instead of “modeled” data**. This seems like a very easy place for the authors to modify their approach so that their analysis is based as much as possible on real data. If repeating the analysis using the real observed $\delta^2\text{H}$ for the benthic algae does not change the outcomes of their analysis that would be good to know. **But regardless of the circumstances (Bayesian, frequentist, mass balance), I will always advocate for the use of real data in lieu of “modeled” data**. The authors should also report the raw $\delta^2\text{H}$ benthic algae data for each of their lakes.

The author’s decision to use “modeled” algal $\delta^2\text{H}$ values instead of real data means that in all cases the ambient water and primary producer $\delta^2\text{H}$ values were nearly PERFECTLY CORRELATED (see left-side plot below). This is a completely unrealistic condition. I was curious how strongly the ambient water and algal $\delta^2\text{H}$ values might be correlated if as Keva stated photosynthetic hydrogen discrimination averaged $-106.7 \pm 17.9\text{‰}$ (± 1 SD) and the ambient pelagic water in their lakes had an average $\delta^2\text{H}$ value of $-73.9 \pm 8.7\text{‰}$. I did this by using a random number generator to generate hypothetical photosynthetic hydrogen discrimination and ambient water $\delta^2\text{H}$ distributions with means of $-106.7 \pm 17.9\text{‰}$ and $-73.9 \pm 8.7\text{‰}$ ($n = 1,000$), respectively. I then added the photosynthetic hydrogen discrimination values to the ambient water value to generate “modeled” algal $\delta^2\text{H}$ values. I then regressed the ambient water $\delta^2\text{H}$ values against the modeled algal $\delta^2\text{H}$ values (see the right-side plot below). This Monte Carlo simulation showed that given the inherent variability in the photosynthetic hydrogen discrimination and ambient water $\delta^2\text{H}$ values, there is likely only a weak statistical association between ambient water and modeled

algal d2H values (see right-side plot below). It is also worth pointing out that Keva et al. reported the spatiotemporal variation in within benthic algal d2H values was $\pm 23.6\%$ (± 1 SD), during a time frame when spatiotemporal variation in within lake water d2H values was only $\pm 2.2\%$. Data presented in Table 2 of Brett et al. (2018) similarly showed that when estimated multiple times within a system photosynthetic hydrogen discrimination varied by $\pm \pm 23.1\%$. Based on this evidence, I can say that it was unjustified for Keva et al. to assume that the d2H values for ambient water and the modeled algal in their lakes were perfectly correlated.

If the perfect correlation between the ambient water and modeled algal d2H values did not affect the mixing model outcomes, this point would matter less. But since ambient water and algal d2H values are two of the five variables needed to estimate allochthony in aquatic consumers (with the other variables being the terrestrial resource and consumer d2H values, and the assumed environmental water contribution), it is likely that this quirk of Keva's et al.'s modeling approach also impacts the outcomes. During the process of reviewing the third permutation of this MS, I used a classic algebraic mass balance approach to reanalyze all of the Keva et al.'s consumer allochthony estimates. When doing this I noticed that the ambient lake water d2H values were one of the best predictors of allochthony in each of the 19 consumers that Keva et al. sampled and analyzed (see Table on page 3). Intuitively, this makes no sense. Ambient water d2H values were on average a 4% weaker predictor of allochthony than was PC1 ($r^2 = 0.282$ vs 0.323 , respectively) and a 7% better predictor than forest area ($r^2 = 0.282$ vs 0.214 , respectively). Ambient water d2H values were also a better predictor than agricultural area. This is quite meaningful because one of the main conclusions of the Keva et al. MS is that forest/agricultural land cover determines the degree to which consumers in their lakes are supported by terrestrial organic matter. The statistical results summarized below suggest the d2H value of the ambient water is a substantially better predictor. But how could this be? Why would the d2H values of ambient water be related to the estimated allochthony in these lakes? Surely the deuterium ratio of ambient water would have no mechanistic influence on whether consumers utilize more or less t-OM in these lakes! Conversely, the d2H value of real algal in these lakes should be important

because a more negative algal d2H value in these calculations will directly lead to a lower allochthony estimates in the consumers. By artificially creating a perfect correlation between the ambient water and algal d2H values in their mixing models, I believe the outcomes of these calculations are substantially influenced. Finally, because the ambient water d2H values are also moderately correlated with the forest and wetland areas, pH, etc. in these lakes, one could create a spurious relationship with the “modeled” algal d2H values. Perfectly correlating the d2H data for ambient water and algal data may be part of the cause of the environmental gradient response that Keva et al. reported for their lakes. This conclusion may simply be a result of the authors’ unorthodox approach for obtaining algal d2H values. The only way to resolve this conundrum is to rerun the mixing models with the directly determined benthic algae d2H values, and to provide the raw data for the algal 2dH values.

8. The Bayesian algorithm distorts information contained in the raw data!

One of my most important points in the previous rounds of review is that the problem that Keva et al. are working on is a very simple mass balance calculation that could easily be taught to any undergraduate student in a STEM field. This mass balance problem is based on the Law of Mass Conservation, one of the fundamental principles of physics and chemistry. **By using a Bayesian algorithm to analyze these data, the outcomes are often distorted in ways that contradict this law.** Most importantly, this is completely unnecessary. During the process of reviewing the paper a third time, I reanalyzed all of the Keva et al. data using the correct algebraic approach. In the plot below, I plotted four cases that represented high (littoral benthic macroinvertebrates), intermediate-high (ruffe), intermediate-low (large roach), and low estimated allochthony (small perch). In all cases, I used the exact same assumptions as specified by Keva et al. So, the only difference between the outcomes I obtained and those reported by the authors should be solely due to using an algebraic versus Bayesian approach. As the plots show, the differences in the two approaches are often profound.

I also created a histogram that plotted the estimated allochthony values for all 19 consumers (n = 1,737) (see below). When analyzing the data using an algebraic approach it is clear that a quite large portion (i.e., 44%) of the allochthony estimates were in the negative domain, and 2.3% of the estimates were in the greater than 100% terrestrial resource utilization domain. The overall median estimate for their dataset was 8.5% terrestrial resource utilization. This overall estimate would shift to zero if the environmental water assumption was only slightly increased.

9. What is the proper interpretation of a negative allochthony estimate?

In their response, Keva et al. stated *“taking the argument of Reviewer #2 that all negative proportions are true zeros, in other words the codomain is bounded to biological feasibility, then you can no longer perfectly reconstruct all consumer isotope data as the inverse function is no longer injective – the value of 0% allochthony in their example would now map to many observed consumer $\delta 2H$ values.”* **The authors have completely misunderstood my point.** A small negative allochthony estimate (e.g., -5%) could easily arise due to sampling or measurement error. However, the large portion of strongly negative allochthony estimates indicates that there is a serious problem with Keva et al.’s dataset. Most likely the $\delta 2H$ data for the hypothetical resources are wrong. This is not too surprising given the very indirect way that these authors estimated the $\delta 2H$ values for the terrestrial and aquatic resources in their mixing model. As a famous quote goes *“On two occasions I have been asked, ‘Pray, Mr. Babbage, if you put into the machine wrong figures, will the right answers come out?’ I am not able rightly to apprehend the kind of confusion of ideas that could provoke such a question”* (Charles Babbage). This statement is as true today as it was 150+ years ago. Ironically, in their reply Keva et al. made nearly the same point when they stated *“If the “estimation, observational and measurement error” of these data would be so great as to alone cause such discrepancies, then clearly the data are not suitable to be analysed in such an approach.”*

The problem with a Bayesian approach is it completely obscures obvious problems with model misspecification by rounding all estimates to the 0-100% domain. In the plot below, I present the estimated allochthony values for the classic algebraic approach as well for an approach where all estimates are rounded to fall within the 0-100% range. These outcomes are presented for medium perch (see following plot) because this consumer had nearly the median overall estimated allochthony for the 19 consumers as well as a large sample size ($n = 174$). As this comparison shows, the original raw predictions (see left-side panel) show an overall distribution that hovers close to a mean of zero (i.e., $1 \pm 42\%$, ± 1 SD). Conversely, when the predictions are constrained to 0-100% (see right-side panel) the overall response seems compelling with a mean of $17 \pm 23\%$ and an interesting response pattern with the Environmental gradient represented by PC1. At low values of PC1, the estimated allochthony was 20-60% and at high values of PC1 the estimated allochthony was 5-10%. I will also note that the “adjusted” plot looks very similar to the outcome Keva et al. reported for medium perch. However, the scatter around the trend line and their within lake error bars are much smaller than in my adjusted plot, which is hard to reconcile since the uncertainty shown in my adjusted plot is only due to the within lake variation in point estimates. Since the Keva et al. estimates supposedly also took into account uncertainty the resource stable isotope values, the prediction error for their approach should have been substantially larger than for my approach. Most likely the small scatter around the trend line and smaller error bars for the Keva et al. plot are because these authors crossed the outcomes across “lakes for the same species, and across species from the same lake”.

My overall point is that the superficially convincing results presented in Keva et al.'s Figure 5 (see results for medium perch above) are largely due to a quirk of the Bayesian algorithm that misrepresents the true results of the mass balance calculations by forcing all of the outcomes into the 0-100% domain. It is also problematic that the uncertainty depicted in Keva et al.'s plots was much less than what I found when reanalyzing their data. The true relationship between the Environmental gradient (PC1) and the estimated consumer allochthony is much more ambiguous than they claim because of the large portion of negative estimates.

I want to use another example based on the Keva et al. data to clarify this point. Their data indicated very low allochthony estimates for five consumers (Chaoborus, Copepods, small roach, small perch, and smelt). The average estimate for these consumers combined is -28 ± 39 allochthony, $n = 405$. For these examples, the algebraic analysis is 100% correct, but it suggests a biological outcome that is not possible! We all agree that consumers cannot obtain -28% of their resources from terrestrial sources. What this outcome does show is that in $> 70\%$ of these cases, there is something seriously wrong with the data used to calculate allochthony for these consumers. Either the trophic levels assigned, or the environmental water value assumed, or the ambient water, consumer, and aquatic and terrestrial d2H values used in the calculations are wrong. I am guessing the ambient water and consumer d2H values are solid. I suspect the algal and terrestrial d2H values are the main source of this problem. Knowing that these data give unreasonable outcomes in many cases is very important information.

Recalculated environmental water (ω) estimates

The authors also need to fully detail the protocol that they used to revise the published environmental water values for their analysis. It seems somewhat fortuitous that they originally assumed an environmental water value of 0.22 and after I noted that that value was based on the erroneous results from Wilkinson et al. (2015), Keva et al. revisited this topic and found a correction was necessary for the average of 0.27 ± 0.11 that I published in Brett et al. (2018) and that this correction resulted in a new estimated environmental water value (i.e., 0.23 ± 0.09) that was only slightly different from the value that they originally used. I am no expert on the detailed intricacies of how d2H is quantified with mass spectrometry, and how standards should be employed, etc. But if Keva et al. are going to argue that the best available estimate for environmental water is now based on their reanalysis of the original data (i.e., 0.23 ± 0.09), they need to very carefully explain why some studies were included in their reanalysis and others were excluded. This is done only briefly in the notes for Supplementary Table 9, but not nearly in sufficient detail for someone unfamiliar with this topic. For example, the inclusion of the estimate from Macko et al. (1983) which did not use standards, and exclusion of the estimates from Graham et al. (2014) which did use standards, both had the effect of lowering the recalculated environmental water estimate. If Macko et al. (1983) was excluded and Graham et al. (2014) was included the revised estimate would have been 0.27 ± 0.09 . Why is it necessary to use water labeling experiments to calculate ω ? I am admittedly not an expert on this, but it would seem it would only necessary to know the d2H values of a pure defined diet, the ambient water and the consumer to experimentally determine the environmental water contribution. Additionally, Keva et al. need to specify exactly which equations from Soto et al. (2017) they used to recalculate the

previous environmental water estimates. Without a clearly described protocol, the recalibration process seems quite mysterious and even coincidental.

Access to all raw data

I very much appreciate that the authors have provided access to most of their raw data during the review process. If this MS is finally published in Nature Communications, or elsewhere, **I will officially request access the directly determined benthic algal d2H data for the 35 lakes sampled for this study.** I will also officially request the original uncorrected consumer d2H data, as well as the predicted d2H values for these consumers from their validation analysis.

Minor points

8. The authors conceded my point that herbivorous zooplankton like *Daphnia*, as well as the BMI *Asellus*, most likely have a trophic position of 2.1. If that is true, then a zooplanktivorous fish like a smelt or a vendace that consumes *Daphnia* would have a trophic position of 3.1, and a pike that consumes smelt/vendace would have a trophic position of 4.1. The authors should be consistent in this regard.

9. Fig. S7 appears to show some Cladocera are still assumed to have a trophic position somewhat below 2.0, which is not physiologically possible because Cladocera are not mixotrophic as discussed in my previous review. This appears to also be the case for copepods, profundal chironomids, bulk profundal BMI, littoral chironomids, bulk zooplankton, littoral *Asellus*, and bulk littoral BMI. Maybe this is just an optical illusion? As previously discussed, it is simply not physiologically possible for any of these consumers to truly have trophic positions below 2.0.

10. Keva et al. did not always use a photosynthetic hydrogen discrimination value of -106.7‰ when generating “modeled” algal d2H values as stated in the MS. In lake Jyväsjärvi they used a photosynthetic hydrogen discrimination value of -109.6‰. Jyväsjärvi was the most sampled lake in the dataset, with about 3X more observations than the other lakes on average.

11. In their response, Keva et al. stated *“The last point of this comment 7 is completely incorrect; we explain in the manuscript: “In four lakes (Hirvijärvi, Niemisjärvi, Pankajärvi and Valkea-Kotinen), the cladoceran $\delta^{15}N$ values were lacking and thus they were derived from a linear model based on $\delta^{15}N$ values of cladoceran and bulk zooplankton samples from the other study lakes (adjusted $R^2 = 0.928$, $p < 0.0001$, $\delta^{15}N_{cladocera} = 0.872 \times \delta^{15}N_{bulk\ zooplankton} - 0.095$).”* which Reviewer #2 may have missed while reviewing our manuscript.” Keva et al. are 100% correct here. **I was mistaken, this was properly described in the MS.**

Brett, M.T., G.W. Holtgrieve, and D.E. Schindler. 2018. An assessment of assumptions and uncertainty in deuterium-based estimates of terrestrial subsidies to aquatic consumers. *Ecology* 99: 1073-1088.

Fogel, M. L., Griffin, P. L., & Newsome, S. D. (2016). Hydrogen isotopes in individual amino acids reflect differentiated pools of hydrogen from water and food in bacteria. *Proceedings of the National Academy of Sciences USA*, 113, E4648–E4653.

Osburn, M. R., Dawson, K. S., Fogel, M. L., & Sessions, A. L. (2016). Fractionation of hydrogen isotopes by sulfate- and nitrate-reducing bacteria. *Frontiers in microbiology*, 7, 196727.

Soto, D. X., Koehler, G., Wassenaar, L. I. & Hobson, K. A. 2017. Re-evaluation of the hydrogen stable isotopic composition of keratin calibration standards for wildlife and forensic science applications. *Rapid Commun. Mass Spectrom.* 31, 1193–1203.

Wilkinson, G. M., Cole, J. J. & Pace, M. L. 2015. Deuterium as a food source tracer: Sensitivity to environmental water, lipid content, and hydrogen exchange. *Limnol. Oceanogr. Meth.* 13, 213–223.